# ONE-STEP SCORE-BASED DENSITY RATIO ESTIMATION: SOLVER-FREE WITH ANALYTIC FRAMES

## ABSTRACT

Score-based density ratio estimation is essential for measuring discrepancies between probability distributions, yet existing methods often suffer from high computational costs, requiring many function evaluations to maintain accuracy. We propose One-Step Score-Based Density Ratio Estimation (OS-DRE), an analytic and efficient framework that eliminates the need for numerical solvers. Our approach is based on a spatiotemporal decomposition of the time score function, where its temporal component is represented with an RBF-based (radial basis function) analytic frame. This transforms the intractable temporal integral into a closed-form weighted sum, enabling OS-DRE to estimate density ratios with only one function evaluation while preserving high accuracy. Theoretical analysis provides a rigorous truncation error bounds, ensuring provable accuracy with finite bases. Empirical results show that OS-DRE achieves competitive performance while completing density ratio estimation in a single step, effectively resolving the long-standing accuracy-efficiency trade-off in score-based methods.

## 1 INTRODUCTION

Density ratio estimation (DRE) is a fundamental task in machine learning and statistics, used to quantify the discrepancies between two probability distributions (Sugiyama et al., 2012). It plays a central role in a variety of applications, including continual learning (Zhang et al., 2023), mutual information estimation (Letizia et al., 2024; Chen et al., 2025), Large Language Models (LLMs) alignment (Higuchi & Suzuki, 2025; Xiao et al., 2025), and causal inference (Wang et al., 2025). However, classical DRE faces a significant challenge known as the density-chasm problem, where non-overlapping (Srivastava et al., 2023; Chen et al., 2025) or high-discrepancy distributions (Rhodes et al., 2020; Wang et al., 2025) lead to unstable and inaccurate estimates.

A significant advance has been the emergence of continuous, score-based methods (Choi et al., 2022; Yu et al., 2025; Chen et al., 2025), which reframe the log-density ratio as the path integral of a time-dependent score function along a smooth interpolation between the two distributions (see Fig. 5 for illustration). This continuous formulation transforms the DRE between $p_0$ and $p_1$, i.e., $r(\boldsymbol{x}) = p_1(\boldsymbol{x})/p_0(\boldsymbol{x})$, into solving the integral $\log r(\boldsymbol{x}) = \int_0^1 \partial_t \log p_t(\boldsymbol{x}) \mathrm{d}t$, with $\partial_t \log p_t(\boldsymbol{x})$ being the *time score*. While this mitigates the density-chasm problem, existing score-based methods still rely on computationally expensive numerical integration techniques, including ODE solvers (Choi et al., 2022) and fine-grained quadratures (Norcliffe & Deisenroth, 2023). Achieving reliable estimates requires many repeated score evaluations, which results in a high number of function evaluations (**NFE**) and substantial computational overhead.

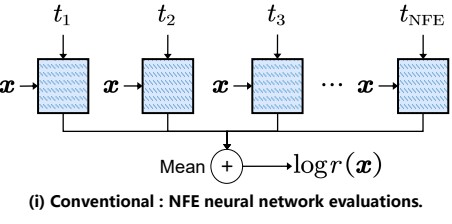

(i) Conventional : NFE neural network evaluations.

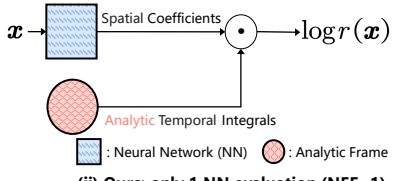

(ii) Ours: only 1 NN evaluation (NFE=1).

Figure 1: Illustrative comparison of conventional (i) and proposed (ii) score-based DRE methods. The NFE of conventional methods depends on the chosen numerical solver.

In this work, we introduce *One-Step Score-Based Density Ratio Estimation* (**OS-DRE**), a novel framework that is *solver-free* and computes the density ratio integral in a single step. Our key

innovation is to replace numerical integration with an analytic solution. We achieve this by proposing a spatiotemporal decomposition of the time score, where we represent its temporal component using an *analytic frame*, which is a mathematical frame whose elements $\{g_k\}_{k=1}^{\infty}$ possess closed-form temporal integrals. This allows us to re-express the integral as a simple weighted sum:

$$\log r(\boldsymbol{x}) = \int_0^1 \partial_t \log p_t(\boldsymbol{x}) \mathrm{d}t = \sum_{k=1}^{\infty} h_k(\boldsymbol{x}) \int_0^1 g_k(t) \mathrm{d}t \approx \left\langle \boldsymbol{h}^{(K)}(\boldsymbol{x}), \bar{\boldsymbol{g}}^{(K)} \right\rangle, \tag{1}$$

where $\boldsymbol{h}^{(K)}(\boldsymbol{x}) = [h_1^{(K)}(\boldsymbol{x}), h_2^{(K)}(\boldsymbol{x}), \ldots, h_K^{(K)}(\boldsymbol{x})]$ and $\bar{\boldsymbol{g}}^{(K)} = [\bar{g}_1, \bar{g}_2, \ldots, \bar{g}_K]$ are two $K$-dimensional vectors with $\bar{g}_k = \int_0^1 g_k(t)\mathrm{d}t$. As illustrated in Fig. 1, our method employs a neural network (blue squares) to predict the spatial coefficients $\boldsymbol{h}^{(K)}(\boldsymbol{x})$ in a single forward pass, which then weight the pre-computed, analytic integrals of our frame elements (red circles). This approach reduces the NFE to just **one**, drastically improving computational efficiency.

Our framework is grounded in rigorous approximation theory. We construct our analytic frames using radial basis functions (RBFs) and prove that this construction yields a temporal basis that is both *complete* in the infinite limit (guaranteeing convergence) and *stable* for any finite number of basis functions (ensuring numerical robustness). Furthermore, we provide a theoretical truncation error bound, which guarantees that the approximation accuracy can be systematically controlled. The main contributions of this work are:

- We propose OS-DRE, the first path-integral-based, score-based DRE method that analytically estimates the density ratio in a single step, eliminating the need for numerical solvers while preserving the flexibility of the continuous formulation.
- We introduce the concept of an analytic frame, a temporal basis with closed-form integrals, which enables the direct, analytic computation of the density ratio integral.
- We provide a complete theoretical framework for our method, including proofs for the completeness of our RBF-based construction, and a formal analysis of the truncation error.
- We validate OS-DRE through extensive experiments, demonstrating that it achieves competitive performance on several benchmark datasets with unparalleled computational efficiency.

## 2 RELATED WORKS AND PROBLEM STATEMENT

**Related Works.** Density Ratio Estimation (DRE) is a cornerstone task in machine learning (Sugiyama et al., 2012). Research in this area has largely followed two main trajectories. The first line of work is based on discriminative or contrastive objectives, such as in KLIEP (Sugiyama et al., 2012), NCE (Gutmann & Hyvärinen, 2012), and TR-DRE (Liu et al., 2017). While foundational, these methods often struggle with the "density-chasm" problem, where distributions with high discrepancies or complex settings lead to unstable training and poor estimates. Subsequent efforts to mitigate this issue within the same paradigm, such as FDRE (Choi et al., 2021), iterated regularization (Gruber et al., 2024), $\gamma$-DRE (Nagumo & Fujisawa, 2024), IMDRE (Kimura & Bondell, 2025) and PP-DRE (Wang et al., 2025), have often introduced significant computational overhead by requiring complex machinery like normalizing flows or additional importance sampling steps. A second, more recent line of research, known as score-based DRE, emerged as a powerful alternative for resolving the density-chasm problem. Pioneered by methods like TRE (Rhodes et al., 2020) and DRE-$\infty$ (Choi et al., 2022), this approach reframes the problem in a continuous setting, which inherently provides a smoother and more stable estimation path between the two distributions. Despite their robustness, subsequent innovations within this paradigm (Chen et al., 2025; Yu et al., 2025) have remained dependent on computationally expensive numerical solvers for integration. A related line of work, Guth et al. (2025), trains a time-varying energy via dual score matching, enabling single-step evaluation of normalized log-densities when $p_0$ or $p_1$ is Gaussian and global calibration is available. However, it does not address general DRE between two arbitrary non-Gaussian distributions, where aligning the global constants of separately trained energy models is infeasible. Our work, OS-DRE, operates within the robust score-based framework of the path integral but is the first to propose a *solver-free* approach that entirely eliminates the computational bottleneck by providing an analytic solution to the integral itself, thereby preserving the core advantages of the integration paradigm.

**Problem Statement.** Modern score-based DRE reframes the estimation of the ratio $r(\boldsymbol{x}) = p_1(\boldsymbol{x})/p_0(\boldsymbol{x})$ as the computation of an integral. By constructing a continuous path of densities $p_t(\boldsymbol{x})$ that interpolates between $p_0$ and $p_1$, the log-density ratio is expressed as:

$$\log r(\boldsymbol{x}) = \log \frac{p_1(\boldsymbol{x})}{p_0(\boldsymbol{x})} = \log p_1(\boldsymbol{x}) - \log p_0(\boldsymbol{x}) = \int_0^1 \partial_t \log p_t(\boldsymbol{x}) \mathrm{d}t. \tag{2}$$

The integrand, $\partial_t \log p_t(\boldsymbol{x})$, is known as the *time score*. In practice, the true time score is unknown and is approximated by a neural network, $s_t^{\boldsymbol{\theta}}(\boldsymbol{x}, t)$, trained to match the true score, typically by minimizing a time score matching (TSM) objective (Choi et al., 2022). After training, the log-density ratio is estimated by computing the integral of the learned score model: $\log \hat{r}(\boldsymbol{x}) = \int_0^1 s_t^{\boldsymbol{\theta}^\star}(\boldsymbol{x}, t) \mathrm{d}t$. ***The central problem addressed in this work is the computational bottleneck of this final step***. Existing methods rely on expensive numerical techniques like ODE solvers or quadratures to approximate this integral, requiring numerous iterative evaluations of the model $s_t^{\boldsymbol{\theta}^\star}$. Our goal is to develop a method that computes this integral accurately and efficiently, without resorting to any numerical solvers.

## 3 ONE-STEP DENSITY RATIO ESTIMATION

**Notations.** Let $\mathcal{S} \triangleq \{s_t \mid s_t(\boldsymbol{x}, t) \triangleq \partial_t \log p_t(\boldsymbol{x}), \ \boldsymbol{x} \in \mathcal{X}, \ p_t \in \mathcal{P}(\mathcal{X}), \ t \in [0, 1]\}$ denote the set of *time score* functions, where $\mathcal{P}(\mathcal{X})$ is a family of probability densities over the sample space $\mathcal{X}$.

To ensure the analytical tractability of the time score function $s_t$, we impose some mild regularity conditions on the probability density $p_t(\boldsymbol{x})$, which are detailed in Sec. A.1. Under these conditions, the space $\mathcal{S}$ is embedded in the Hilbert space $L^2(\mathcal{X} \times [0, 1])$.

**Lemma 3.1.** *Under Assumptions A.1 and A.2, the space $\mathcal{S}$ is a subset of $L^2(\mathcal{X} \times [0, 1])$.*

See Sec. A.3 for a detailed proof. This embedding allows us to leverage the tools of Hilbert space theory to analyze and approximate the time score function $s_t$.

For notational convenience, we denote $L^2(\mathcal{X} \times [0, 1])$ by $\mathcal{H}_{\boldsymbol{x}, t}$ throughout this paper. The Hilbert space $\mathcal{H}_{\boldsymbol{x}, t}$ is isometrically isomorphic to the Hilbert tensor product $\mathcal{H}_{\boldsymbol{x}} \hat{\otimes} \mathcal{H}_t$ (Kadison & Ringrose, 1986), where $\mathcal{H}_{\boldsymbol{x}} \triangleq L^2(\mathcal{X})$ and $\mathcal{H}_t \triangleq L^2([0, 1])$. This equivalence guarantees that any time score $s_t \in \mathcal{S}$ can be represented by separating its spatial and temporal components (see Lemma A.3).

**Decomposition via Orthonormal Bases.** We first propose to represent the temporal component of the time score using a complete orthonormal basis $\{g_k\}_{k=1}^\infty$ for $\mathcal{H}_t$. This decomposition, analogous to the Karhunen-Loève expansion (Karhunen, 1947; Loève, 1977), allows us to express the time score $s_t(\boldsymbol{x}, t)$ for each fixed $\boldsymbol{x}$ as a weighted sum of its spatial and temporal components:

$$s_t(\boldsymbol{x}, t) = \sum_{k=1}^\infty h_k(\boldsymbol{x}) g_k(t), \quad \text{where } h_k(\boldsymbol{x}) = \langle s_t(\boldsymbol{x}, \cdot), g_k \rangle_{\mathcal{H}_t} \triangleq \int_0^1 s_t(\boldsymbol{x}, t) g_k(t) \mathrm{d}t. \quad (3)$$

By integrating this series with respect to time, we derive our initial formulation for the $\log r(\boldsymbol{x})$.

**Lemma 3.2.** *Let $\{g_k\}_{k=1}^\infty$ be a complete orthonormal basis for $\mathcal{H}_t$. The target log-density ratio $\log r(\boldsymbol{x})$ can be estimated by:*

$$\log r(\boldsymbol{x}) = \sum_{k=1}^\infty h_k(\boldsymbol{x}) \int_0^1 g_k(t) \mathrm{d}t. \quad (4)$$

See Sec. A.4 for details. While theoretically sound, this approach faces a critical practical limitation. For many standard orthonormal bases (e.g., Fourier or Legendre bases), all basis elements except the constant function (say, $g_1$) have zero integrals, i.e., $\int_0^1 g_k(t) \mathrm{d}t = \langle g_k, 1 \rangle_{\mathcal{H}_t} = 0, \forall k > 1$, causing the expansion of Eq. (4) to collapse to a single term and discard high-frequency information.

**Generalization via Frame-Based Decomposition.** To resolve this degeneracy, we relax the strict orthogonality condition and adopt a more flexible ***frame*** for $\mathcal{H}_t$. Frames retain the completeness property of orthonormal bases but allow for redundancy and non-orthogonality, enabling the use of elements with non-zero integrals.

**Definition 3.3** (Frame, Mallat (2009)). *Let $\mathcal{H}$ be a Hilbert space with inner product $\langle \cdot, \cdot \rangle_{\mathcal{H}}$. A sequence $\{g_k\}_{k=1}^\infty$ in a Hilbert space $\mathcal{H}$ is a frame if there exist constants $0 < A \le B < \infty$, called the frame bounds, such that for any $g \in \mathcal{H}$:*

$$A \|g\|_{\mathcal{H}}^2 \le \sum_{k=1}^\infty |\langle g, g_k \rangle_{\mathcal{H}}|^2 \le B \|g\|_{\mathcal{H}}^2. \quad (5)$$

The frame bounds ensure that $\{g_k\}_{k=1}^\infty$ provides a stable representation of any $g \in \mathcal{H}$, even if the frame elements $g_k$ are not linearly independent Mallat (2009).

By employing frames for both the spatial and temporal spaces, we arrive at our final, powerful representation for the time score and log-density ratio.

**Theorem 3.4.** *Let $\{f_l\}_{l=1}^{\infty}$ and $\{g_k\}_{k=1}^{\infty}$ be frames for $\mathcal{H}_{\boldsymbol{x}}$ and $\mathcal{H}_t$, respectively. Then, any time score function $s_t \in \mathcal{S}$ can be expressed as:*

$$s_t(\boldsymbol{x}, t) = \sum_{k=1}^{\infty} \sum_{l=1}^{\infty} c_{l,k} f_l(\boldsymbol{x}) g_k(t), \tag{6}$$

*where the coefficients $c_{l,k}$ depend on $s_t$. By defining spatial coefficients $h_k(\boldsymbol{x}) \triangleq \sum_{l=1}^{\infty} c_{l,k} f_l(\boldsymbol{x})$ and integral $\bar{g}_k \triangleq \int_0^1 g_k(t)\mathrm{d}t$, the time score and the corresponding log-density ratio can be expressed as:*

$$s_t(\boldsymbol{x}, t) = \sum_{k=1}^{\infty} h_k(\boldsymbol{x}) g_k(t), \quad \log r(\boldsymbol{x}) = \sum_{k=1}^{\infty} h_k(\boldsymbol{x}) \bar{g}_k. \tag{7}$$

This representation resolves the degeneracy issue, as the integrals $\bar{g}_k$ are generally non-zero for all $k$ if $\{g_k\}_{k=1}^{\infty}$ is a frame. Furthermore, Theorem 3.4 allows for the computation of derivatives.

**Corollary 3.5.** *If each function $g_k$ in the frame expansion belongs to the Sobolev space $\mathcal{W}^{1,2}([0,1])$ and the coefficients $\{h_k(\boldsymbol{x})\}$ are such that the series $\sum_{k=1}^{\infty} h_k(\boldsymbol{x}) g_k'(t)$ converges in $\mathcal{H}_{\boldsymbol{x},t}$, then the weak derivative of the time score $s_t$ with respect to $t$ exists and is given by term-by-term differentiation:*

$$\partial_t s_t(\boldsymbol{x}, t) = \sum_{k=1}^{\infty} h_k(\boldsymbol{x}) g_k'(t). \tag{8}$$

See Sec. A.5 and Sec. A.6 for the proofs of Theorem 3.4 and Corollary 3.5, respectively.

The infinite-dimensional representation in Theorem 3.4, while theoretically powerful, is not directly computable. This necessitates a transition to a practical, finite-dimensional approximation. The subsequent section is dedicated to this crucial step, detailing the construction of a suitable temporal basis $\{g_k\}_{k=1}^{\infty}$ using RBFs (Sec. 4.1) and providing a theoretical analysis of the error introduced by truncating the series to a finite number of terms (Sec. 4.2).

# 4 CONSTRUCTING THE TEMPORAL BASIS FOR OS-DRE

In the preceding section, we established the theoretical foundation for our method using an infinite-dimensional series expansion (Theorem 3.4). To operationalize this framework, we now transition from the infinite-dimensional ideal to a practical, finite-dimensional approximation scheme. This section details the construction of this scheme, analyzes its theoretical error bounds, and presents concrete examples of the basis functions used.

## 4.1 THE FINITE-DIMENSIONAL APPROXIMATION SCHEME

**General RBF Construction.** The core idea of our scheme is to project the target function onto a sequence of nested, finite-dimensional subspaces $\{V_K\}_{K=1}^{\infty}$, where each $V_K$ is spanned by a set of RBFs. For this scheme to be a valid and stable implementation of the frame-based theory from Sec. 3, the chosen RBF family $\{g_k\}_{k=1}^{\infty}$ must inherit the two essential properties of a mathematical frame: *completeness*, which ensures the approximation can converge, and *stability*, which ensures the computation is robust. This leads to the following formal requirements.

**Proposition 4.1.** *Let $\{g_k\}_{k=1}^{\infty}$ be an infinite family of RBFs in $\mathcal{H}_t$, defined by $g_k(t) = \phi(|t - c_k|/\sigma_k)$. $c_k$ and $\sigma_k$ are the center and shape paramaters of $g_k$. This family generates a convergent and well-posed approximation scheme if it meets two conditions: (i) Denseness: The infinite family's linear span is dense in $\mathcal{H}_t$, i.e., $\overline{\mathrm{span}\{g_k\}_{k=1}^{\infty}} = \mathcal{H}_t$. (ii) Finite-dimensional stability: For any finite $K \geq 1$, the subset $\{g_k\}_{k=1}^{K}$ is linearly independent.*

see Sec. A.8 for a detailed proof. Proposition 4.1 provides a clear blueprint for our construction. The abstract conditions (i) and (ii) can be satisfied by imposing concrete requirements on the RBF generating function $\phi$. Specifically, the denseness condition (i) is fulfilled when $\phi$ and the RBF parameters are chosen to satisfy the premises of our Denseness Lemma (Lemma A.4). The stability condition (ii) is ensured by requiring $\phi$ to correspond to a *strictly positive definite kernel*, which guarantees that the basis functions generated from distinct centers are linearly independent.

With these requirements for $\phi$ in mind, we construct our basis functions in the general form $g_k(t) = \phi(|t - c_k|/\sigma_k)$. A key advantage of this approach is the potential for closed-form expressions for their integrals and derivatives, which are crucial for our application. The temporal integral $\bar{g}_k$ and derivative $g'_k(t)$ are given by:

$$\bar{g}_k = \int_0^1 \phi\left(\frac{|t - c_k|}{\sigma_k}\right) \mathrm{d}t, \quad g'_k(t) = \frac{\mathrm{sgn}(t - c_k)}{\sigma_k}\phi'\left(\frac{|t - c_k|}{\sigma_k}\right). \tag{9}$$

**Application to Time Score Approximation.** We now connect this approximation scheme back to the central goal of our work. For a fixed $\boldsymbol{x}$, we approximate the true time score $s_t(\boldsymbol{x}, t)$ by its orthogonal projection onto the finite-dimensional subspace $V_K = \mathrm{span}\{g_k\}_{k=1}^K$. Let this approximation be $s_t^{(K)}(\boldsymbol{x}, t)$. The stability guaranteed by Proposition 4.1 ensures that the coefficients $\{h_k^{(K)}(\boldsymbol{x})\}_{k=1}^K$ in the expansion are unique and can be robustly computed. By integrating this finite expansion, we obtain a practical, computable approximation for the log-density ratio, denoted $\log r^{(K)}(\boldsymbol{x})$:

$$s_t(\boldsymbol{x}, t) \approx s_t^{(K)}(\boldsymbol{x}, t) = \sum_{k=1}^K h_k^{(K)}(\boldsymbol{x})g_k(t), \quad \log r(\boldsymbol{x}) \approx \log r^{(K)}(\boldsymbol{x}) = \sum_{k=1}^K h_k^{(K)}(\boldsymbol{x})\bar{g}_k. \tag{10}$$

These equations form the basis of our numerical implementation. The subsequent sections will detail specific choices for the generating function $\phi$ and analyze the error introduced by this truncation.

## 4.2 TRUNCATION ERROR ANALYSIS

The truncation of the infinite series to a finite sum of $K$ terms introduces an approximation error. We now provide a rigorous theoretical analysis of this error.

**Convergence Rates for RBF Approximation.** The convergence rate of the error $\|s_t - s_t^{(K)}\|_{\mathcal{H}_t}$ depends on the interplay between the smoothness of the target function $s_t$ and the regularity of the RBF generating function $\phi$. This regularity is characterized by the kernel's native space $\mathcal{N}_\phi$, the Reproducing Kernel Hilbert Space (RKHS) for which the kernel of $\phi$ serves as the reproducing kernel. Informally, it consists of functions that are naturally smooth with respect to $\phi$. To derive a rigorous error bound, we link this native space to standard Sobolev spaces, following the foundational work on Sobolev error estimates for RBFs (Narcowich et al., 2006).

**Proposition 4.2.** *Let the RBF generating function $\phi$ be such that its native space $\mathcal{N}_\phi$ is equivalent to $\mathcal{W}^{\tau,2}(\mathbb{R})$ for some $\tau > 1/2$. Let the target function $s_t(\boldsymbol{x}, \cdot)$ belong to a Sobolev space of lower smoothness, $s_t(\boldsymbol{x}, \cdot) \in \mathcal{W}^{\beta,2}([0,1])$ with $1/2 < \beta \leq \tau$. Let $s_t^{(K)}(\boldsymbol{x}, \cdot)$ be the best approximation of $s_t$ in the subspace $V_K = \mathrm{span}\{g_k\}_{k=1}^K$, where the centers $\mathcal{C}_K = \{c_k\}_{k=1}^K$ are quasi-uniform. Then, there exists a constant $C$, independent of $s_t$ and $K$, such that the approximation error is bounded by:*

$$\|s_t(\boldsymbol{x}, \cdot) - s_t^{(K)}(\boldsymbol{x}, \cdot)\|_{\mathcal{H}_t} \leq C \cdot K^{-\beta} \cdot \|s_t(\boldsymbol{x}, \cdot)\|_{\mathcal{W}^{\beta,2}([0,1])}. \tag{11}$$

See Sec. A.9 for details. This proposition complements Proposition 4.1 by establishing a quantitative convergence rate. It introduces a third requirement for the generating function $\phi$, the native space condition, which ensures rapid convergence of the approximation error for smooth target functions.

## 4.3 A SUITE OF ANALYTIC RBF KERNELS

We conclude with specific choices for the RBF generating function $\phi$. In our implementation, the centers $\{c_k\}_{k=1}^K$ are fixed to a quasi-uniform grid over $[0, 1]$ (e.g., equispaced points), while the shape parameters $\{\sigma_k\}_{k=1}^K$ are learnable. This design satisfies the denseness and quasi-uniformity conditions from our theory. The required linear independence is guaranteed when $\phi$ corresponds to a *strictly positive definite kernel*. The kernels below are chosen because they meet the denseness, stability and native space conditions and admit closed-form integrals and derivatives.

**Example 1: Gaussian RBFs.** The generating function is $\phi(r) = \exp(-r^2)$. It satisfies all theoretical requirements: (1) continuity and integrability for the Denseness Lemma, (2) strict positive

definiteness for stability, and (3) infinite smoothness, making its native space equivalent to $W^{\tau,2}$ for any $\tau > 1/2$. The basis function is $g_k(t) = \exp\left(-\frac{|t-c_k|^2}{\sigma_k^2}\right)$, with integral and derivative:

$$\bar{g}_k = \frac{\sigma_k\sqrt{\pi}}{2}\left[\text{erf}\left(\frac{1-c_k}{\sigma_k}\right) + \text{erf}\left(\frac{c_k}{\sigma_k}\right)\right], \ \ g_k'(t) = -\frac{2(t-c_k)}{\sigma_k^2}g_k(t), \tag{12}$$

where $\text{erf}(\cdot)$ is the error function, $\text{erf}(z) = \frac{2}{\sqrt{\pi}}\int_0^z \exp(-x^2)\text{d}x$.

**Example 2: Inverse Multiquadric RBFs.** The generating function is $\phi(r) = (r^2 + 1)^{-1/2}$, which likewise meets the three conditions: (1) denseness, (2) stability via strict positive definiteness, and (3) infinite smoothness, ensuring a Sobolev-equivalent native space. The basis function is $g_k(t) = \frac{\sigma_k}{\sqrt{(t-c_k)^2+\sigma_k^2}}$, with integral and derivative:

$$\bar{g}_k = \sigma_k \ln\left(\frac{(1-c_k) + \sqrt{(1-c_k)^2 + \sigma_k^2}}{-c_k + \sqrt{c_k^2 + \sigma_k^2}}\right), \ \ g_k'(t) = -\frac{\sigma_k(t-c_k)}{((t-c_k)^2 + \sigma_k^2)^{3/2}}. \tag{13}$$

We also implement other RBFs, including rational quadratic and Matérn kernels (detailed in Sec. B).

### 4.4 Training Objective and Computational Advantages

To implement OS-DRE, we parameterize the spatial coefficients $\{h_k^{(K)}(\boldsymbol{x})\}$ using a single neural network with parameters $\boldsymbol{\theta}$. Given an input sample $\boldsymbol{x}$, the network outputs $K$ coefficients: $[h_1^{\boldsymbol{\theta}}(\boldsymbol{x}), \ldots, h_K^{\boldsymbol{\theta}}(\boldsymbol{x})] = \text{NN}(\boldsymbol{x}; \boldsymbol{\theta})$. Our time score model and its derivative and integral are given by:

$$s_t^{\boldsymbol{\theta}}(\boldsymbol{x}, t) = \sum_{k=1}^K h_k^{\boldsymbol{\theta}}(\boldsymbol{x})g_k(t), \ \ \partial_t s_t^{\boldsymbol{\theta}}(\boldsymbol{x}, t) = \sum_{k=1}^K h_k^{\boldsymbol{\theta}}(\boldsymbol{x})g_k'(t), \ \ \int_0^1 s_t^{\boldsymbol{\theta}}(\boldsymbol{x}, t)\text{d}t = \sum_{k=1}^K h_k^{\boldsymbol{\theta}}(\boldsymbol{x})\bar{g}_k. \tag{14}$$

We train this model by minimizing the sliced time score matching (STSM) objective from Choi et al. (2022), a tractable objective function independent of the unknown true score:

$$\mathcal{L}_{\text{STSM}}(\boldsymbol{\theta}) = 2\mathbb{E}_{p_0(\boldsymbol{x}_0)p_1(\boldsymbol{x}_1)}\left[\lambda(0)s_t^{\boldsymbol{\theta}}(\boldsymbol{x}_0, 0) - \lambda(1)s_t^{\boldsymbol{\theta}}(\boldsymbol{x}_1, 1)\right]$$
$$+ \mathbb{E}_{p(t)p_t(\boldsymbol{x})}\left[2\lambda(t)\partial_t s_t^{\boldsymbol{\theta}}(\boldsymbol{x}, t) + 2\lambda'(t)s_t^{\boldsymbol{\theta}}(\boldsymbol{x}, t) + \lambda(t)s_t^{\boldsymbol{\theta}}(\boldsymbol{x}, t)^2\right], \tag{15}$$

where $p(t) = \mathcal{U}[0, 1]$ and $\lambda(\cdot) : [0, 1] \to \mathbb{R}_+$ is a weighting function with $\lambda'$ being its derivative.

Our framework offers key computational benefits in both training and inference. During training, the derivative term $\partial_t s_t^{\boldsymbol{\theta}}$ is computed analytically using Eq. (14). This eliminates the need for automatic differentiation w.r.t. $t$, which in prior work (e.g., DRE-$\infty$) required expensive second-order gradients. By reducing optimization to a first-order problem, we enable faster and more stable training.

Once the optimal parameters $\boldsymbol{\theta}^\star$ are found, the log-density ratio is estimated in a single step:

$$\log \hat{r}(\boldsymbol{x}) = \int_0^1 s_t^{\boldsymbol{\theta}}(\boldsymbol{x}, t)\text{d}t = \sum_{k=1}^K h_k^{\boldsymbol{\theta}^\star}(\boldsymbol{x})\bar{g}_k = \left\langle \text{NN}(\boldsymbol{x}; \boldsymbol{\theta}^\star), \bar{\boldsymbol{g}}^{(K)} \right\rangle, \tag{16}$$

where $\langle \cdot, \cdot \rangle$ denotes the inner product operator, $\text{NN}(\boldsymbol{x}; \boldsymbol{\theta}^\star) = [h_1^{\boldsymbol{\theta}^\star}(\boldsymbol{x}), h_2^{\boldsymbol{\theta}^\star}(\boldsymbol{x}), \ldots, h_K^{\boldsymbol{\theta}^\star}(\boldsymbol{x})]$ and $\bar{\boldsymbol{g}}^{(K)} = [\bar{g}_1, \bar{g}_2, \ldots, \bar{g}_K]$ are two vectors. Since the basis integrals $\bar{g}_k$ are pre-computed analytic constants (e.g., Eq. (12)), estimation requires only a single forward pass to obtain the coefficients, **leading to only 1 NFE** and offering a substantial speedup over iterative ODE-based or quadrature methods. The training and inference procedures of OS-DRE are summarized in Algorithms 1 and 2. See Algorithm 3 for details of training procedure and Algorithm 4 for Pytorch implementation.

| **Algorithm 1** One Training Step of OS-DRE | **Algorithm 2** One-Step Estimation |
|---|---|
| **Input:** A batch $\boldsymbol{x}_0 \sim p_0, \boldsymbol{x}_1 \sim p_1, t \sim \mathcal{U}(0,1)$. | **Input:** Sample $\boldsymbol{x}$ and pre-calculated $\{\bar{g}_k\}_{k=1}^K$. |
| 1: Derive $\boldsymbol{x}_t$ with $(\boldsymbol{x}_0, \boldsymbol{x}_1)$ (see Sec. C.1.3). | **Output:** Estimated log-density ratio $\log \hat{r}(\boldsymbol{x})$. |
| 2: Compute $s_t^{\boldsymbol{\theta}}$ and $\partial_t s_t^{\boldsymbol{\theta}}$ using Eq. (14). | 1: $\{h_k^{\boldsymbol{\theta}^\star}(\boldsymbol{x})\}_{k=1}^K \leftarrow \text{NN}(\boldsymbol{x}; \boldsymbol{\theta}^\star)$. |
| 3: Compute loss $\mathcal{L}_{\text{STSM}}(\boldsymbol{\theta})$ using Eq. (15). | 2: Compute $\log \hat{r}(\boldsymbol{x})$ using Eq. (16). |
| 4: Update trainable parameters $\boldsymbol{\theta}$ and $\{\sigma_k\}_{k=1}^K$. | |

## 5 EXPERIMENTAL SETTINGS AND RESULTS

We conduct extensive experiments to evaluate OS-DRE, using DRE-$\infty$ (Choi et al., 2022) and D$^3$RE (Chen et al., 2025) as baselines. For fair comparison, all methods adopt the same quadrature scheme (trapezoidal rule) and weighting function $\lambda(t) = t(1 - t)$.

**Density Estimation.** In density estimation, let $p_0(\boldsymbol{x}) = \mathcal{N}(\mathbf{0}, \boldsymbol{I}_d)$ be a simple noise distribution, and $p_1(\boldsymbol{x})$ denote the complex and intractable data distribution. The log-likelihood of $p_1$ for a given sample $\boldsymbol{x}$ can be estimated as $\log p_1(\boldsymbol{x}) = \log r(\boldsymbol{x}) + \log p_0(\boldsymbol{x})$, where $r(\boldsymbol{x}) = p_1(\boldsymbol{x})/p_0(\boldsymbol{x})$ is the density ratio between $p_1$ and $p_0$. After training, the estimated log-density ratio $\log \hat{r}$ can be derived based on Eq. (16). Thus, the log-likelihood of $p_1$ can be estimated as $\log p_1(\boldsymbol{x}) \approx \log \hat{r}(\boldsymbol{x}) + \log p_0(\boldsymbol{x})$. Detailed experimental settings can be found in Sec. C.3.

STRUCTURED AND MULTIMODAL DATASETS. We evaluate OS-DRE on nine standard synthetic benchmarks (Bansal et al., 2023; Chen et al., 2025). Results are shown part in Fig. 2 and full in Fig. 7 (Sec. C.3). As shown in Fig. 2, our solver-free method achieves accurate density estimates with only one function evaluation (NFE = 1), while DRE-$\infty$ and D$^3$RE with NFE = 2 often yield blurred or distorted results. OS-DRE reliably captures challenging structures, including disconnected rings (circles), curved manifolds (swissroll), sharp discontinuities (checkerboard), and branching topologies (tree). These results demonstrate that OS-DRE learns complex multimodal densities efficiently under tight inference constraints.

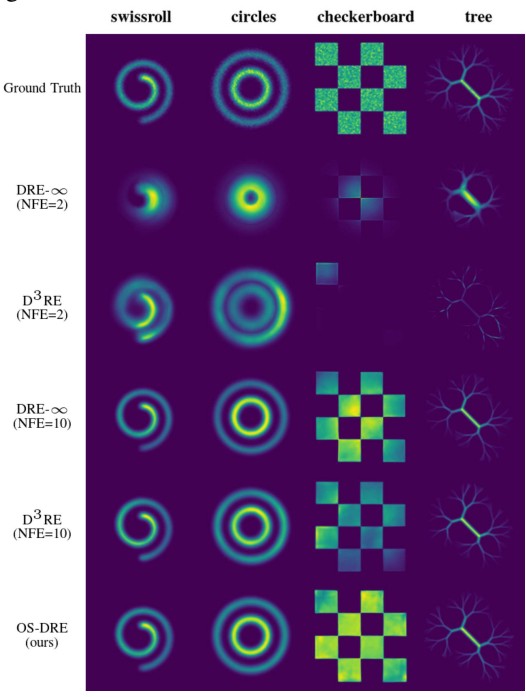

Figure 2: Comparison of density estimates from three score-based DRE methods on four structured and multimodal datasets. DRE-$\infty$ and D$^3$RE use NFE = 2, whereas our OS-DRE uses only NFE = 1. Additional results are in Fig. 7 (Sec. C.3). OS-DRE is the best one with lowest NFE.

REAL-WORLD TABULAR DATASETS. We further test OS-DRE on five real-world tabular datasets (Grathwohl et al., 2019), spanning domains from physics to image statistics. As shown in Tab. 1, our one-step method (NFE = 1) not only demonstrates remarkable efficiency but also achieves state-of-the-art (SOTA) performance in terms of negative log-likelihood (NLL). Notably, OS-DRE with the IMQ and RQ kernels consistently outperforms the baseline methods across all datasets, even when the baselines are allocated a significantly larger computational budget (NFE up to 50). This result is particularly pronounced on the high-dimensional MINIBOONE and BSDS300 datasets, where OS-DRE achieves superior accuracy with a fraction of the model parameters used by competing methods. These findings confirm that our analytic framework successfully resolves the efficiency-accuracy trade-off, delivering both speed and SOTA performance.

Table 1: Comparison of negative log-likelihood (NLL) and wall-clock time on five tabular datasets. Lower is better. All timing results were obtained on a single NVIDIA TITAN X GPU. The best NLL in each dataset is shown in **bold**, and the best wall-clock time is underlined.

| Method | NFE | RBF Kernel | POWER NLL ↓ | Time (s) | GAS NLL ↓ | Time (s) | HEPMASS NLL ↓ | Time (s) | MINIBOONE NLL ↓ | Time (s) | BSDS300 NLL ↓ | Time (s) |
|---|---|---|---|---|---|---|---|---|---|---|---|---|
| DRE-$\infty$ | 2 | - | $0.05_{\pm1.84}$ | 0.317 | $-4.37_{\pm1.44}$ | 0.207 | $19.30_{\pm1.31}$ | 0.311 | $41.55_{\pm2.07}$ | 0.099 | $-130.68_{\pm4.17}$ | 0.448 |
| D$^3$RE | 2 | - | $3.57_{\pm1.84}$ | 0.334 | $5.74_{\pm15.28}$ | 0.218 | $23.90_{\pm0.36}$ | 0.319 | $55.83_{\pm9.36}$ | 0.100 | $-149.53_{\pm9.06}$ | 0.454 |
| DRE-$\infty$ | 5 | - | $0.35_{\pm0.50}$ | 0.561 | $-3.63_{\pm0.78}$ | 0.310 | $20.24_{\pm0.47}$ | 0.612 | $20.90_{\pm0.84}$ | 0.113 | $-83.70_{\pm1.35}$ | 1.061 |
| D$^3$RE | 5 | - | $1.26_{\pm0.38}$ | 0.579 | $-1.15_{\pm4.20}$ | 0.316 | $21.05_{\pm0.52}$ | 0.554 | $43.11_{\pm26.20}$ | 0.117 | $-101.97_{\pm1.67}$ | 1.057 |
| DRE-$\infty$ | 10 | - | $0.03_{\pm0.17}$ | 0.982 | $-4.34_{\pm0.60}$ | 0.517 | $20.43_{\pm0.52}$ | 0.987 | $20.57_{\pm0.93}$ | 0.146 | $-87.65_{\pm2.24}$ | 2.043 |
| D$^3$RE | 10 | - | $0.49_{\pm0.39}$ | 1.051 | $-3.27_{\pm2.00}$ | 0.544 | $20.30_{\pm0.55}$ | 0.987 | $42.65_{\pm26.87}$ | 0.123 | $-102.01_{\pm2.43}$ | 2.042 |
| DRE-$\infty$ | 50 | - | $0.25_{\pm0.28}$ | 4.128 | $-4.33_{\pm0.71}$ | 2.018 | $20.67_{\pm0.57}$ | 4.078 | $20.97_{\pm0.51}$ | 0.223 | $-90.24_{\pm2.14}$ | 10.035 |
| D$^3$RE | 50 | - | $0.89_{\pm0.33}$ | 4.232 | $-3.16_{\pm0.62}$ | 2.072 | $20.05_{\pm0.35}$ | 4.002 | $42.73_{\pm26.78}$ | 0.216 | $-78.26_{\pm0.96}$ | 10.062 |
| OS-DRE (ours) | 1 | Matérn | $0.57_{\pm0.11}$ | 0.084 | $-3.49_{\pm0.01}$ | 0.025 | $23.66_{\pm0.02}$ | 0.064 | $31.71_{\pm0.11}$ | 0.003 | $-52.38_{\pm0.42}$ | 0.073 |
| OS-DRE (ours) | 1 | Gaussian | $-0.35_{\pm0.10}$ | 0.104 | $-16.39_{\pm0.17}$ | 0.038 | $17.44_{\pm0.00}$ | 0.118 | $10.95_{\pm0.33}$ | 0.005 | $-191.22_{\pm3.19}$ | 0.076 |
| OS-DRE (ours) | 1 | IMQ | $\mathbf{-0.69}_{\pm0.18}$ | 0.084 | $\mathbf{-18.33}_{\pm0.04}$ | 0.039 | $17.45_{\pm0.05}$ | 0.071 | $\mathbf{9.97}_{\pm0.37}$ | 0.005 | $\mathbf{-217.99}_{\pm3.39}$ | 0.070 |
| OS-DRE (ours) | 1 | RQ | $-0.66_{\pm0.17}$ | 0.082 | $-17.86_{\pm0.03}$ | 0.037 | $\mathbf{16.88}_{\pm0.03}$ | 0.051 | $11.34_{\pm0.28}$ | 0.003 | $-201.37_{\pm2.21}$ | 0.071 |

ENERGY-BASED MODELING ON MNIST. We conduct density estimation on the MNIST dataset, leveraging pre-trained energy-based models (EBMs) following (Choi et al., 2022; Chen et al., 2025). The log-likelihood of the data distribution $p_1(\boldsymbol{x})$ is estimated via the density ratio and reported in bits-per-dimension (BPD) (see Sec. C.3 for a detail). We use IMQ kernel. The results are summarized in Tab. 2. OS-DRE achieves a BPD of 1.278, setting a new

Table 2: Energy-based modeling on MNIST. Results are reported in bits-per-dim (BPD). Timing measured over the full test set on a single TITAN X GPU. "BS" = batch size.

| Method | Params | BS | NFE | Time (s) | BPD ↓ |
|---|---|---|---|---|---|
| DRE-∞ | 11.2M | 512 | 75 | 21.443 | 1.302 |
| D³RE | 11.2M | 512 | 75 | 21.424 | 1.281 |
| OS-DRE | 11.5M | 512 | 1 | **0.312** | **1.278** |

SOTA for DRE-based methods on this benchmark, surpassing both D³RE (1.281) and DRE-∞ (1.302). Crucially, while achieving better accuracy, OS-DRE maintains its computational advantage. It estimates the BPD with NFE = 1, achieving a test-set inference time of 0.312 seconds, representing a $\sim 68\times$ speedup over D³RE and DRE-∞ (NFE = 75, $\sim 21s$).

**Mutual Information Estimation.** Mutual information (MI) quantifies the dependency between random variables $\mathbf{x} \sim p(\boldsymbol{x})$ and $\mathbf{y} \sim q(\boldsymbol{y})$, quantifying how much information one reveals about the other. We estimate MI between two $d$-dimensional variables using OS-DRE. Formally, $\mathrm{MI}(\mathbf{x}, \mathbf{y}) = \mathbb{E}_{p(\boldsymbol{x}, \boldsymbol{y})}\left[\log \frac{p(\boldsymbol{x}, \boldsymbol{y})}{p(\boldsymbol{x})q(\boldsymbol{y})}\right]$, where the density ratio $\frac{p(\boldsymbol{x}, \boldsymbol{y})}{p(\boldsymbol{x})q(\boldsymbol{y})}$ is directly approximable via DRE.

BEYOND NORMAL: GEOMETRICALLY PATHOLOGICAL DISTRIBUTIONS. We further probe the robustness of OS-DRE on a suite of four MI estimation tasks involving geometrically pathological distributions. These benchmarks, inspired by the suite from Czyż et al. (2023), are specifically designed to challenge the underlying assumptions of many standard estimators by featuring properties like heavy tails, sharp density peaks, and non-differentiable boundaries. For each task, we compute the MI estimate over 10 random seeds and report the mean squared error (MSE) against the known ground-truth MI value. The results, presented in Tab. 3 (full in Tab. 6), demonstrate the stability and accuracy of our method. OS-DRE consistently achieves a lower MSE than the baseline methods across the wide range of challenging data geometries, particularly in scenarios with heavy tails (Half-Cube Map) and complex dependencies (Gamma-Exponential). This highlights the robustness of our analytic, one-step framework in scenarios where traditional score-based methods can struggle.

Table 3: MSE results on the Additive Noise (sharp discontinuities, top) and Gamma–Exponential (non-linear dependency, bottom) datasets. Across all correlation levels (top row of each sub-table), OS-DRE achieves consistently superior or competitive performance. Full results given in Tab. 6.

| Method | RBF Kernel | 0.1 | 0.2 | 0.3 | 0.4 | 0.5 | 0.6 | 0.7 | 0.8 | 0.9 |
|---|---|---|---|---|---|---|---|---|---|---|
| DRE-infty | - | 0.0029 | 0.0018 | 0.0015 | 0.0012 | 0.0013 | 0.0013 | 0.0011 | 0.0011 | 0.000 |
| D3RE | - | 0.0108 | 0.0077 | 0.0065 | 0.0071 | 0.0085 | 0.0076 | 0.0064 | 0.0045 | 0.0055 |
| OS-DRE (ours) | Matérn | 0.0061 | 0.0029 | 0.0017 | 0.0015 | 0.0015 | 0.0013 | 0.0011 | 0.0009 | 0.0008 |
| OS-DRE (ours) | Gaussian | 0.0016 | 0.0015 | 0.0016 | **0.0011** | 0.0015 | 0.0014 | 0.0012 | 0.0010 | 0.0010 |
| OS-DRE (ours) | IMQ | **0.0010** | **0.0010** | **0.0010** | 0.0012 | **0.0009** | **0.0008** | **0.0007** | **0.0009** | **0.0007** |
| OS-DRE (ours) | RQ | 0.0019 | 0.0015 | 0.0015 | 0.0012 | 0.0010 | 0.0010 | 0.0010 | 0.0010 | 0.0009 |

| Method | RBF Kernel | 1.0 | 1.1 | 1.2 | 1.3 | 1.4 | 1.5 | 1.6 | 1.7 | 1.8 |
|---|---|---|---|---|---|---|---|---|---|---|
| DRE-infty | - | 2.1328 | 0.8939 | 0.0725 | 0.0115 | 0.0213 | 0.0051 | 0.0114 | 0.0069 | 0.0051 |
| D3RE | - | 0.1919 | 0.1018 | 0.0154 | 0.0119 | 0.0063 | 0.0110 | 0.0050 | 0.0125 | 0.0114 |
| OS-DRE (ours) | Matérn | **0.1768** | **0.0315** | **0.0035** | **0.0026** | **0.0008** | 0.0017 | **0.0009** | **0.0006** | **0.0005** |
| OS-DRE (ours) | Gaussian | 0.2933 | 0.0503 | 0.0060 | 0.0028 | 0.0032 | **0.0014** | 0.0014 | 0.0007 | 0.0009 |
| OS-DRE (ours) | IMQ | 0.2821 | 0.1185 | 0.0901 | 0.0492 | 0.0200 | 0.0275 | 0.0072 | 0.0080 | 0.0087 |
| OS-DRE (ours) | RQ | 0.5182 | 0.0925 | 0.0330 | 0.0109 | 0.0052 | 0.0040 | 0.0015 | 0.0015 | 0.0012 |

HIGH-DISCREPANCY & HIGH-DIMENSIONAL DISTRIBUTIONS. To evaluate OS-DRE under extreme conditions, we test mutual information estimation between two high-dimensional Gaussians with large and increasing discrepancy, a setup that triggers the "density-chasm" problem (Rhodes et al., 2020). Results in Tab. 4 (full in Tab. 6) show that, unlike DRE-∞ and D³RE, which fail at low NFE and remain unstable even with NFE = 50, OS-DRE with Gaussian or IMQ kernels achieves accurate and stable estimates across all dimensions. This demonstrates that our analytic framework effectively overcomes the density-chasm challenge where iterative methods falter.

**Continual Learning.** To evaluate OS-DRE in online scenarios such as real-time change point detection (Chen et al., 2021) and continuous covariate shift adaptation (Zhang et al., 2023), we test its ability to track evolving distributions across three challenging benchmarks, termed as Linearly Drifting Gaussian, Progressive Noise Corruption and Controlled Divergence Shift (see Sec. C.4 for a detail). In this continual learning setup, the target distribution $p_t$ shifts over discrete timesteps,

Table 4: MI estimation under high-discrepancy settings (MI $\in \{10, 20, 30, 40\}$ nats). We report the estimated MI (mean $\pm$ std over 3 seeds), MSE and wall-clock time. All timing results were obtained on a single NVIDIA TITAN X GPU. **Bolded** MSE values indicate the best performance for each setting. The best wall-clock time is underlined. Full results for NFE $\in \{2, 5, 10, 50\}$ given in Tab. 7.

| Method | NFE | RBF Kernel | MI = 10 | | | MI = 20 | | | MI = 30 | | | MI = 40 | | |
|---|---|---|---|---|---|---|---|---|---|---|---|---|---|---|
| | | | Est. MI | MSE | Time (s) | Est. MI | MSE | Time (s) | Est. MI | MSE | Time (s) | Est. MI | MSE | Time (s) |
| DRE-$\infty$ | 2 | - | $1.40_{\pm 0.01}$ | 73.91 | 0.045 | $3.16_{\pm 0.01}$ | 283.52 | 0.045 | $5.21_{\pm 0.01}$ | 614.62 | 0.045 | $5.13_{\pm 0.02}$ | 1215.69 | 0.046 |
| D$^3$RE | 2 | - | $11.61_{\pm 0.08}$ | 2.58 | 0.048 | $21.91_{\pm 0.08}$ | 3.65 | 0.047 | $27.51_{\pm 0.07}$ | 6.21 | 0.046 | $17.64_{\pm 0.17}$ | 500.04 | 0.044 |
| DRE-$\infty$ | 50 | - | $9.84_{\pm 0.06}$ | 0.03 | 0.226 | $19.81_{\pm 0.04}$ | 0.04 | 0.249 | $29.31_{\pm 0.06}$ | 0.48 | 0.228 | $38.06_{\pm 0.07}$ | 3.77 | 0.271 |
| D$^3$RE | 50 | - | $10.07_{\pm 0.04}$ | **0.01** | 0.234 | $20.30_{\pm 0.03}$ | 0.09 | 0.256 | $27.01_{\pm 0.03}$ | 8.94 | 0.256 | $32.37_{\pm 0.04}$ | 58.19 | 0.260 |
| OS-DRE (ours) | 1 | Matérn | $10.31_{\pm 0.02}$ | 0.09 | 0.024 | $15.73_{\pm 0.05}$ | 18.30 | 0.028 | $15.55_{\pm 0.02}$ | 208.98 | 0.032 | $18.65_{\pm 0.15}$ | 456.11 | 0.028 |
| OS-DRE (ours) | 1 | Gaussian | $10.05_{\pm 0.04}$ | **0.01** | 0.025 | $20.03_{\pm 0.04}$ | **0.00** | 0.027 | $29.37_{\pm 0.07}$ | **0.07** | 0.013 | $38.68_{\pm 0.09}$ | 2.30 | 0.014 |
| OS-DRE (ours) | 1 | IMQ | $10.37_{\pm 0.02}$ | 0.11 | 0.035 | $21.25_{\pm 0.04}$ | 1.56 | 0.030 | $28.10_{\pm 0.08}$ | 5.86 | 0.029 | $39.35_{\pm 0.09}$ | **0.47** | 0.028 |
| OS-DRE (ours) | 1 | RQ | $9.89_{\pm 0.03}$ | 0.03 | 0.022 | $19.49_{\pm 0.04}$ | 0.83 | 0.012 | $28.94_{\pm 0.10}$ | 1.52 | 0.012 | $38.92_{\pm 0.07}$ | 1.41 | 0.019 |

creating a challenging environment that requires the model to continuously adapt to and quantify the change from a fixed source distribution $p_0$. We measure this ability by estimating the KL-divergence between $p_0$ and the evolving target $p_t$ at each step via DRE, i.e., $D_{\mathrm{KL}}(p_t \| p_0) = \mathbb{E}_{p_t(\boldsymbol{x})} \left[ \log \frac{p_t(\boldsymbol{x})}{p_0(\boldsymbol{x})} \right]$.

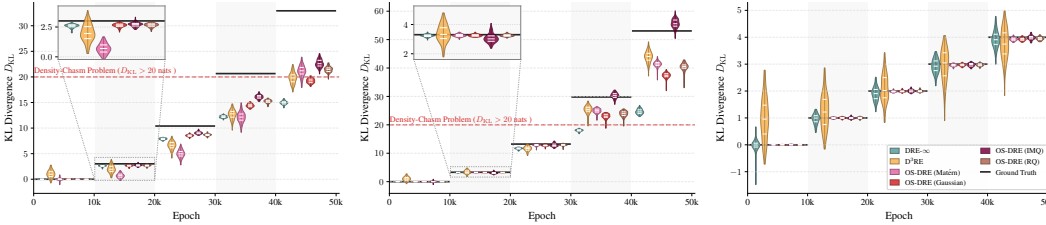

(a) Linearly Drifting Gaussian.     (b) Progressive Noise Corruption.     (c) Controlled Divergence Shift.

Figure 3: Kullback-Leibler (KL) divergence tracking on time-varying distributions. OS-DRE (NFE = 1) provides stable, low-variance estimates that track the ground truth (black line), while baselines (NFE = 50) exhibit significant lag and high variance.

As shown in Fig. 3, OS-DRE delivers real-time, low-variance KL estimates with only a single function evaluation (NFE = 1), closely matching the ground truth across dynamic shifts. In contrast, baseline methods, even with NFE = 50, suffer from lag, variance, and instability, especially under rapid or large shifts. Together, these results demonstrate that our analytic, solver-free formulation avoids the error accumulation and instability inherent in iterative solvers, enabling real-time, reliable tracking of distributional dynamics in continual learning.

**Ablation Studies.** Proposition 4.2 highlights two key hyperparameters: the number of basis functions $K$ and the choice of kernel $\phi$, both governing the trade-off between accuracy and complexity.

NUMBER OF BASIS FUNCTIONS ($K$). We varied $K \in \{100, 200, 400, 800\}$ to study its effect. On GAS, performance improved with larger $K$ up to 400 (NLLs: $-14.51, -15.82, -16.39$) but degraded at 800 (-11.12) due to overfitting, confirming that excessively large $K$ harms generalization. For MI estimation, results were stable across $K$ (e.g., MSEs at MI = 40 with IMQ: $0.55, 0.48, 0.47, 0.49$), indicating diminishing returns once $K$ is sufficient. We thus use $K = 400$ for tabular data and $K = 200$ for pathological distributions as a balanced choice.

CHOICE OF RBF KERNEL ($\phi$). The kernel $\phi$ determines inductive bias and approximation power. Among four tested kernels (Gaussian, Inverse Multiquadric (IMQ), Rational Quadratic (RQ), and Matérn), **IMQ** and **RQ** were strong general-purpose options, achieving state-of-the-art density estimation (Tab. 1 and Fig. 6) and robust MI estimation (Tab. 4), while also stabilizing continual learning tasks (Fig. 3). The **Gaussian** kernel, with localized influence, excelled at capturing sharp or disconnected structures in 2D synthetic benchmarks (Fig. 7) and moderate-discrepancy MI tasks. The **Matérn** kernel, with limited smoothness, was best on geometrically pathological tasks such as Gamma-Exponential, where less smooth inductive bias aligned with the target function.

**Error of the Density Ratio (NLL / MSE) vs. Computational Cost (NFE).** In score-based DRE, the overall error is often dominated by the bias introduced by numerical integration at low NFE, rather than by the score estimation itself. OS-DRE eliminates this bottleneck by replacing numerical integration with a closed-form, solver-free estimation. We measure computational cost using the NFE and evaluate the error of the density ratio using NLL for density estimation and MSE for MI

estimation. As shown in Fig. 4, OS-DRE consistently matches or surpasses the estimation quality of DRE-$\infty$ and D$^3$RE while using only NFE $= 1$. In the density estimation task (Tab. 1 and Fig. 4a), it achieves comparable or better NLL across all five tabular datasets, whereas the baseline methods require NFE values between $2$ and $50$ to reach similar performance. This corresponds to a $50\times$ reduction in computation. A similar pattern is observed in MI estimation (Tab. 4 and Fig. 4b), where OS-DRE attains near-zero MSE at NFE $= 1$ for MI $\in \{10, 20, 30\}$, while the other methods rely on substantially larger NFE. These results show that OS-DRE maintains high estimation quality without costly numerical integration and is therefore well suited for real-time applications.

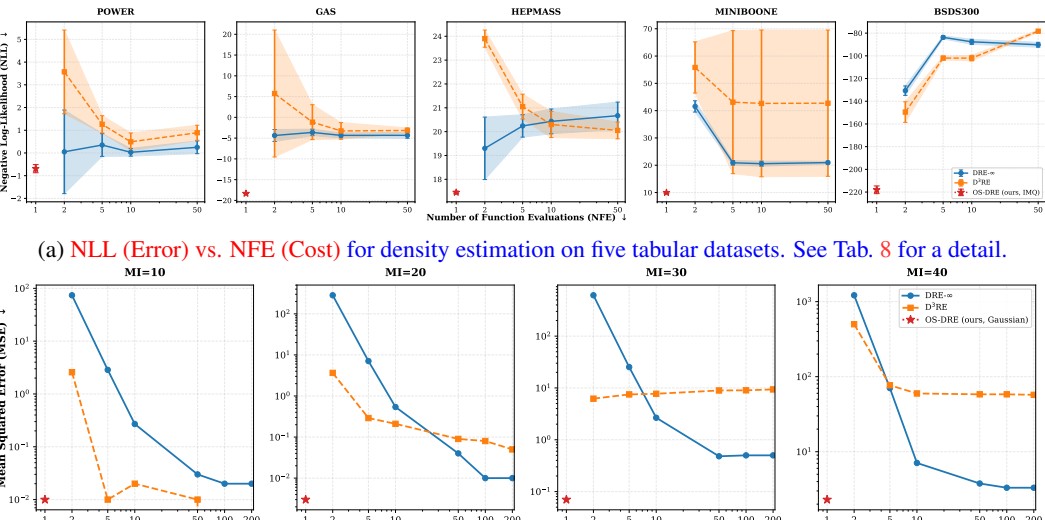

(a) NLL (Error) vs. NFE (Cost) for density estimation on five tabular datasets. See Tab. 8 for a detail.

(b) MSE (Error) vs. NFE (Cost) for MI estimation under high-discrepancy settings. See Tab. 9 for a detail.

Figure 4: Trade-off between error of the density ratio and computational cost. The error is measured by negative log-likelihood (NLL) and mean squared error (MSE), and the computational cost by the number of function evaluations (NFE). OS-DRE maintains high estimation quality with minimal computational cost (NFE $= 1$), whereas the baseline methods require substantially larger NFE to reach comparable performance. **This shows that OS-DRE effectively resolves the longstanding trade-off between estimation quality and integration cost in score-based DRE methods.**

## 6 CONCLUSION

We proposed OS-DRE, a one-step, solver-free framework for score-based density ratio estimation that resolves the long-standing trade-off between accuracy and computational efficiency. By introducing a spatiotemporal decomposition of the time score, our method replaces expensive numerical integration with a single, analytic computation. This is achieved by representing the temporal component of the time score using what we term an analytic frame, a stable approximation basis constructed from radial basis functions, for which the necessary temporal integrals are known in closed form. Our theoretical analysis provides a complete framework for this approach, with proofs for the completeness and stability of the basis, alongside a rigorous truncation error bound that guarantees convergence. Our empirical results demonstrate that this analytic approach achieves competitive accuracy with only a single function evaluation, drastically outperforming iterative, solver-based methods in terms of speed. These findings establish OS-DRE as a powerful and practical tool, opening up new directions for efficient probabilistic inference and statistical estimation.

**Limitations and Future Works.** While OS-DRE achieves efficient and accurate DRE, its effectiveness partly depends on the choice of the target time score function $\{\partial_t \log p_t\}_{t \in [0,1]}$, analyzed in Proposition 4.2. When this target is misaligned with the ideal score (optimal yet unknown), approximation quality may deteriorate. While our work provides a comprehensive analysis of the approximation error, future research could explore training objectives beyond standard time score matching to improve robustness and calibration. For instance, integrating conditional score matching (Yu et al., 2025) or dual score matching (Guth et al., 2025) into our framework could merge analytic integration with energy consistency, potentially yielding more reliable density ratio estimates.

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

## REPRODUCIBILITY STATEMENT

To ensure the reproducibility of our results, we provide a comprehensive account of our work. All experimental setups, including dataset descriptions and key hyperparameter settings, are detailed in the main paper and this appendix. We provide clear pseudocode for our core algorithms, detailing the training procedure (Algorithm 3) and the final one-step estimation process (Algorithm 2). A complete implementation of our OS-DRE framework, along with scripts to replicate all experiments, will be made publicly available upon publication. All experiments were conducted using the PyTorch framework on a single NVIDIA RTX 3070 GPU and four TITAN X GPUs.

## LLM DISCLAIMER

The authors acknowledge the use of a large language model (LLM), specifically Google's Gemini, to assist in the writing and editing process of this paper. The uses of the LLM were primarily for two purposes: (1) to assist in polishing the writing, including improving grammar, clarity, and phrasing of sentences; and (2) for retrieval and discovery, such as finding related work and summarizing existing literature to help situate our contributions within the broader academic context. The core ideas, theoretical derivations, experimental design, and interpretation of results were conducted by the authors.

## BROADER IMPACT

This paper presents work whose goal is to advance the field of density ratio estimation, which does not involve any potential ethical risks. While direct societal impacts are limited, future extensions to applied domains (e.g., via our open-source codebase) should incorporate domain-specific ethical reviews per deployment contexts.

APPENDIX

APPENDIX CONTENTS

## A   ASSUMPTIONS AND PROOFS

### A.1   ASSUMPTIONS

The theoretical results in Sec. 3 rely on the following mild assumptions regarding the analytical properties of the probability density function $p_t(\boldsymbol{x})$.

**Assumption A.1.** There exists a constant $C > 0$ such that $p_t(\boldsymbol{x}) \geq C$ for all $\boldsymbol{x} \in \mathcal{X}$ and $t \in [0, 1]$.

**Assumption A.2** (Uniformly boundedness). The partial derivative $\partial_t p_t(\boldsymbol{x})$ is uniformly bounded; that is, there exists a constant $D > 0$ such that $|\partial_t p_t(\boldsymbol{x})| \leq D$ for all $\boldsymbol{x} \in \mathcal{X}$ and $t \in [0, 1]$.

Assumption A.1 ensures that $\log p_t(\boldsymbol{x})$ is well-defined, while Assumption A.2 ensures that the time score $s_t$ is well-behaved, specifically that it is an element of $L^2(\mathcal{X} \times [0, 1])$, as stated in Lemma 3.1.

### A.2   THEORETICAL FOUNDATION FOR SPATIOTEMPORAL DECOMPOSITION

This section establishes the mathematical foundation for the frame-based decomposition of the spatiotemporal Hilbert space $\mathcal{H}_{\boldsymbol{x},t}$, which underlies Theorem 3.4. The key observation is that $\mathcal{H}_{\boldsymbol{x},t}$ can be realized as the tensor product of the spatial and temporal Hilbert spaces (i.e., $\mathcal{H}_{\boldsymbol{x}}$ and $\mathcal{H}_t$), and that frames of the product space can be constructed from those of the constituent spaces.

We begin with the *algebraic tensor product* $\mathcal{H}_{\boldsymbol{x}} \otimes \mathcal{H}_t$, consisting of finite sums of elementary tensors $f \otimes g$ with $f \in \mathcal{H}_{\boldsymbol{x}}$ and $g \in \mathcal{H}_t$. Although dense in $\mathcal{H}_{\boldsymbol{x},t}$, this space is not complete. Its completion under the induced inner product is the *Hilbert tensor product*, $\mathcal{H}_{\boldsymbol{x}} \hat{\otimes} \mathcal{H}_t$, which is isometrically isomorphic to the space of square-integrable functions on the product domain, $\mathcal{H}_{\boldsymbol{x},t} = L^2(\mathcal{X} \times [0, 1])$ (Kadison & Ringrose, 1986). This isomorphism allows us to construct a frame for the spatiotemporal space from frames of the individual spaces, a result we formalize in the following lemma.

**Lemma A.3.** *Let $\{f_l\}_{l=1}^{\infty}$ be a frame for the spatial space $\mathcal{H}_{\boldsymbol{x}}$ with frame bounds $A_f, B_f$, and let $\{g_k\}_{k=1}^{\infty}$ be a frame for the temporal space $\mathcal{H}_t$ with frame bounds $A_g, B_g$. Then, the set of elementary tensors $\{f_l \otimes g_k\}_{l,k=1}^{\infty}$ forms a frame for the Hilbert tensor product space $\mathcal{H}_{\boldsymbol{x}} \hat{\otimes} \mathcal{H}_t$, with frame bounds $A_f A_g$ and $B_f B_g$.*

*Proof.* Let $F$ be an arbitrary element in $\mathcal{H}_{\boldsymbol{x}} \hat{\otimes} \mathcal{H}_t \cong \mathcal{H}_{\boldsymbol{x},t}$. Let $h_k(\boldsymbol{x}) \triangleq \langle F(\boldsymbol{x}, \cdot), g_k \rangle_{\mathcal{H}_t}$. The sum of the squared frame coefficients for $F$ can be bounded as follows:

$$\sum_{l=1}^{\infty} \sum_{k=1}^{\infty} |\langle F, f_l \otimes g_k \rangle|^2 = \sum_{k=1}^{\infty} \left( \sum_{l=1}^{\infty} |\langle h_k, f_l \rangle_{\mathcal{H}_{\boldsymbol{x}}}|^2 \right) \quad (\star)$$

$$\leq \sum_{k=1}^{\infty} B_f \|h_k\|_{\mathcal{H}_{\boldsymbol{x}}}^2 = B_f \sum_{k=1}^{\infty} \|h_k\|_{\mathcal{H}_{\boldsymbol{x}}}^2 \quad (\star\star)$$

$$= B_f \int_{\mathcal{X}} \left( \sum_{k=1}^{\infty} |\langle F(\boldsymbol{x}, \cdot), g_k \rangle_{\mathcal{H}_t}|^2 \right) \mathrm{d}\boldsymbol{x} \quad (\star\star\star) \tag{17}$$

$$\leq B_f \int_{\mathcal{X}} B_g \|F(\boldsymbol{x}, \cdot)\|_{\mathcal{H}_t}^2 \mathrm{d}\boldsymbol{x} \quad (\star\star\star\star)$$

$$= B_f B_g \|F\|_{\mathcal{H}_{\boldsymbol{x},t}}^2.$$

An analogous derivation provides the lower bound, $A_f A_g \|F\|_{\mathcal{H}_{\boldsymbol{x},t}}^2 \leq \sum_{l,k} |\langle F, f_l \otimes g_k \rangle|^2$. The key steps are: $(\star)$ Rewriting the sum by substituting the definition of $h_k$. $(\star\star)$ Applying the upper frame bound for the spatial frame $\{f_l\}$ for each fixed $k$. $(\star\star\star)$ Using Fubini's theorem to swap the summation and integration. $(\star\star\star\star)$ Applying the upper frame bound for the temporal frame $\{g_k\}$ for each fixed $\boldsymbol{x}$. This completes the proof. $\qquad \square$

This lemma provides the direct theoretical justification for the spatiotemporal expansion used in Theorem 3.4, allowing us to represent any time score function $s_t \in \mathcal{S} \subseteq \mathcal{H}_{\boldsymbol{x},t}$ as a double summation over the tensor product frame elements.

### A.3 PROOF OF LEMMA 3.1

**Lemma 3.1.** *Under Assumptions A.1 and A.2, the space $\mathcal{S}$ is a subset of $L^2(\mathcal{X} \times [0,1])$.*

*Proof.* We want to show that for $\forall s_t \in \mathcal{S}$. It satisfies:

$$\|s_t\|_{L^2(\mathcal{X} \times [0,1])}^2 = \int_{\mathcal{X}} \int_0^1 |s_t(\boldsymbol{x}, t)|^2 \, \mathrm{d}t \mathrm{d}\boldsymbol{x} < \infty. \tag{18}$$

Based on Assumption A.1 and Assumption A.2, we have: (1) $p_t(\boldsymbol{x}) \geq C > 0$, so $\frac{1}{p_t(\boldsymbol{x})} \leq \frac{1}{C}$; (2) $|\partial_t p_t(\boldsymbol{x})| \leq D$. Then, using the chain rule, we write $s_t(\boldsymbol{x}, t)$ as:

$$|s_t(\boldsymbol{x}, t)|^2 = |\partial_t \log p_t(\boldsymbol{x})|^2 = \left| \frac{\partial_t p_t(\boldsymbol{x})}{p_t(\boldsymbol{x})} \right|^2 \leq \left( \frac{D}{C} \right)^2. \tag{19}$$

We now integrate over $\mathcal{X} \times [0,1]$:

$$
\begin{aligned}
\|s_t\|_{L^2(\mathcal{X} \times [0,1])}^2 &= \int_{\mathcal{X}} \int_0^1 |s_t(\boldsymbol{x}, t)|^2 \, \mathrm{d}t \mathrm{d}\boldsymbol{x} \\
&\leq \int_{\mathcal{X}} \int_0^1 \left( \frac{D}{C} \right)^2 \mathrm{d}t \mathrm{d}\boldsymbol{x} \\
&= \left( \frac{D}{C} \right)^2 \int_{\mathcal{X}} \int_0^1 \mathrm{d}t \mathrm{d}\boldsymbol{x} \\
&= \left( \frac{D}{C} \right)^2 \int_{\mathcal{X}} \mathrm{d}\boldsymbol{x}.
\end{aligned}
\tag{20}
$$

Since $p_t(\boldsymbol{x})$ is a probability density function, we know that $\int_{\mathcal{X}} p_t(\boldsymbol{x}) \mathrm{d}\boldsymbol{x} = 1, \forall t \in [0,1]$, which means that the integral over $\mathcal{X}$ is finite, i.e., $\int_{\mathcal{X}} \mathrm{d}\boldsymbol{x} < \infty$. Therefore, the space $\mathcal{S}$ of functions $s_t$ is a subspace of $L^2(\mathcal{X} \times [0,1])$. This complete the proof.

$\square$

### A.4 PROOF OF LEMMA 3.2

**Lemma 3.2.** *Let $\{g_k\}_{k=1}^\infty$ be a complete orthonormal basis for $\mathcal{H}_t$. The target log-density ratio $\log r(\boldsymbol{x})$ can be estimated by:*

$$\log r(\boldsymbol{x}) = \sum_{k=1}^\infty h_k(\boldsymbol{x}) \int_0^1 g_k(t) \mathrm{d}t. \tag{4}$$

*Proof.* The target log-density ratio is defined as the temporal integral of the time score function:

$$\log r(\boldsymbol{x}) = \int_0^1 s_t(\boldsymbol{x}, t) \mathrm{d}t. \tag{21}$$

For a fixed $\boldsymbol{x}$, we can express this integral as an inner product in the Hilbert space $\mathcal{H}_t = L^2([0,1])$ between the function $s_t(\boldsymbol{x}, \cdot)$ and the constant function $1(t) \equiv 1$.

$$\log r(\boldsymbol{x}) = \langle s_t(\boldsymbol{x}, \cdot), 1 \rangle_{\mathcal{H}_t}. \tag{22}$$

Since $\{g_k\}_{k=1}^\infty$ is a complete orthonormal basis for $\mathcal{H}_t$, the time score has the series expansion $s_t(\boldsymbol{x}, t) = \sum_{k=1}^\infty h_k(\boldsymbol{x}) g_k(t)$, which converges in the $L^2$-norm. Due to the continuity of the inner

product in a Hilbert space, we can interchange the inner product and the infinite summation:

$$
\begin{aligned}
\log r(\boldsymbol{x}) &= \left\langle \sum_{k=1}^{\infty} h_k(\boldsymbol{x}) g_k(t), 1(t) \right\rangle_{\mathcal{H}_t} \\
&= \sum_{k=1}^{\infty} \langle h_k(\boldsymbol{x}) g_k(t), 1(t) \rangle_{\mathcal{H}_t} \\
&= \sum_{k=1}^{\infty} h_k(\boldsymbol{x}) \langle g_k(t), 1(t) \rangle_{\mathcal{H}_t} \\
&= \sum_{k=1}^{\infty} h_k(\boldsymbol{x}) \int_0^1 g_k(t) \mathrm{d}t.
\end{aligned}
\tag{23}
$$

This completes the proof. $\qquad\qquad\square$

### A.5 PROOF OF THEOREM 3.4

**Theorem 3.4.** *Let $\{f_l\}_{l=1}^{\infty}$ and $\{g_k\}_{k=1}^{\infty}$ be frames for $\mathcal{H}_{\boldsymbol{x}}$ and $\mathcal{H}_t$, respectively. Then, any time score function $s_t \in \mathcal{S}$ can be expressed as:*

$$
s_t(\boldsymbol{x}, t) = \sum_{k=1}^{\infty} \sum_{l=1}^{\infty} c_{l,k} f_l(\boldsymbol{x}) g_k(t),
\tag{6}
$$

*where the coefficients $c_{l,k}$ depend on $s_t$. By defining spatial coefficients $h_k(\boldsymbol{x}) \triangleq \sum_{l=1}^{\infty} c_{l,k} f_l(\boldsymbol{x})$ and integral $\bar{g}_k \triangleq \int_0^1 g_k(t) \mathrm{d}t$, the time score and the corresponding log-density ratio can be expressed as:*

$$
\boxed{s_t(\boldsymbol{x}, t) = \sum_{k=1}^{\infty} h_k(\boldsymbol{x}) g_k(t), \quad \log r(\boldsymbol{x}) = \sum_{k=1}^{\infty} h_k(\boldsymbol{x}) \bar{g}_k.}
\tag{7}
$$

*Proof.* The proof proceeds in three steps: establishing the existence of the expansion for $s_t$, deriving the corresponding expansion for $\log r(\boldsymbol{x})$, and simplifying the expressions.

By Lemma A.3, since $\{f_l\}$ and $\{g_k\}$ are frames for $\mathcal{H}_{\boldsymbol{x}}$ and $\mathcal{H}_t$ respectively, the set of elementary tensors $\{f_l \otimes g_k\}_{l,k=1}^{\infty}$ forms a frame for the spatiotemporal space $\mathcal{H}_{\boldsymbol{x},t}$. A fundamental property of a frame is that any element in the Hilbert space can be represented as a series expansion of the frame elements. Therefore, for any $s_t \in \mathcal{S} \subseteq \mathcal{H}_{\boldsymbol{x},t}$, there exist coefficients $\{c_{l,k}\}$ such that:

$$
s_t(\boldsymbol{x}, t) = \sum_{l=1}^{\infty} \sum_{k=1}^{\infty} c_{l,k} (f_l \otimes g_k)(\boldsymbol{x}, t) = \sum_{l=1}^{\infty} \sum_{k=1}^{\infty} c_{l,k} f_l(\boldsymbol{x}) g_k(t),
\tag{24}
$$

where the series converges in the norm of $\mathcal{H}_{\boldsymbol{x},t}$.

The log-density ratio is obtained by integrating the time score. For a fixed $\boldsymbol{x}$, the integration operator $I : h(t) \mapsto \int_0^1 h(t) \mathrm{d}t$ is a continuous linear functional on $\mathcal{H}_t$. The continuity allows us to interchange the functional with the infinite summations:

$$
\begin{aligned}
\log r(\boldsymbol{x}) &= \int_0^1 s_t(\boldsymbol{x}, t) \mathrm{d}t \\
&= \int_0^1 \left( \sum_{l=1}^{\infty} \sum_{k=1}^{\infty} c_{l,k} f_l(\boldsymbol{x}) g_k(t) \right) \mathrm{d}t \\
&= \sum_{l=1}^{\infty} \sum_{k=1}^{\infty} c_{l,k} f_l(\boldsymbol{x}) \int_0^1 g_k(t) \mathrm{d}t \\
&= \sum_{l=1}^{\infty} \sum_{k=1}^{\infty} c_{l,k} f_l(\boldsymbol{x}) \bar{g}_k.
\end{aligned}
\tag{25}
$$

By defining the spatial coefficient functions $h_k(\boldsymbol{x}) \triangleq \sum_{l=1}^{\infty} c_{l,k} f_l(\boldsymbol{x})$, we can group the terms in the double summations. This simplification yields the final expressions for the time score and the log-density ratio as presented in the theorem statement:

$$s_t(\boldsymbol{x}, t) = \sum_{k=1}^{\infty} \left( \sum_{l=1}^{\infty} c_{l,k} f_l(\boldsymbol{x}) \right) g_k(t) = \sum_{k=1}^{\infty} h_k(\boldsymbol{x}) g_k(t), \tag{26}$$

$$\log r(\boldsymbol{x}) = \sum_{k=1}^{\infty} \left( \sum_{l=1}^{\infty} c_{l,k} f_l(\boldsymbol{x}) \right) \bar{g}_k = \sum_{k=1}^{\infty} h_k(\boldsymbol{x}) \bar{g}_k. \tag{27}$$

This completes the proof. □

### A.6 PROOF OF COROLLARY 3.5

**Corollary 3.5.** *If each function $g_k$ in the frame expansion belongs to the Sobolev space $\mathcal{W}^{1,2}([0,1])$ and the coefficients $\{h_k(\boldsymbol{x})\}$ are such that the series $\sum_{k=1}^{\infty} h_k(\boldsymbol{x}) g_k'(t)$ converges in $\mathcal{H}_{\boldsymbol{x},t}$, then the weak derivative of the time score $s_t$ with respect to $t$ exists and is given by term-by-term differentiation:*

$$\partial_t s_t(\boldsymbol{x}, t) = \sum_{k=1}^{\infty} h_k(\boldsymbol{x}) g_k'(t). \tag{8}$$

*Proof.* Let $v(\boldsymbol{x}, t) \triangleq \sum_{k=1}^{\infty} h_k(\boldsymbol{x}) g_k'(t)$. By assumption, this series converges to a function $v \in \mathcal{H}_{\boldsymbol{x},t}$. We must show that $v$ is the weak derivative of $s_t(\boldsymbol{x}, t)$ with respect to time.

By definition, this requires showing that for any smooth test function $\psi \in C_c^{\infty}((0,1))$, the following equality holds for almost every $\boldsymbol{x} \in \mathcal{X}$:

$$\int_0^1 s_t(\boldsymbol{x}, t) \psi'(t) \mathrm{d}t = -\int_0^1 v(\boldsymbol{x}, t) \psi(t) \mathrm{d}t. \tag{28}$$

Let's evaluate the left-hand side. For a fixed $\boldsymbol{x}$, we have:

$$\int_0^1 s_t(\boldsymbol{x}, t) \psi'(t) \mathrm{d}t = \int_0^1 \left( \sum_{k=1}^{\infty} h_k(\boldsymbol{x}) g_k(t) \right) \psi'(t) \mathrm{d}t$$

$$= \sum_{k=1}^{\infty} h_k(\boldsymbol{x}) \int_0^1 g_k(t) \psi'(t) \mathrm{d}t \quad (\star)$$

$$= \sum_{k=1}^{\infty} h_k(\boldsymbol{x}) \left( -\int_0^1 g_k'(t) \psi(t) \mathrm{d}t \right) \quad (\star\star) \tag{29}$$

$$= -\int_0^1 \left( \sum_{k=1}^{\infty} h_k(\boldsymbol{x}) g_k'(t) \right) \psi(t) \mathrm{d}t \quad (\star\star\star)$$

$$= -\int_0^1 v(\boldsymbol{x}, t) \psi(t) \mathrm{d}t.$$

The key steps are justified as follows: $(\star)$ The interchange of summation and integration is permitted because the series for $s_t$ converges in $L^2$, and the operator $h \mapsto \int h\psi' \mathrm{d}t$ is a continuous linear functional on $L^2$. $(\star\star)$ Since each $g_k \in \mathcal{W}^{1,2}([0,1])$, it has a weak derivative $g_k'$. By the definition of the weak derivative and the fact that $\psi$ has compact support in $(0,1)$ (meaning boundary terms vanish), we can apply integration by parts. $(\star\star\star)$ The interchange of integration and summation is again justified by the assumed $L^2$ convergence of the series defining $v(\boldsymbol{x}, t)$.

This confirms that $v(\boldsymbol{x}, t)$ is the weak derivative of $s_t(\boldsymbol{x}, t)$, completing the proof. □

### A.7 DENSENESS OF TEMPORAL BASIS IN $\mathcal{H}_t$

**Lemma A.4.** *Let $\mathcal{H}_t \triangleq L^2([0,1])$. Consider a family of radial basis functions $\{g_k\}_{k=1}^{\infty}$ defined by*

$$g_k(t) = \phi\left( \frac{|t - c_k|}{\sigma_k} \right), \quad t \in [0,1], \tag{30}$$

*subject to the following conditions:*

  (i) *The generating function $\phi : [0, \infty) \to \mathbb{R}_+$ is continuous, non-negative, not identically zero, and integrable, i.e., $\int_0^\infty \phi(r)\mathrm{d}r < \infty$.*

  (ii) *The set of centers $\{c_k\}_{k=1}^\infty$ is dense in $[0, 1]$.*

  (iii) *For any point $t_0 \in (0, 1)$, there exists a subsequence of indices $\{k_n\}_{n=1}^\infty$ such that the centers $c_{k_n} \to t_0$ and the corresponding shape parameters $\sigma_{k_n} \to 0$ as $n \to \infty$.*

*Then, the linear span of this family, $\mathcal{A} \triangleq \mathrm{span}\{g_k\}_{k=1}^\infty$, is dense in $\mathcal{H}_t$.*

*Proof.* We use a proof by contradiction. Assume that the linear span $\mathcal{A}$ is not dense in $\mathcal{H}_t$. A fundamental theorem of Hilbert spaces states that a subspace is dense if and only if its orthogonal complement contains only the zero vector. Therefore, our assumption implies the existence of a **non-zero** function $u \in \mathcal{H}_t$ (i.e., $\|u\|_{\mathcal{H}_t} > 0$) that is orthogonal to every function in the basis family $\{g_k\}_{k=1}^\infty$. This orthogonality condition is expressed as:

$$\langle u, g_k \rangle_{\mathcal{H}_t} = \int_0^1 u(t) g_k(t)\mathrm{d}t = 0, \quad \forall k \geq 1. \tag{31}$$

Our objective is to show that this assumption forces $u$ to be the zero function in $\mathcal{H}_t$, which will establish the contradiction.

By the Lebesgue Differentiation Theorem, for any function $u \in L^1([0, 1])$ (and thus for any $u \in L^2([0, 1])$), almost every point in $(0, 1)$ is a Lebesgue point. Let us choose an arbitrary such Lebesgue point, $t_0 \in (0, 1)$.

Based on our assumptions, we can construct a specific sequence of functions. Since the centers $\{c_k\}$ are dense and condition (iii) holds, we can select a subsequence of indices $\{k_n\}_{n=1}^\infty$ such that $c_{k_n} \to t_0$ and $\sigma_{k_n} \to 0$ as $n \to \infty$. Let us denote the corresponding functions as $g_n(t) \triangleq g_{k_n}(t)$.

We now define a sequence of normalized functions $\{F_n(t)\}_{n=1}^\infty$:

$$F_n(t) = \frac{g_n(t)}{D_n}, \quad \text{where } D_n = \int_{-\infty}^\infty g_n(s)\mathrm{d}s. \tag{32}$$

The normalization constant $D_n$ is computed over $\mathbb{R}$ to capture the total mass of the kernel, which is standard practice for constructing an approximate identity. It can be calculated via a change of variables:

$$D_n = \int_{-\infty}^\infty \phi\left(\frac{|s - c_{k_n}|}{\sigma_{k_n}}\right)\mathrm{d}s = \sigma_{k_n} \int_{-\infty}^\infty \phi(|v|)\mathrm{d}v = 2\sigma_{k_n} \int_0^\infty \phi(r)\mathrm{d}r. \tag{33}$$

Let $C_\phi = 2\int_0^\infty \phi(r)\mathrm{d}r$. By condition (i), $C_\phi$ is a finite positive constant, so $D_n = C_\phi \sigma_{k_n} > 0$. The sequence $\{F_n\}$ forms an "approximate identity" (or a summability kernel) centered around $t_0$, which is characterized by three key properties:

  1. Non-negativity: Since $\phi(r) \geq 0$ and $D_n > 0$, we have $F_n(t) \geq 0$ for all $t$.

  2. Unit Integral: By construction, $\int_{-\infty}^\infty F_n(t)\mathrm{d}t = 1$ for all $n$.

  3. Concentration of Mass: For any fixed $\delta > 0$, the integral of $F_n$ outside the neighborhood $(c_{k_n} - \delta, c_{k_n} + \delta)$ vanishes as $n \to \infty$.

$$\begin{aligned}
\lim_{n\to\infty} \int_{|t-c_{k_n}|\geq\delta} F_n(t)\mathrm{d}t &= \lim_{n\to\infty} \frac{1}{D_n} \int_{|t-c_{k_n}|\geq\delta} \phi\left(\frac{|t - c_{k_n}|}{\sigma_{k_n}}\right)\mathrm{d}t \\
&= \lim_{n\to\infty} \frac{\sigma_{k_n}}{D_n} \int_{|v|\geq\delta/\sigma_{k_n}} \phi(|v|)\mathrm{d}v \quad \text{(letting } v = (t - c_{k_n})/\sigma_{k_n}) \\
&= \frac{1}{C_\phi} \lim_{n\to\infty} 2 \int_{\delta/\sigma_{k_n}}^\infty \phi(r)\mathrm{d}r. \tag{34}
\end{aligned}$$

  Since $\sigma_{k_n} \to 0$, the lower limit of integration $\delta/\sigma_{k_n} \to \infty$. As $\int_0^\infty \phi(r)\mathrm{d}r$ is finite, the tail of the integral must go to zero, i.e., $\lim_{x\to\infty} \int_x^\infty \phi(r)\mathrm{d}r = 0$. Thus, this limit is zero.

Now, let us examine the convolution-like integral $\int_0^1 u(t)F_n(t)\mathrm{d}t$. From our initial orthogonality assumption in Eq. (31), we have $\langle u, g_n \rangle_{\mathcal{H}_t} = 0$ for all $n$. This directly implies:

$$\int_0^1 u(t)F_n(t)\mathrm{d}t = \frac{1}{D_n} \int_0^1 u(t)g_n(t)\mathrm{d}t = \frac{1}{D_n}\langle u, g_n \rangle_{\mathcal{H}_t} = 0, \quad \forall n. \tag{35}$$

On the other hand, because $t_0$ is a Lebesgue point of $u$ and $\{F_n\}$ is an approximate identity sequence concentrating at $t_0$ (since $c_{k_n} \to t_0$), a standard result of analysis (a key part of the Lebesgue Differentiation Theorem's proof) states that:

$$\lim_{n \to \infty} \int_0^1 u(t)F_n(t)\mathrm{d}t = u(t_0). \tag{36}$$

Comparing Eq. (35) and Eq. (36), we must conclude that $u(t_0) = 0$.

Since $t_0$ was an arbitrary Lebesgue point and the set of Lebesgue points has full measure in $[0, 1]$, we have shown that $u(t) = 0$ almost everywhere on $[0, 1]$. In the space $L^2([0, 1])$, a function that is zero almost everywhere is equivalent to the zero vector.

This contradicts our initial assumption that $u$ was a non-zero function. Therefore, the assumption that $\mathcal{A}$ is not dense in $\mathcal{H}_t$ must be false. This completes the proof. □

## A.8   PROOF OF PROPOSITION 4.1

**Proposition 4.1.** *Let $\{g_k\}_{k=1}^{\infty}$ be an infinite family of RBFs in $\mathcal{H}_t$, defined by $g_k(t) = \phi(|t - c_k|/\sigma_k)$. $c_k$ and $\sigma_k$ are the center and shape paramaters of $g_k$. This family generates a convergent and well-posed approximation scheme if it meets two conditions: (i) Denseness: The infinite family's linear span is dense in $\mathcal{H}_t$, i.e., $\overline{\mathrm{span}\{g_k\}_{k=1}^{\infty}} = \mathcal{H}_t$. (ii) Finite-dimensional stability: For any finite $K \geq 1$, the subset $\{g_k\}_{k=1}^{K}$ is linearly independent.*

*Proof.* The proof consists of verifying that these two conditions ensure the desired properties of the approximation scheme.

Convergence: Condition (i), established by our Denseness Lemma (Lemma A.4), guarantees the scheme's convergence. It ensures that for any function $h \in \mathcal{H}_t$ and any error tolerance $\epsilon > 0$, there exists a sufficiently large dimension $K$ and a function $g \in V_K$ such that $\|h - g\|_{\mathcal{H}_t} < \epsilon$. This means the approximation error of the best-fit projection, $\inf_{g \in V_K} \|h - g\|$, can be made arbitrarily small.

Well-posed Approximation: Condition (ii) guarantees that for any fixed, finite $K$, the approximation problem within the subspace $V_K$ is well-posed. Since $\{g_k\}_{k=1}^{K}$ is a linearly independent set, it forms a basis for the subspace $V_K = \mathrm{span}\{g_k\}_{k=1}^{K}$. In a finite-dimensional Hilbert space, any basis is a Riesz basis (a specific type of frame). This implies the existence of frame bounds $A_K$ and $B_K$ that depend on $K$, satisfying $0 < A_K \leq B_K < \infty$. The existence of a strictly positive lower bound $A_K$ ensures that the projection of any function onto $V_K$ is a stable and well-defined operation. □

## A.9   PROOF OF PROPOSITION 4.2

**Proposition 4.2.** *Let the RBF generating function $\phi$ be such that its native space $\mathcal{N}_\phi$ is equivalent to $\mathcal{W}^{\tau,2}(\mathbb{R})$ for some $\tau > 1/2$. Let the target function $s_t(\boldsymbol{x}, \cdot)$ belong to a Sobolev space of lower smoothness, $s_t(\boldsymbol{x}, \cdot) \in \mathcal{W}^{\beta,2}([0, 1])$ with $1/2 < \beta \leq \tau$. Let $s_t^{(K)}(\boldsymbol{x}, \cdot)$ be the best approximation of $s_t$ in the subspace $V_K = \mathrm{span}\{g_k\}_{k=1}^{K}$, where the centers $\mathcal{C}_K = \{c_k\}_{k=1}^{K}$ are quasi-uniform. Then, there exists a constant $C$, independent of $s_t$ and $K$, such that the approximation error is bounded by:*

$$\|s_t(\boldsymbol{x}, \cdot) - s_t^{(K)}(\boldsymbol{x}, \cdot)\|_{\mathcal{H}_t} \leq C \cdot K^{-\beta} \cdot \|s_t(\boldsymbol{x}, \cdot)\|_{\mathcal{W}^{\beta,2}([0,1])}. \tag{11}$$

*Proof.* The proof is a direct application of the main results presented in Narcowich et al. (2006). Let $I_{\mathcal{C}_K} s_t$ denote the RBF interpolant to $s_t$ at the centers $\mathcal{C}_K$. The best approximation error in the subspace $V_K$ is, by definition, the infimum of the error over all functions in that subspace, which is bounded above by the error of any specific function in $V_K$, such as the RBF interpolant $I_{\mathcal{C}_K} s_t$. Thus,

$$\|s_t - s_t^{(K)}\|_{\mathcal{H}_t} = \inf_{h \in V_K} \|s_t - h\|_{\mathcal{H}_t} \leq \|s_t - I_{\mathcal{C}_K} s_t\|_{\mathcal{H}_t}. \tag{37}$$

We now bound the interpolation error using the results from Narcowich et al. (2006). The cited work provides error estimates for functions defined on a general compact domain $\Omega \subset \mathbb{R}^d$. Crucially, the validity of these results hinges on the assumption that the RBF generating function $\phi$ has a native space $\mathcal{N}_\phi$ equivalent to $W^{\tau,2}(\mathbb{R})$. Their key result, Theorem 4.2, provides an estimate for functions $f \in \mathcal{W}^{\beta,2}(\Omega)$ that are less smooth than the native space order $\tau$. Applying this theorem to our specific one-dimensional case where $\Omega = [0,1]$ and setting the error norm order $\mu = 0$ (for the $L^2$ norm) gives:

$$\|s_t - I_{\mathcal{C}_K} s_t\|_{L^2([0,1])} \le A h^\beta_{\mathcal{C}_K,[0,1]} \rho^\tau_{\mathcal{C}_K,[0,1]} \|s_t\|_{\mathcal{W}^{\beta,2}([0,1])}, \tag{38}$$

where $h_{\mathcal{C}_K,[0,1]}$ is the fill distance and $\rho_{\mathcal{C}_K,[0,1]}$ is the mesh ratio.

The proposition assumes that the centers $\mathcal{C}_K$ are quasi-uniform. For such a set of points, the mesh ratio is bounded by a constant independent of $K$, i.e., $\rho_{\mathcal{C}_K,[0,1]} \le \rho_{max}$. Furthermore, the fill distance is directly related to the number of points, $h_{\mathcal{C}_K,[0,1]} = \mathcal{O}(1/K)$.

Substituting these into the bound in Eq. (38):

$$\|s_t - I_{\mathcal{C}_K} s_t\|_{L^2([0,1])} \le A(\mathcal{O}(1/K))^\beta (\rho_{max})^\tau \|s_t\|_{\mathcal{W}^{\beta,2}([0,1])}. \tag{39}$$

By defining a new constant $C \triangleq A \cdot (\rho_{max})^\tau$ that absorbs all terms independent of $s_t$ and $K$, we arrive at the final error bound:

$$\begin{aligned}
\|s_t(\boldsymbol{x},\cdot) - s_t^{(K)}(\boldsymbol{x},\cdot)\|_{\mathcal{H}_t} &\le \|s_t(\boldsymbol{x},\cdot) - I_{\mathcal{C}_K} s_t(\boldsymbol{x},\cdot)\|_{\mathcal{H}_t} \\
&\le A(\mathcal{O}(1/K))^\beta (\rho_{max})^\tau \|s_t\|_{\mathcal{W}^{\beta,2}([0,1])} \quad (\mathcal{H}_t = L^2([0,1])) \\
&= C \cdot K^{-\beta} \cdot \|s_t(\boldsymbol{x},\cdot)\|_{\mathcal{W}^{\beta,2}([0,1])}.
\end{aligned} \tag{40}$$

This completes the proof. $\qquad\square$

# B ANALYTIC FORMULAS FOR RBF KERNELS

This section provides a summary of the Radial Basis Function (RBF) generating functions, $\phi(r)$, used and referenced in this work. All kernels listed below are *strictly positive definite*, satisfying the conditions of our approximation framework. Their respective closed-form integrals and derivatives are detailed in the subsequent sections.

Table 5: A summary of different RBF generating functions used in this paper.

| Kernel Name | $\phi(r)$ | Key Properties | Analytic Formulas |
|---|---|---|---|
| Gaussian | $\exp(-r^2)$ | Infinitely smooth, localized influence (fast decay). | Eqs. (41) and (42) |
| Inverse Multiquadric | $(r^2+1)^{-1/2}$ | Infinitely smooth, global influence (slow decay). | Eqs. (43) and (44) |
| Rational Quadratic | $(r^2+1)^{-1}$ | Infinitely smooth, multi-scale, medium decay. | Eqs. (45) and (46) |
| Matérn ($\nu = 3/2$) | $(1+\sqrt{3}r)\exp(-\sqrt{3}r)$ | Limited smoothness ($C^2$), local influence. | Eqs. (47) and (48) |

## B.1 GAUSSIAN RBFS

The Gaussian RBF is defined by the generating function $\phi(r) = \exp(-r^2)$. The basis functions are therefore given by:

$$g_k(t) = \exp\left(-\frac{|t - c_k|^2}{\sigma_k^2}\right), \tag{41}$$

where $c_k$ and $\sigma_k > 0$ are the center and shape parameters of $g_k$.

**Closed-Form Expression for the Temporal Integral.** The integral $\bar{g}_k = \int_0^1 g_k(t)\mathrm{d}t$ is calculated as follows. We use the substitution $u = (t - c_k)/\sigma_k$, which implies $\mathrm{d}t = \sigma_k \mathrm{d}u$.

$$\bar{g}_k = \int_0^1 \exp\left(-\frac{|t - c_k|^2}{\sigma_k^2}\right) \mathrm{d}t = \sigma_k \int_{-c_k/\sigma_k}^{(1-c_k)/\sigma_k} \exp(-u^2)\mathrm{d}u.$$

This integral can be expressed using the error function, $\mathrm{erf}(z) = \frac{2}{\sqrt{\pi}} \int_0^z \exp(-x^2)\mathrm{d}x$. Since $\int_a^b \exp(-u^2)\mathrm{d}u = \frac{\sqrt{\pi}}{2}(\mathrm{erf}(b) - \mathrm{erf}(a))$, we have:

$$\bar{g}_k = \sigma_k \frac{\sqrt{\pi}}{2} \left[ \mathrm{erf}\left(\frac{1 - c_k}{\sigma_k}\right) - \mathrm{erf}\left(-\frac{c_k}{\sigma_k}\right) \right]$$
$$= \frac{\sigma_k \sqrt{\pi}}{2} \left[ \mathrm{erf}\left(\frac{1 - c_k}{\sigma_k}\right) + \mathrm{erf}\left(\frac{c_k}{\sigma_k}\right) \right],$$

where the last step uses the property $\mathrm{erf}(-z) = -\mathrm{erf}(z)$.

**Closed-Form Expression for the Temporal Derivative.**    The derivative $g_k'(t)$ is found by applying the chain rule:

$$g_k'(t) = \frac{\mathrm{d}}{\mathrm{d}t} \exp\left(-\frac{(t - c_k)^2}{\sigma_k^2}\right)$$
$$= \exp\left(-\frac{(t - c_k)^2}{\sigma_k^2}\right) \cdot \frac{\mathrm{d}}{\mathrm{d}t}\left(-\frac{(t - c_k)^2}{\sigma_k^2}\right)$$
$$= \exp\left(-\frac{(t - c_k)^2}{\sigma_k^2}\right) \cdot \left(-\frac{2(t - c_k)}{\sigma_k^2}\right).$$

The final expressions for the integral and derivative are summarized below.

$$\bar{g}_k = \frac{\sigma_k \sqrt{\pi}}{2} \left[ \mathrm{erf}\left(\frac{1 - c_k}{\sigma_k}\right) + \mathrm{erf}\left(\frac{c_k}{\sigma_k}\right) \right],$$
$$g_k'(t) = -\frac{2(t - c_k)}{\sigma_k^2} g_k(t). \tag{42}$$

### B.2  INVERSE MULTIQUADRIC RBFS

The Inverse Multiquadric (IMQ) RBF is defined by the generating function $\phi(r) = (r^2 + 1)^{-1/2}$. The basis functions are therefore given by:

$$g_k(t) = \left( \frac{(t - c_k)^2}{\sigma_k^2} + 1 \right)^{-\frac{1}{2}} = \frac{\sigma_k}{\sqrt{(t - c_k)^2 + \sigma_k^2}}. \tag{43}$$

**Closed-Form Expression for the Temporal Integral.**    The temporal integral $\bar{g}_k = \int_0^1 g_k(t)\mathrm{d}t$ is calculated using the standard integral for the inverse hyperbolic sine function. We use the substitution $u = t - c_k$, which implies $\mathrm{d}t = \mathrm{d}u$.

$$\bar{g}_k = \int_0^1 \frac{\sigma_k}{\sqrt{(t - c_k)^2 + \sigma_k^2}}\mathrm{d}t = \sigma_k \int_{-c_k}^{1 - c_k} \frac{1}{\sqrt{u^2 + \sigma_k^2}}\mathrm{d}u.$$

The integral of $1/\sqrt{u^2 + a^2}$ is $\ln(u + \sqrt{u^2 + a^2})$. Applying this, we get:

$$\bar{g}_k = \sigma_k \left[ \ln\left( u + \sqrt{u^2 + \sigma_k^2} \right) \right]_{-c_k}^{1 - c_k}$$
$$= \sigma_k \left( \ln\left( (1 - c_k) + \sqrt{(1 - c_k)^2 + \sigma_k^2} \right) - \ln\left( -c_k + \sqrt{c_k^2 + \sigma_k^2} \right) \right)$$
$$= \sigma_k \ln\left( \frac{(1 - c_k) + \sqrt{(1 - c_k)^2 + \sigma_k^2}}{-c_k + \sqrt{c_k^2 + \sigma_k^2}} \right).$$

**Closed-Form Expression for the Temporal Derivative.** The derivative $g_k'(t)$ is found by applying the chain rule to $g_k(t) = \sigma_k \left( (t - c_k)^2 + \sigma_k^2 \right)^{-1/2}$:

$$
\begin{aligned}
g_k'(t) &= \sigma_k \cdot \frac{\mathrm{d}}{\mathrm{d}t} \left( (t - c_k)^2 + \sigma_k^2 \right)^{-1/2} \\
&= \sigma_k \cdot \left( -\frac{1}{2} \right) \left( (t - c_k)^2 + \sigma_k^2 \right)^{-3/2} \cdot \frac{\mathrm{d}}{\mathrm{d}t} \left( (t - c_k)^2 + \sigma_k^2 \right) \\
&= \sigma_k \cdot \left( -\frac{1}{2} \right) \left( (t - c_k)^2 + \sigma_k^2 \right)^{-3/2} \cdot 2(t - c_k) \\
&= -\frac{\sigma_k (t - c_k)}{\left( (t - c_k)^2 + \sigma_k^2 \right)^{3/2}}.
\end{aligned}
$$

The final expressions for the integral and derivative are summarized below.

$$
\begin{aligned}
\bar{g}_k &= \sigma_k \ln \left( \frac{(1 - c_k) + \sqrt{(1 - c_k)^2 + \sigma_k^2}}{-c_k + \sqrt{c_k^2 + \sigma_k^2}} \right), \\
g_k'(t) &= -\frac{\sigma_k (t - c_k)}{\left( (t - c_k)^2 + \sigma_k^2 \right)^{3/2}}.
\end{aligned}
\tag{44}
$$

### B.3 RATIONAL QUADRATIC RBFs

The Rational Quadratic (RQ) kernel can be viewed as an infinite sum of Gaussian kernels of different scales. This property makes it a robust choice, capable of modeling data at multiple scales. It is strictly positive definite, and its generating function is $\phi(r) = (1 + r^2)^{-1}$. The basis functions are therefore given by:

$$
g_k(t) = \left( 1 + \frac{(t - c_k)^2}{\sigma_k^2} \right)^{-1} = \frac{\sigma_k^2}{(t - c_k)^2 + \sigma_k^2}.
\tag{45}
$$

**Closed-Form Expression for the Temporal Integral.** The temporal integral $\bar{g}_k = \int_0^1 g_k(t) \mathrm{d}t$ is calculated using the standard integral for the arctangent function. We use the substitution $u = t - c_k$, which implies $\mathrm{d}t = \mathrm{d}u$.

$$
\bar{g}_k = \int_0^1 \frac{\sigma_k^2}{(t - c_k)^2 + \sigma_k^2} \mathrm{d}t = \sigma_k^2 \int_{-c_k}^{1 - c_k} \frac{1}{u^2 + \sigma_k^2} \mathrm{d}u.
$$

The integral of $1/(u^2 + a^2)$ is $\frac{1}{a} \arctan(\frac{u}{a})$. Applying this, we get:

$$
\begin{aligned}
\bar{g}_k &= \sigma_k^2 \left[ \frac{1}{\sigma_k} \arctan \left( \frac{u}{\sigma_k} \right) \right]_{-c_k}^{1 - c_k} \\
&= \sigma_k \left( \arctan \left( \frac{1 - c_k}{\sigma_k} \right) - \arctan \left( -\frac{c_k}{\sigma_k} \right) \right) \\
&= \sigma_k \left( \arctan \left( \frac{1 - c_k}{\sigma_k} \right) + \arctan \left( \frac{c_k}{\sigma_k} \right) \right),
\end{aligned}
$$

where the last step uses the property $\arctan(-z) = -\arctan(z)$.

**Closed-Form Expression for the Temporal Derivative.** The derivative $g_k'(t)$ is found by applying the chain rule to $g_k(t) = \sigma_k^2 \left( (t - c_k)^2 + \sigma_k^2 \right)^{-1}$:

$$
\begin{aligned}
g_k'(t) &= \sigma_k^2 \cdot \frac{\mathrm{d}}{\mathrm{d}t} \left( (t - c_k)^2 + \sigma_k^2 \right)^{-1} \\
&= \sigma_k^2 \cdot (-1) \left( (t - c_k)^2 + \sigma_k^2 \right)^{-2} \cdot \frac{\mathrm{d}}{\mathrm{d}t} \left( (t - c_k)^2 + \sigma_k^2 \right) \\
&= -\sigma_k^2 \left( (t - c_k)^2 + \sigma_k^2 \right)^{-2} \cdot 2(t - c_k) \\
&= -\frac{2\sigma_k^2 (t - c_k)}{\left( (t - c_k)^2 + \sigma_k^2 \right)^2}.
\end{aligned}
$$

The final expressions for the integral and derivative are summarized below.

$$\bar{g}_k = \sigma_k \left( \arctan\left( \frac{1-c_k}{\sigma_k} \right) + \arctan\left( \frac{c_k}{\sigma_k} \right) \right), \quad g'_k(t) = -\frac{2\sigma_k^2(t-c_k)}{\left((t-c_k)^2 + \sigma_k^2\right)^2}. \tag{46}$$

### B.4 MATÉRN RBFs

The Matérn family of RBFs is widely used in machine learning, particularly in Gaussian processes, as their smoothness is controlled by a parameter $\nu$. We consider the common case where $\nu = 3/2$, which corresponds to a once-differentiable function. The generating function is strictly positive definite and is given by $\phi(r) = (1 + \sqrt{3}r)\exp(-\sqrt{3}r)$. The basis functions, which are in the Sobolev space $W_2^2(\mathbb{R})$, are:

$$g_k(t) = \left( 1 + \frac{\sqrt{3}|t-c_k|}{\sigma_k} \right) \exp\left( -\frac{\sqrt{3}|t-c_k|}{\sigma_k} \right). \tag{47}$$

**Closed-Form Expression for the Temporal Integral.** The integral $\bar{g}_k = \int_0^1 g_k(t)\mathrm{d}t$ is computed by splitting the integral at the center $c_k$ due to the absolute value. The indefinite integral of the generating function is $\int \phi(r)\mathrm{d}r = -re^{-\sqrt{3}r} - \frac{2}{\sqrt{3}}e^{-\sqrt{3}r}$. Evaluating this over the respective intervals yields the final closed form.

**Closed-Form Expression for the Temporal Derivative.** The derivative of the generating function is $\phi'(r) = -3r\exp(-\sqrt{3}r)$. Applying the chain rule, we find the derivative of $g_k(t)$:

$$g'_k(t) = \phi'\left( \frac{|t-c_k|}{\sigma_k} \right) \cdot \frac{\mathrm{sgn}(t-c_k)}{\sigma_k}$$

$$= -3\frac{|t-c_k|}{\sigma_k} \exp\left( -\frac{\sqrt{3}|t-c_k|}{\sigma_k} \right) \cdot \frac{t-c_k}{|t-c_k|} \cdot \frac{1}{\sigma_k}$$

$$= -\frac{3(t-c_k)}{\sigma_k^2} \exp\left( -\frac{\sqrt{3}|t-c_k|}{\sigma_k} \right).$$

The final expressions are summarized below.

$$\bar{g}_k = \frac{2\sigma_k}{\sqrt{3}} - \sigma_k \left[ \left( \frac{1-c_k}{\sigma_k} + \frac{2}{\sqrt{3}} \right) e^{-\frac{\sqrt{3}(1-c_k)}{\sigma_k}} + \left( \frac{c_k}{\sigma_k} + \frac{2}{\sqrt{3}} \right) e^{-\frac{\sqrt{3}c_k}{\sigma_k}} \right],$$

$$g'_k(t) = -\frac{3(t-c_k)}{\sigma_k^2} \exp\left( -\frac{\sqrt{3}|t-c_k|}{\sigma_k} \right). \tag{48}$$

# C  EXPERIMENTAL DETAILS AND MORE RESULTS

All experiments were conducted on four NVIDIA TITAN X (Pascal) 12GB GPUs using PyTorch (2.1.2) and PyTorch-Lightning (2.1.2). Our code is developed based on the official code for both DRE-∞ at https://github.com/ermongroup/dre-infinity and Neural ODE at https://github.com/rtqichen/torchdiffeq. **Our code will be made available once the paper is accepted.**

## C.1  EXPERIMENTAL DETAILS

### C.1.1  INTERPOLATING PATHS AND THE TEMPORAL-INTEGRAL VIEW

This section clarifies how interpolating paths are constructed and why the resulting temporal integral offers a stable formulation of density ratio estimation (DRE), addressing a common source of confusion for readers outside the score-based modeling community.

**Temporal Integral Intuition.**  The log-density ratio can be written as $\log r(\boldsymbol{x}) = \int_0^1 \partial_t \log p_t(\boldsymbol{x}) \mathrm{d}t$. Here, $\partial_t \log p_t(\boldsymbol{x})$ is the instantaneous rate of change of the log-density along a smooth interpolation $p_t$. The integral simply accumulates these infinitesimal changes over $t \in [0, 1]$. Because the path is smooth and non-vanishing, this temporal accumulation remains numerically stable. This avoids the divergence that occurs when directly computing $\log \frac{p_1(\boldsymbol{x})}{p_0(\boldsymbol{x})}$ between distributions with little or no overlapping support (the density-chasm problem (Rhodes et al., 2020)).

**Path Schedules Used in This Paper.**  Let $\boldsymbol{x}_0$ and $\boldsymbol{x}_1$ be samples drawn from $p_0$ and $p_1$, respectively. We consider two standard path schedules. The first is the Linear path, a widely-used schedule in stochastic interpolants (Albergo et al., 2023), defined by $a_t = 1 - t$ and $b_t = t$. The second is the variance-preserving (VP) path (Song et al., 2021), satisfying $a_t = \exp\left(-0.25t^2(\beta_1 - \beta_0) - 0.5t\beta_0\right)$ and $b_t = \sqrt{1 - a_t^2}$. We use the standard diffusion constants $\beta_0 = 0.1$ and $\beta_1 = 20$.

**Path Visualization.**  Fig. 5 shows the VP path on the checkerboard dataset. The samples $\boldsymbol{x}_t$ and densities $p_t$ evolve smoothly from a simple Gaussian ($p_0$) to a complex multimodal target ($p_1$). The color-coded time steps illustrate that the support remains connected and well-behaved for all $t \in (0, 1)$, confirming the existence of a tractable and stable path integral.

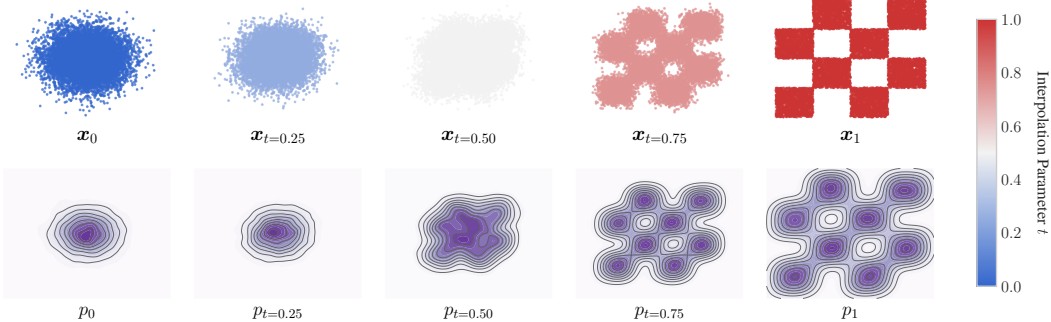

Figure 5: Traditional DRE methods require directly estimating $\log \frac{p_1}{p_0}$ between potentially non-overlapping densities, leading to numerical instability. Continuous score-based DRE avoids this by constructing a smooth interpolating path $\{p_t\}_{t \in [0,1]}$. The figure visualizes this path at five time steps ($t \in \{0, 0.25, 0.50, 0.75, 1.0\}$) under a variance-preserving (VP) schedule, showing the transition from a simple Gaussian ($p_0$) to a multimodal checkerboard distribution ($p_1$). Along the path, all intermediate densities remain connected and well-behaved, turning the hard ratio estimation into the tractable path integral $\int_0^1 \partial_t \log p_t(\boldsymbol{x}) \mathrm{d}t$. The top row shows sample evolution (color-coded by $t$), and the bottom row shows the corresponding density contours.

### C.1.2 Joint Score Matching

Let $s_t = \partial_t \log p_t(\boldsymbol{x})$ and $\boldsymbol{s_x} = \nabla_{\boldsymbol{x}} \log p_t(\boldsymbol{x})$ be the ***time score*** and ***data score***, respectively. In this section, we integrate the parameterized time score model $s_t^{\boldsymbol{\theta}} \in \mathbb{R}$ and data score model $\boldsymbol{s_x^{\theta}} \in \mathbb{R}^d$ to formulate the joint score model $\boldsymbol{s_{t,x}^{\theta}} : [s_t^{\boldsymbol{\theta}}, \boldsymbol{s_x^{\theta}}] \in \mathbb{R}^{d+1}$. This joint score is incorporated into the training objective defined in Eq. (15), resulting in a joint score matching objective (Choi et al., 2022):

$$
\begin{aligned}
\mathcal{L}_{\text{joint}}(\boldsymbol{\theta}) = {} & 2\mathbb{E}_{p_0(\boldsymbol{x}_0)p_1(\boldsymbol{x}_1)}[\lambda(0)\boldsymbol{s_{t,x}^{\theta}}(\boldsymbol{x}_0, 0)[t] - \lambda(1)\boldsymbol{s_{t,x}^{\theta}}(\boldsymbol{x}_1, 1)[t]] \\
& + \mathbb{E}_{p(t)p_t(\boldsymbol{x})}\mathbb{E}_{p(\boldsymbol{v})}\left[2\lambda(t)\partial_t \boldsymbol{s_{t,x}^{\theta}}(\boldsymbol{x}, t)[t] + 2\lambda'(t)\boldsymbol{s_{t,x}^{\theta}}(\boldsymbol{x}, t)[t] \right. \\
& \left. + \lambda(t)\|\boldsymbol{s_{t,x}^{\theta}}(\boldsymbol{x}, t)[\boldsymbol{x}]\|_2^2 + 2\lambda(t)\boldsymbol{v}^{\mathsf{T}}\nabla_{\boldsymbol{x}}\boldsymbol{s_{t,x}^{\theta}}(\boldsymbol{x}, t)[\boldsymbol{x}]\boldsymbol{v}\right],
\end{aligned}
\tag{49}
$$

where $\boldsymbol{v} \sim p(\boldsymbol{v}) = \mathcal{N}(\boldsymbol{0}, \boldsymbol{I}_d)$ follows a standard Gaussian distribution, the terms $\boldsymbol{s_{t,x}^{\theta}}(\boldsymbol{x}, t)[\boldsymbol{x}]$ and $\boldsymbol{s_{t,x}^{\theta}}(\boldsymbol{x}, t)[t]$ represent the data and time score components of $\boldsymbol{s_{t,x}^{\theta}}(\boldsymbol{x}, t)$, respectively.

### C.1.3 Training Procedure

In each training step, we sample a batch of pairs $(\boldsymbol{x}_0, \boldsymbol{x}_1)$ from the source and target distributions, $p_0$ and $p_1$, respectively. We also sample a time $t$ from a distribution $p(t)$ over $[0, 1]$. The interpolated sample $\boldsymbol{x}_t$ is then constructed via a interpolation $\boldsymbol{x}_t = a_t\boldsymbol{x}_0 + b_t\boldsymbol{x}_1$. We use the coefficients $(a_t, b_t)$ corresponding to the variance-preserving (VP) and linear path schedules following Choi et al. (2022); Chen et al. (2025), as detailed in Sec. C.1.1. The detailed training process is outlined in Algorithm 3.

---

**Algorithm 3** Training of OS-DRE

---

**Input:** Data distributions $p_0$ and $p_1$, number of basis functions $K$.
**Output:** Trained model parameters $\boldsymbol{\theta}^\star$ and $\{\sigma_k^\star\}_{k=1}^K$.
1: Initialize trainable parameters $\boldsymbol{\theta}$ of neural network NN and shape parameters $\{\sigma_k\}_{k=1}^K$.
2: Define fixed, quasi-uniform centers $\{c_k\}_{k=1}^K$ over $[0, 1]$.
3: **for** each training step **do**
4:     Sample a batch: $\boldsymbol{x}_0 \sim p_0, \boldsymbol{x}_1 \sim p_1, t \sim p(t)$.
5:     Construct interpolated samples $\boldsymbol{x}_t = a_t\boldsymbol{x}_0 + b_t\boldsymbol{x}_1$ (see Sec. C.1.1 for details).
6:     Compute coefficients: $\{h_k^{\boldsymbol{\theta}}(\boldsymbol{x})\}_{k=1}^K \leftarrow \text{NN}(\boldsymbol{x}; \boldsymbol{\theta})$ for each sample in the batch.
7:     Construct score model $s_t^{\boldsymbol{\theta}}$ and its derivative $\partial_t s_t^{\boldsymbol{\theta}}$ using the model definition in Sec. 4.4.
8:     Compute the STSM loss $\mathcal{L}_{\text{STSM}}(\boldsymbol{\theta})$ using Eq. (15).
9:     Update trainable parameters $\boldsymbol{\theta}$ and $\{\sigma_k\}_{k=1}^K$ via gradient descent on the loss.
10: **end for**

---

## C.2 Model Parameterization and Implementation Details

Our implementation of the OS-DRE framework consists of two main components: the neural network that parameterizes the spatial coefficients and the RBF temporal basis itself.

### C.2.1 Implementation Details

**Network Architecture.** Our model maps an input sample $\boldsymbol{x}$ to its $K$ spatial coefficients through a feed-forward network with a backbone and a single output head. The backbone is formed by a sequence of residual blocks that transform $\boldsymbol{x}$ into a high-level feature embedding, which is then fed into the spatial-coefficient head, which applies another set of residual blocks and a final linear layer to produce a $K$-dimensional vector. The output directly corresponds to the spatial coefficients:

$$
[h_1^{\boldsymbol{\theta}}(\boldsymbol{x}), h_2^{\boldsymbol{\theta}}(\boldsymbol{x}), \dots, h_K^{\boldsymbol{\theta}}(\boldsymbol{x})] = \text{NN}(\boldsymbol{x}; \boldsymbol{\theta}).
\tag{50}
$$

For each input $\boldsymbol{x}$, the network computes all $K$ scalar coefficients in a single forward pass.

**Trainable Parameters.** The trainable parameters of our model consist of two groups: (1) The parameters (weights and biases) of NN, collectively denoted by $\boldsymbol{\theta}$; (2) The shape parameters $\{\sigma_k\}_{k=1}^K$ of the RBF temporal basis $\{g_k\}_{k=1}^K$. The RBF centers $\{c_k\}_{k=1}^K$ are fixed hyperparameters, chosen as a quasi-uniform grid over $[0, 1]$ to satisfy the theoretical conditions. All trainable parameters are optimized jointly by minimizing the STSM loss $\mathcal{L}_{\text{STSM}}$ defined in Eq. (15).

### C.2.2 Neural Network Parameterization

To implement OS-DRE, we employ a neural network to approximate the joint score. Its core function is to map an input $\boldsymbol{x}$ to the spatial coefficients $\{h_k(\boldsymbol{x})\}_{k=1}^K$ and data score. The network consists of a shared backbone and two lightweight task-specific heads for time and data score estimation.

**Architectural Overview.** The model computes the time score $s_t(\boldsymbol{x}, t)$ and, optionally, the data score $\boldsymbol{s_x}(\boldsymbol{x}, t) \triangleq \nabla_{\boldsymbol{x}} \log p_t(\boldsymbol{x})$ for joint score matching (see Eq. (49)). The architecture consists of three main components:

- **Shared Backbone:** Extracts a high-level feature embedding from $\boldsymbol{x}$. The backbone is a stack of residual blocks mapping $\boldsymbol{x}$ to a latent representation $\Phi(\boldsymbol{x}) \in \mathbb{R}^{d_{\text{hidden}}}$. This shared embedding serves as input to both heads.

- **Spatial Coefficient Head:** Predicts the $K$ spatial coefficients $\{h_k^{\boldsymbol{\theta}}(\boldsymbol{x})\}_{k=1}^K$. This head processes $\Phi(\boldsymbol{x})$ through additional residual blocks and a final linear layer nn.Linear$(d_{\text{hidden}}, K)$, producing

$$[h_0^{\boldsymbol{\theta}}(\boldsymbol{x}), h_1^{\boldsymbol{\theta}}(\boldsymbol{x}), \ldots, h_{K-1}^{\boldsymbol{\theta}}(\boldsymbol{x})] = \text{SpatialCoefficientHead}(\Phi(\boldsymbol{x})). \tag{51}$$

  Hence, all $K$ coefficients are predicted in one forward pass.

- **Data Score Head:** (Optional) Predicts the data score $\boldsymbol{s_x^{\boldsymbol{\theta}}}(\boldsymbol{x}, t)$. For joint score matching, this head augments $\Phi(\boldsymbol{x})$ with a positional encoding of time $t$ (Vaswani et al., 2017). The fused representation is processed by residual blocks and projected to a $d$-dimensional output approximating $\boldsymbol{s_x}(\boldsymbol{x}, t)$.

**Algorithm 4** PyTorch implementation of RBF-based Analytic Frame (for instance, Gaussian RBF).

```python
class GaussianRBFFrame(nn.Module):
    """Gaussian Radial Basis Function (RBF)."""
    def __init__(self, K):
        super().__init__(K)
        pi = torch.tensor(torch.pi)
        self.register_buffer("sqrt_pi", torch.sqrt(pi))
        self.sigma_fn = lambda : torch.exp(self.log_sigma)

    @property
    def sigma(self):
        return self._compute_sigma()

    def _compute_sigma(self):
        return self.sigma_fn()

    def forward(self, t):
        sigma_squared = self.sigma ** 2
        squared_dist = (t - self.c_k) ** 2    # [batch_size, K]
        return torch.exp(-squared_dist/ sigma_squared)#[batch_size, K]

    def grad_function(self, t):
        sigma_squared = self.sigma ** 2
        squared_distances = (t - self.c_k) ** 2
        exp_term = torch.exp(-squared_distances / sigma_squared)
        return -2*(t-self.c_k)/sigma_squared*exp_term #[batch_size, K]

    def _compute_integrals(self):
        sigma = self.sigma
        sigma_sqrt_pi = sigma * self.sqrt_pi
        erf_term_1 = torch.erf((1 - self.c_k) / sigma)
        erf_term_2 = torch.erf(self.c_k / sigma)
        return (sigma_sqrt_pi / 2) * (erf_term_1 + erf_term_2)
```

## C.3 Experimental Settings and Results for Density Estimation

In density estimation, let $p_0(\boldsymbol{x}) = \mathcal{N}(\mathbf{0}, \boldsymbol{I}_d)$ be a simple noise distribution, and $p_1(\boldsymbol{x})$ denote the complex and intractable data distribution. The log-likelihood of $p_1$ for a given sample $\boldsymbol{x}$ can be estimated as $\log p_1(\boldsymbol{x}) = \log r(\boldsymbol{x}) + \log p_0(\boldsymbol{x})$, where $r(\boldsymbol{x}) = p_1(\boldsymbol{x})/p_0(\boldsymbol{x})$ is the density ratio between $p_1$ and $p_0$. After training, the estimated log-density ratio $\log \hat{r}$ can be derived based on Eq. (16). Thus, the log-likelihood of $p_1$ can be estimated as $\log p_1(\boldsymbol{x}) \approx \log \hat{r}(\boldsymbol{x}) + \log p_0(\boldsymbol{x})$.

**Structured and Multimodal Datasets.** This section provides a detailed analysis of the density estimation results presented in Fig. 7, covering all nine benchmark datasets. In all experiments, our OS-DRE model was configured with $K = 400$ basis functions using the RQ kernel and was restricted to NFE = 1 for inference. The baseline methods, DRE-$\infty$ and D³RE, were evaluated using a simple quadrature method (the trapezoidal rule) with a fixed NFE = 2. We optimize the model using the joint score matching loss with a learning rate of 0.01. The batch size is set to $10,000$, with 100 batches per epoch. The weighting function is defined as $\lambda(t) = t(1 - t)$.

- **Disconnected Topologies.** The circles and rings datasets test the model's ability to capture distributions with multiple, disconnected components and assign zero density to the regions between them. Fig. 7 shows that OS-DRE perfectly learns both topologies, generating crisp rings with sharp boundaries. The baselines, particularly DRE-$\infty$, struggle with this, producing blurry estimates that incorrectly assign density to the space between the rings. The 8gaussians dataset, featuring eight distinct clusters, further showcases this strength. OS-DRE accurately identifies and models all eight modes, whereas the competing methods tend to merge some of the clusters.

- **Intricate Structures.** The 2spirals, pinwheel, and swissroll datasets feature highly structured, non-linear manifolds that require the model to learn complex, curving paths. OS-DRE demonstrates exceptional performance on all three, accurately tracing the thin spiral arms and the swiss roll manifold. D³RE captures the general shape but loses significant detail, while DRE-$\infty$ fails to resolve the structures, resulting in a single, diffuse cloud of density.

- **Discontinuous and Branching Densities.** The checkerboard dataset presents a particularly difficult challenge with its discontinuous, grid-like density. OS-DRE successfully recovers the sharp, alternating high-density squares, a task where both baseline methods fail, producing heavily smoothed and inaccurate approximations. Similarly, the tree dataset is designed to assess a model's capacity to generate sharp, branching topological structures. OS-DRE excels, yielding crisp, well-defined branches. In contrast, the solver-based methods are unable to capture these fine details, illustrating a fundamental advantage of our analytic, one-step approach for modeling distributions with complex, high-frequency features. The moons dataset further confirms this, with OS-DRE producing significantly sharper and better-separated modes than the baselines.

**Real-world Tabular Datasets.** We evaluate on five tabular datasets that are standard benchmarks in density estimation: POWER, GAS, HEPMASS, MINIBOONE, and BSDS300. These datasets pose challenging, non-Gaussian structures with unknown generative processes and complex correlations, making them suitable for testing model expressiveness. We follow the preprocessing and data splits of Papamakarios et al. (2017); Grathwohl et al. (2019) for a fair comparison. All baseline methods were evaluated using a quadrature scheme with varying numbers of function evaluations (NFE = $\{2, 5, 10, 50\}$). Our method, OS-DRE, was evaluated with a fixed NFE = 1. In this experiment, $K$ is set to 400. We use joint score matching loss with learning rate 0.01. The weighting function is set to $\lambda(t) = t(1 - t)$.

The full quantitative results are presented in Tab. 1. The results clearly demonstrate the superiority of OS-DRE. Across all five datasets, OS-DRE with an appropriate RBF kernel achieves a significantly lower (better) NLL than both DRE-$\infty$ and D³RE, regardless of the NFE allocated to the baselines.

An interesting observation is the instability of the baseline methods. Their performance does not consistently improve with an increased NFE. For example, on MINIBOONE, the performance of DRE-$\infty$ is better at NFE = 10 than at NFE = 50. This highlights the inherent difficulty and potential instability of relying on numerical quadrature for complex, high-dimensional score functions. In contrast, OS-DRE's analytic, one-step computation is deterministic and robust.

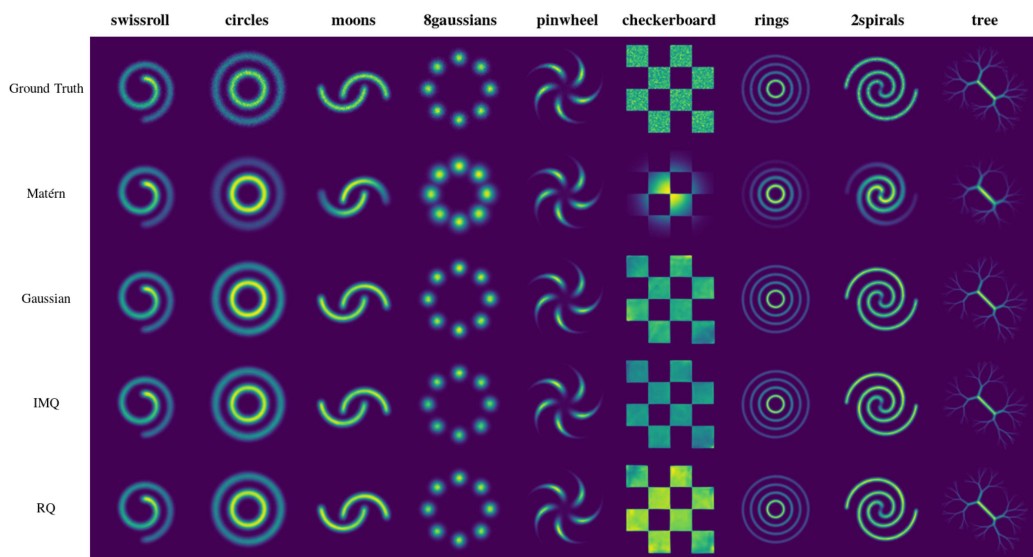

Figure 6: Ablation study on the choice of RBF kernel for structured and multimodal datasets.

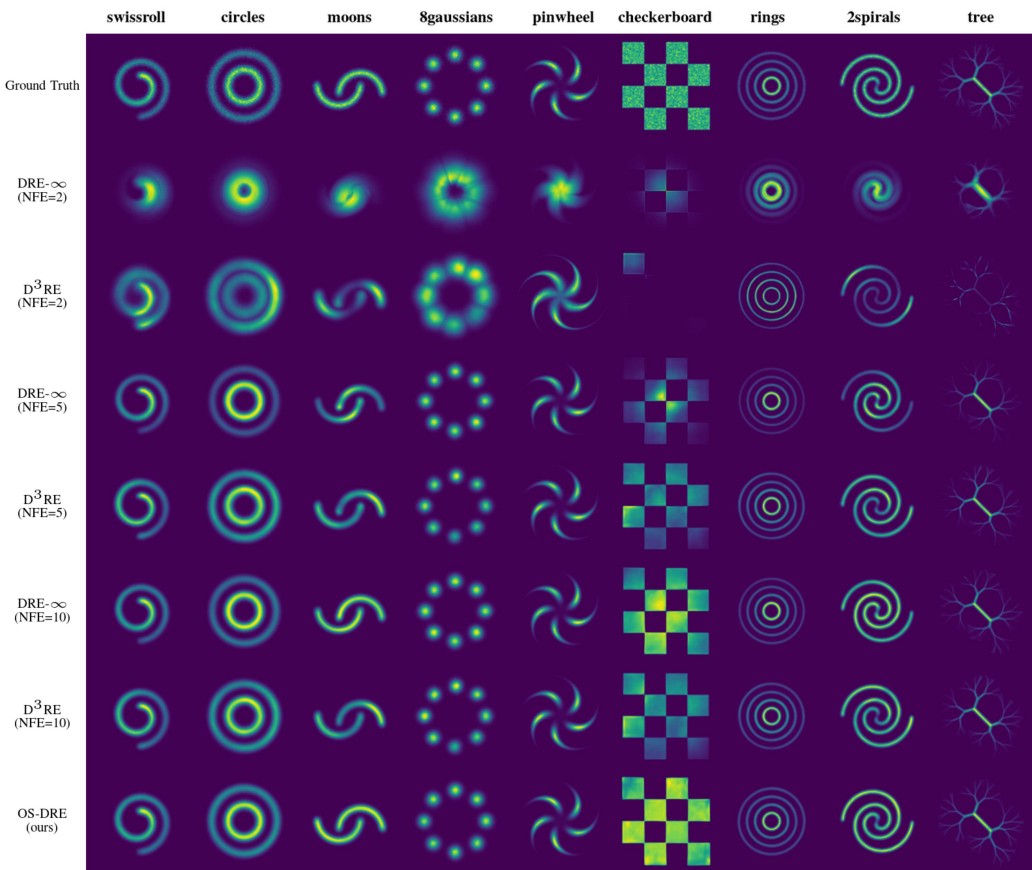

Figure 7: Comparison of density estimates using three score-based DRE methods on nine datasets.

Finally, we observe a clear performance difference among the RBF kernels within OS-DRE. While all kernels outperform the baselines, the IMQ and RQ kernels consistently deliver the best or near-best performance. The Gaussian kernel is also a strong performer, particularly on GAS and BSDS300.

This confirms the importance of the kernel choice as a key hyperparameter, as discussed in our theoretical analysis, with the heavier-tailed IMQ and RQ kernels often providing the best inductive bias for these complex, real-world data distributions.

**Energy-based modeling on MNIST.** We applied our OS-DRE framework for density estimation on the MNIST dataset, leveraging pre-trained energy-based models (EBMs) as the target density $p_1(\boldsymbol{x})$. We specifically use the setup described in Chen et al. (2025) and replicate the results for DRE-$\infty$ and D$^3$RE for a direct comparison. Let $p_1(\boldsymbol{x})$ denote the MNIST data distribution, and $p_0(\boldsymbol{x}) = \mathcal{N}(\mathbf{0}, \boldsymbol{I}_d)$ be the simple Gaussian noise distribution. We applied an variance-preserving interpolant (Song et al., 2021; Choi et al., 2022) of the form $\mathbf{x}_t = a_t\mathbf{x}_0 + b_t\text{EBM}(\mathbf{x}_1) + \sqrt{t(1-t)\gamma^2}\mathbf{z}$, where $\mathbf{x}_0 \sim p_0(\boldsymbol{x}), \mathbf{x}_1 \sim p_1(\boldsymbol{x}), \mathbf{z} \sim \mathcal{N}(\mathbf{0}, \boldsymbol{I}_d)$, $a_t = \exp\{-0.25(b_{\max} - b_{\min})t^2 - 0.5b_{\min}t\}$ and $b_t = \sqrt{1 - a_t^2}$. $b_{\min}$ and $b_{\max}$ are set to 0.1 and 20, respectively. We employ the joint score matching objective and set $\gamma^2 = 2$, consistent with the framework used in Chen et al. (2025). We use IMQ kernel.

A specific advantage of OS-DRE is its one-step evaluation of the log-density ratio $\log \hat{r}(\boldsymbol{x}) = \sum_{k=1}^K h_k^{\boldsymbol{\theta}}(\boldsymbol{x})\bar{g}_k$. We exploit this speed to introduce an additional regularization term during training. This term minimizes the NLL of the MNIST datasets:

$$\mathcal{L}_{\text{NLL-reg}}(\boldsymbol{\theta}) = -\frac{1}{2\ln 2}\mathbb{E}_{p_1(\boldsymbol{x})}\left[\log \hat{r}(\text{EBM}(\boldsymbol{x})) + \log p_0(\text{EBM}(\boldsymbol{x}))\right]. \tag{52}$$

This regularization helps align the OS-DRE prediction directly with the desired log-density, which we find beneficial for high-dimensional, complex data.

## C.4 Experimental Settings and Results for $f$-divergence Estimation

**Continual Learning.** To demonstrate the applicability of OS-DRE to online settings, such as real-time change point detection, we evaluate its ability to track dynamically evolving distributions. In this continual learning setup, the target distribution $p_t$ shifts over discrete timesteps, creating a challenging environment that requires the model to continuously adapt to and quantify the change from a fixed source distribution $p_0$. We measure this ability by estimating the KL-divergence between $p_0$ and the evolving target $p_t$ at each step. The following complex benchmarks are used.

- **Linearly Drifting Gaussian.** This benchmark simulates a gradual, linear drift in both the mean and covariance of a Gaussian distribution, testing the model's ability to track a smoothly evolving target. The source distribution is a standard $d$-dimensional Gaussian, $p_0(\boldsymbol{x}) = \mathcal{N}(\mathbf{0}, \boldsymbol{I}_d)$. At each discrete step $s$, the target distribution $p_t$ is defined as $p_t(\boldsymbol{x}) = \mathcal{N}(\boldsymbol{\mu}_s, \Sigma_s)$, where the parameters evolve linearly:

$$\boldsymbol{\mu}_s = s \cdot \Delta\boldsymbol{\mu}, \quad \Sigma_s = (1 - s \cdot \Delta\sigma)\boldsymbol{I}_d, \tag{53}$$

  with small, constant drift rates $\Delta\boldsymbol{\mu}$ and $\Delta\sigma$.

- **Progressive Noise Corruption.** Inspired by the Gaussian noise corruption in the CIFAR-10-C benchmark (Hendrycks & Dietterich, 2019), this task evaluates the model's response to a progressive increase in isotropic variance, simulating a common type of data corruption. The source distribution is $p_0(\boldsymbol{x}) = \mathcal{N}(\mathbf{0}, \boldsymbol{I}_d)$. At each step $s$, the target distribution is a zero-mean Gaussian with a linearly increasing covariance $p_t(\boldsymbol{x}) = \mathcal{N}(\mathbf{0}, \sigma_s^2 \boldsymbol{I}_d)$, where $\sigma_s^2 = 1 + s \cdot \Delta\sigma^2$, with $\Delta\sigma^2$ being a constant factor determining the rate of variance inflation.

- **Controlled Divergence Shift.** We follow the setup in Zhang et al. (2023) to derive this benchmark. This setup provides a stringent test of the model's ability to track a distribution whose mean shifts in a random direction at each step, while the KL divergence between the current and initial distribution is precisely controlled. The source distribution is again $p_0(\boldsymbol{x}) = \mathcal{N}(\mathbf{0}, \boldsymbol{I}_d)$. At each step $s$, the target distribution $p_t(\boldsymbol{x}) = \mathcal{N}(\boldsymbol{\mu}_s, \boldsymbol{I}_d)$ is defined such that the KL divergence $\text{KL}(p_t\|p_0) = s$. Since the KL divergence for two Gaussians with identity covariance is $0.5\|\boldsymbol{\mu}_s\|^2$, the mean vector $\boldsymbol{\mu}_s$ is constructed to satisfy $\|\boldsymbol{\mu}_s\| = \sqrt{2s}$, with its direction chosen uniformly at random on the unit hypersphere at each step.

The full results are presented in Fig. 3. The plots clearly show the superior stability and responsiveness of OS-DRE across all three dynamic scenarios.

- On the **Linearly Drifting Gaussian** benchmark (Fig. 3a), OS-DRE, particularly with the IMQ and RQ kernels, provides a smooth and low-variance estimate that closely follows the ground-truth KL divergence. In contrast, DRE-$\infty$ shows a significant underestimation bias, while D$^3$RE's estimates are plagued by extremely high variance, making them unreliable.

- The **Progressive Noise Corruption** task (Fig. 3b) presents a more challenging, accelerating shift. OS-DRE again provides stable estimates that follow the general trend of the ground truth. The baseline methods fail dramatically in this setting, with their estimates exhibiting massive variance and becoming completely decorrelated from the ground truth as the corruption intensifies.

- In the **Controlled Divergence Shift** experiment (Fig. 3c), the KL divergence increases in discrete steps. OS-DRE demonstrates excellent responsiveness. Its estimates are sharp and quickly adapt to the new ground truth level after each change point, with very low variance. The baselines, especially D$^3$RE, are characterized by such high variance that they are unable to reliably detect these discrete changes.

These results collectively highlight the key advantage of our analytic, solver-free approach in continual learning settings. By avoiding the iterative computations that can accumulate error and lead to instability, OS-DRE provides a real-time, robust, and reliable tool for tracking distributional changes.

**Mutual Information Estimation.** Mutual information (MI) measures the dependency between two random variables $\mathbf{x} \sim p(\boldsymbol{x})$ and $\mathbf{y} \sim q(\boldsymbol{y})$, quantifying how much information one variable contains about the other. The MI between $\mathbf{x}$ and $\mathbf{y}$ is defined as $\mathrm{MI}(\mathbf{x}, \mathbf{y}) = \mathbb{E}_{p(\boldsymbol{x},\boldsymbol{y})} \left[ \log \frac{p(\boldsymbol{x},\boldsymbol{y})}{p(\boldsymbol{x})q(\boldsymbol{y})} \right]$, which we approximate using DRE.

BEYOND NORMAL: GEOMETRICALLY PATHOLOGICAL DISTRIBUTIONS. We evaluate the performance of OS-DRE and baseline methods on four mutual information (MI) estimation tasks involving geometrically challenging distributions, inspired by the benchmark suite from Czyż et al. (2023). The parameter $\rho$ controls the strength of the dependency between the two random variables. The four benchmarks are detailed below:

- **Half-Cube Map.** This task tests robustness to heavy-tailed data. Correlated Gaussian variables $(\mathbf{x}, \mathbf{y})$ are transformed by the homeomorphism $\mathbf{x}' = \mathrm{sign}(\mathbf{x})|\mathbf{x}|^{3/2}$ and $\mathbf{y}' = \mathrm{sign}(\mathbf{y})|\mathbf{y}|^{3/2}$. While this preserves the true MI, $I(\mathbf{x}'; \mathbf{y}') = I(\mathbf{x}; \mathbf{y})$, it creates distributions with significantly heavier tails that challenge methods relying on local density assumptions.

- **Asinh Mapping.** Designed to test performance on distributions with highly concentrated densities, this task applies the inverse hyperbolic sine transformation, $\mathrm{asinh}(z) = \log(z + \sqrt{z^2 + 1})$, to two independent Gaussian variables. This creates sharp peaks and regions of high curvature that can cause numerical instability in many estimators.

- **Additive Noise.** This scenario evaluates performance on distributions with sharp, non-differentiable boundaries. We define $\mathbf{y} = \mathbf{x} + \mathbf{n}$, where $\mathbf{x} \sim \mathcal{U}(0, 1)$ and $\mathbf{n} \sim \mathcal{U}(-\epsilon, \epsilon)$ are independent. The resulting joint distribution has a fragmented, piecewise-constant support that violates the smoothness assumptions of many score-based methods. The true MI is $I(\mathbf{x}; \mathbf{y}) = \log(2\epsilon) + 0.5$ for $\epsilon \leq 0.5$.

- **Gamma-Exponential.** This task features a complex, non-linear, and asymmetric dependency. One variable is drawn from a Gamma distribution, $\mathbf{x} \sim \mathrm{Gamma}(\rho, 1)$, and its value is then used as the rate parameter for an Exponential distribution from which the second variable is drawn: $\mathbf{y} \mid \mathbf{x} = x \sim \mathrm{Exponential}(x)$. The true mutual information is $I(\mathbf{x}; \mathbf{y}) = \psi(\rho + 1) - \log(\rho)$, where $\psi$ is the digamma function.

The full results are presented in Tab. 6. OS-DRE demonstrates a clear advantage across all four challenging scenarios. On the **Half-Cube** and **Asinh mapping** tasks, OS-DRE, particularly with the Gaussian and Matérn kernels, achieves an MSE that is orders of magnitude lower than the baselines across nearly all correlation levels. This indicates that our analytic basis is better equipped to handle the heavy tails and high-curvature densities introduced by these transformations. In the **Additive Noise** scenario, which features sharp discontinuities, the IMQ kernel shows remarkable stability, consistently outperforming the baselines. This suggests that the global nature of the IMQ basis

functions provides a more robust representation for distributions with non-differentiable boundaries. Finally, the **Gamma-Exponential** task highlights the flexibility of our approach. The Matérn kernel, which has limited smoothness, provides the most accurate estimates, significantly outperforming the baselines, especially in the high-dependency regime ($\rho > 1.2$). This demonstrates the benefit of being able to select a kernel whose inductive bias (in this case, limited smoothness) matches the complex dependency structure of the data.

Table 6: MSE results for MI estimation on four geometrically pathological datasets. The top row of each sub-table indicates the varying correlation coefficient $\rho$. Our OS-DRE method demonstrates consistently superior or competitive performance across the wide range of challenging data geometries.

(a) MSE results for the Half-Cube Map dataset.

| Method | RBF Kernel | -0.9 | -0.8 | -0.7 | -0.6 | -0.5 | -0.4 | -0.3 | -0.2 | -0.1 | 0.0 | 0.1 | 0.2 | 0.3 | 0.4 | 0.5 | 0.6 | 0.7 | 0.8 | 0.9 |
|---|---|---|---|---|---|---|---|---|---|---|---|---|---|---|---|---|---|---|---|---|
| DRE-$\infty$ | - | 0.0054 | 0.0029 | 0.0037 | 0.0023 | 0.0015 | 0.0021 | 0.0013 | 0.0015 | 0.0010 | 0.0014 | 0.0015 | 0.0022 | 0.0015 | 0.0027 | 0.0033 | 0.0029 | 0.0030 | 0.0037 | 0.0056 |
| D³RE | - | 0.0014 | 0.0006 | 0.0005 | 0.0003 | 0.0003 | 0.0003 | 0.0004 | 0.0003 | 0.0003 | 0.0003 | 0.0004 | 0.0009 | 0.0012 | 0.0006 | 0.0005 | 0.0004 | 0.0006 | 0.0008 | 0.0012 |
| **OS-DRE (ours)** | Matérn | 0.0000 | 0.0000 | 0.0000 | 0.0000 | 0.0000 | 0.0000 | **0.0000** | 0.0000 | **0.0000** | 0.0000 | 0.0000 | 0.0000 | **0.0000** | **0.0000** | **0.0000** | 0.0000 | 0.0000 | **0.0000** | 0.0000 |
| **OS-DRE (ours)** | Gaussian | 0.0002 | **0.0000** | 0.0000 | **0.0000** | 0.0000 | **0.0000** | 0.0000 | 0.0000 | 0.0000 | **0.0000** | **0.0000** | **0.0000** | 0.0000 | 0.0000 | 0.0000 | 0.0000 | 0.0000 | 0.0000 | 0.0000 |
| **OS-DRE (ours)** | IMQ | 0.0001 | 0.0001 | 0.0001 | 0.0001 | 0.0001 | 0.0002 | 0.0001 | 0.0001 | 0.0001 | 0.0001 | 0.0002 | 0.0001 | 0.0001 | 0.0001 | 0.0001 | 0.0001 | 0.0002 | 0.0001 | 0.0002 |
| **OS-DRE (ours)** | RQ | **0.0000** | **0.0000** | **0.0000** | 0.0000 | **0.0000** | 0.0000 | 0.0000 | 0.0000 | 0.0000 | 0.0000 | 0.0000 | 0.0000 | 0.0000 | 0.0000 | 0.0000 | 0.0000 | 0.0000 | 0.0000 | **0.0000** |

(b) MSE results for the Asinh Mapping dataset.

| Method | RBF Kernel | -0.9 | -0.8 | -0.7 | -0.6 | -0.5 | -0.4 | -0.3 | -0.2 | -0.1 | 0.0 | 0.1 | 0.2 | 0.3 | 0.4 | 0.5 | 0.6 | 0.7 | 0.8 | 0.9 |
|---|---|---|---|---|---|---|---|---|---|---|---|---|---|---|---|---|---|---|---|---|
| DRE-$\infty$ | - | 0.0005 | 0.0006 | 0.0006 | 0.0006 | 0.0005 | 0.0004 | 0.0004 | 0.0004 | 0.0003 | 0.0003 | 0.0005 | 0.0006 | 0.0007 | 0.0006 | 0.0007 | 0.0006 | 0.0006 | 0.0007 | 0.0005 |
| D³RE | - | 0.0014 | 0.0010 | 0.0006 | 0.0006 | 0.0005 | 0.0003 | 0.0003 | 0.0003 | 0.0003 | 0.0003 | 0.0004 | 0.0004 | 0.0004 | 0.0004 | 0.0005 | 0.0006 | 0.0005 | 0.0011 | 0.0014 |
| **OS-DRE (ours)** | Matérn | 0.0004 | **0.0001** | **0.0001** | **0.0001** | **0.0001** | 0.0001 | **0.0000** | **0.0000** | 0.0000 | **0.0000** | 0.0000 | **0.0000** | 0.0001 | 0.0001 | 0.0001 | 0.0001 | **0.0001** | 0.0001 | 0.0004 |
| **OS-DRE (ours)** | Gaussian | 0.0002 | **0.0001** | 0.0001 | 0.0001 | **0.0001** | 0.0001 | 0.0000 | 0.0000 | 0.0000 | 0.0000 | 0.0000 | 0.0001 | **0.0001** | 0.0001 | **0.0001** | 0.0001 | **0.0001** | **0.0001** | 0.0002 |
| **OS-DRE (ours)** | IMQ | 0.0002 | 0.0002 | 0.0002 | 0.0002 | 0.0002 | 0.0002 | 0.0002 | 0.0002 | 0.0002 | 0.0002 | 0.0002 | 0.0002 | 0.0003 | 0.0003 | 0.0003 | 0.0003 | 0.0003 | 0.0003 | 0.0004 |
| **OS-DRE (ours)** | RQ | **0.0002** | 0.0002 | 0.0001 | 0.0001 | 0.0001 | 0.0001 | 0.0001 | 0.0001 | 0.0001 | 0.0001 | 0.0001 | 0.0002 | 0.0002 | 0.0002 | 0.0002 | 0.0002 | 0.0002 | 0.0002 | 0.0003 |

(c) MSE results for the Additive Noise dataset.

| Method | RBF Kernel | 0.1 | 0.2 | 0.3 | 0.4 | 0.5 | 0.6 | 0.7 | 0.8 | 0.9 |
|---|---|---|---|---|---|---|---|---|---|---|
| DRE-$\infty$ | - | 0.0029 | 0.0018 | 0.0015 | 0.0012 | 0.0013 | 0.0013 | 0.0011 | 0.0011 | 0.000 |
| D³RE | - | 0.0108 | 0.0077 | 0.0065 | 0.0071 | 0.0085 | 0.0076 | 0.0064 | 0.0045 | 0.0055 |
| **OS-DRE (ours)** | Matérn | 0.0061 | 0.0029 | 0.0017 | 0.0015 | 0.0015 | 0.0013 | 0.0011 | 0.0009 | 0.0008 |
| **OS-DRE (ours)** | Gaussian | 0.0016 | 0.0015 | 0.0016 | **0.0011** | 0.0015 | 0.0014 | 0.0012 | 0.0010 | 0.0010 |
| **OS-DRE (ours)** | IMQ | **0.0010** | **0.0010** | **0.0010** | 0.0012 | **0.0009** | **0.0008** | **0.0007** | **0.0009** | **0.0007** |
| **OS-DRE (ours)** | RQ | 0.0019 | 0.0015 | 0.0015 | 0.0012 | 0.0010 | 0.0010 | 0.0010 | 0.0010 | 0.0009 |

(d) MSE results for the Gamma-Exponential dataset.

| Method | RBF Kernel | 1.0 | 1.1 | 1.2 | 1.3 | 1.4 | 1.5 | 1.6 | 1.7 | 1.8 |
|---|---|---|---|---|---|---|---|---|---|---|
| DRE-$\infty$ | - | 2.1328 | 0.8939 | 0.0725 | 0.0115 | 0.0213 | 0.0051 | 0.0114 | 0.0069 | 0.0051 |
| D³RE | - | 0.1919 | 0.1018 | 0.0154 | 0.0119 | 0.0063 | 0.0110 | 0.0050 | 0.0125 | 0.0114 |
| **OS-DRE (ours)** | Matérn | **0.1768** | **0.0315** | **0.0035** | **0.0026** | **0.0008** | 0.0017 | **0.0009** | **0.0006** | **0.0005** |
| **OS-DRE (ours)** | Gaussian | 0.2933 | 0.0503 | 0.0060 | 0.0028 | 0.0032 | **0.0014** | 0.0014 | 0.0007 | 0.0009 |
| **OS-DRE (ours)** | IMQ | 0.2821 | 0.1185 | 0.0901 | 0.0492 | 0.0200 | 0.0275 | 0.0072 | 0.0080 | 0.0087 |
| **OS-DRE (ours)** | RQ | 0.5182 | 0.0925 | 0.0330 | 0.0109 | 0.0052 | 0.0040 | 0.0015 | 0.0015 | 0.0012 |

We also compare OS-DRE with KSG (Kraskov et al., 2004), MINE (Belghazi et al., 2018) and InfoNet (Hu et al., 2024), as illustrated in Fig. 8. Here, Gauss denotes 2-D complex Gaussian distributions, where $q(\mathbf{y}) = \mathcal{N}(\mathbf{0}, \mathbf{\Sigma})$ and $p(\boldsymbol{x}) = \mathcal{N}(\mathbf{0}, \boldsymbol{I}_d)$, with $\mathbf{\Sigma} = [[1, \rho], [\rho, 1]]$ and $\rho$ varying in $[-0.9, 0.9]$. OS-DRE yields MI estimates that closely match the ground truth (with lower mean absolute error (MAE) values), demonstrating high accuracy in Gaussian scenarios.

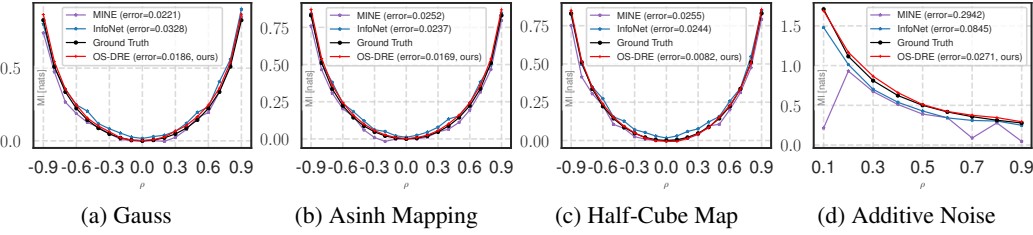

| (a) Gauss | (b) Asinh Mapping | (c) Half-Cube Map | (d) Additive Noise |

Figure 8: Comparison of MI estimates under four complex settings proposed in (Hu et al., 2024) (mean absolute error (MAE) included). OS-DRE consistently outperforms existing methods across all settings, providing estimates that are both accurate and robust. These results highlight the superiority of OS-DRE in estimating MI under challenging, nontrivial distributions.

HIGH-DISCREPANCY & HIGH-DIMENSIONAL DISTRIBUTIONS. To systematically evaluate model robustness as the density-chasm problem becomes progressively more severe, we designed an

experiment for MI estimation between two high-dimensional Gaussian distributions. We define the two distributions as $p_0(\boldsymbol{x}) = \mathcal{N}(\mathbf{0}, \boldsymbol{I}_d)$ and $p_1(\boldsymbol{x}) = \mathcal{N}(\mathbf{0}, \Sigma)$. The covariance matrix $\Sigma$ is constructed to be block-diagonal, where each $2 \times 2$ block along the diagonal is given by $\Lambda = \left( \begin{smallmatrix} 1 & \rho \\ \rho & 1 \end{smallmatrix} \right)$, with $\rho = 0.5$. This structure creates strong pairwise correlations, leading to a highly ill-conditioned covariance matrix, a known challenge for score-based DRE methods (Choi et al., 2022). The experiments are conducted across several dimensions, $d \in \{40, 80, 120, 160\}$, which correspond to true MI values of approximately $\{10, 20, 30, 40\}$ nats, respectively. We report the estimated MI (mean $\pm$ std over 3 seeds) and the Mean Squared Error (MSE).

The full results are presented in Tab. 7. The data clearly illustrates the limitations of iterative, solver-based methods in high-discrepancy scenarios. At low NFE (e.g., NFE= 2), both DRE-$\infty$ and D$^3$RE fail completely, severely underestimating the true MI. Even with a large computational budget of NFE= 50, DRE-$\infty$ provides reasonable estimates, but D$^3$RE remains unstable, collapsing entirely at the highest discrepancy level (MI= 40).

In stark contrast, our one-step OS-DRE demonstrates remarkable robustness. The choice of kernel is critical. The Matérn kernel, which has limited smoothness, struggles as the dimensionality and discrepancy increase, as predicted by theory. However, the infinitely smooth kernels deliver exceptional performance. The Gaussian kernel provides the most accurate estimates for MI levels of 10, 20, and 30 nats, achieving an MSE that is competitive with or better than the best-performing baseline (DRE-$\infty$ at NFE= 50), but with only a single function evaluation. At the most extreme setting of MI= 40, the IMQ kernel proves to be the most robust, delivering the lowest MSE by a significant margin. This highlights the key advantage of our analytic framework: by avoiding the accumulation of numerical errors inherent in iterative solvers, OS-DRE can successfully navigate the density chasm and provide stable, accurate estimates in a single step.

Table 7: Mutual information estimation under high-discrepancy settings (MI $\in \{10, 20, 30, 40\}$ nats). We report the estimated mutual information (mean $\pm$ std) and MSE across different RBF kernels. All timing results were obtained on a single NVIDIA TITAN X GPU. **Bolded** MSE values indicate the best performance for each setting. The best wall-clock time is underlined.

| Method | NFE | RBF Kernel | MI = 10 | | | MI = 20 | | | MI = 30 | | | MI = 40 | | |
|---|---|---|---|---|---|---|---|---|---|---|---|---|---|---|
| | | | Est. MI | MSE | Time (s) | Est. MI | MSE | Time (s) | Est. MI | MSE | Time (s) | Est. MI | MSE | Time (s) |
| DRE-$\infty$ | 2 | - | $1.40_{\pm 0.01}$ | 73.91 | 0.045 | $3.16_{\pm 0.01}$ | 283.52 | 0.045 | $5.21_{\pm 0.01}$ | 614.62 | 0.045 | $5.13_{\pm 0.02}$ | 1215.69 | 0.046 |
| D$^3$RE | 2 | - | $11.61_{\pm 0.08}$ | 2.58 | 0.048 | $21.91_{\pm 0.08}$ | 3.65 | 0.047 | $27.51_{\pm 0.07}$ | 6.21 | 0.046 | $17.64_{\pm 0.17}$ | 500.04 | 0.044 |
| DRE-$\infty$ | 5 | - | $8.31_{\pm 0.05}$ | 2.86 | 0.055 | $17.34_{\pm 0.04}$ | 7.09 | 0.060 | $24.97_{\pm 0.05}$ | 25.29 | 0.068 | $31.61_{\pm 0.06}$ | 70.36 | 0.057 |
| D$^3$RE | 5 | - | $9.91_{\pm 0.04}$ | 0.01 | 0.056 | $19.46_{\pm 0.04}$ | 0.29 | 0.061 | $27.26_{\pm 0.03}$ | 7.50 | 0.058 | $31.24_{\pm 0.05}$ | 76.80 | 0.058 |
| DRE-$\infty$ | 10 | - | $9.48_{\pm 0.06}$ | 0.27 | 0.075 | $19.27_{\pm 0.04}$ | 0.54 | 0.100 | $28.37_{\pm 0.05}$ | 2.66 | 0.085 | $37.34_{\pm 0.06}$ | 7.08 | 0.083 |
| D$^3$RE | 10 | - | $10.13_{\pm 0.04}$ | 0.02 | 0.075 | $20.45_{\pm 0.03}$ | 0.21 | 0.094 | $27.22_{\pm 0.03}$ | 7.72 | 0.076 | $32.27_{\pm 0.04}$ | 59.70 | 0.080 |
| DRE-$\infty$ | 50 | - | $9.84_{\pm 0.06}$ | 0.03 | 0.226 | $19.81_{\pm 0.04}$ | 0.04 | 0.249 | $29.31_{\pm 0.06}$ | 0.48 | 0.228 | $38.06_{\pm 0.07}$ | 3.77 | 0.271 |
| D$^3$RE | 50 | - | $10.07_{\pm 0.04}$ | 0.01 | 0.234 | $20.30_{\pm 0.03}$ | 0.09 | 0.256 | $27.01_{\pm 0.03}$ | 8.94 | 0.256 | $32.37_{\pm 0.04}$ | 58.19 | 0.260 |
| DRE-$\infty$ | 100 | - | $9.87_{\pm 0.06}$ | 0.02 | 0.475 | $19.89_{\pm 0.04}$ | 0.01 | 0.493 | $29.30_{\pm 0.06}$ | 0.50 | 0.478 | $38.18_{\pm 0.07}$ | 3.31 | 0.554 |
| D$^3$RE | 100 | - | $10.01_{\pm 0.04}$ | **0.00** | 0.498 | $20.29_{\pm 0.03}$ | 0.08 | 0.546 | $27.00_{\pm 0.03}$ | 9.00 | 0.487 | $32.37_{\pm 0.04}$ | 58.20 | 0.514 |
| DRE-$\infty$ | 200 | - | $9.86_{\pm 0.06}$ | 0.02 | 0.819 | $19.89_{\pm 0.04}$ | 0.01 | 0.954 | $29.29_{\pm 0.06}$ | 0.50 | 0.879 | $38.18_{\pm 0.07}$ | 3.31 | 0.956 |
| D$^3$RE | 200 | - | $10.04_{\pm 0.04}$ | **0.00** | 0.816 | $20.23_{\pm 0.03}$ | 0.05 | 0.922 | $26.94_{\pm 0.03}$ | 9.36 | 0.907 | $32.43_{\pm 0.04}$ | 57.28 | 0.955 |
| OS-DRE (ours) | 1 | Matérn | $10.31_{\pm 0.02}$ | 0.09 | 0.024 | $15.73_{\pm 0.05}$ | 18.30 | 0.028 | $15.55_{\pm 0.02}$ | 208.98 | 0.032 | $18.65_{\pm 0.15}$ | 456.11 | 0.028 |
| OS-DRE (ours) | 1 | Gaussian | $10.05_{\pm 0.04}$ | 0.01 | 0.025 | $20.03_{\pm 0.04}$ | **0.00** | 0.027 | $29.37_{\pm 0.07}$ | **0.07** | 0.013 | $38.68_{\pm 0.09}$ | 2.30 | 0.014 |
| OS-DRE (ours) | 1 | IMQ | $10.37_{\pm 0.02}$ | 0.11 | 0.035 | $21.25_{\pm 0.04}$ | 1.56 | 0.030 | $28.10_{\pm 0.08}$ | 5.86 | 0.029 | $39.35_{\pm 0.09}$ | **0.47** | 0.028 |
| OS-DRE (ours) | 1 | RQ | $9.89_{\pm 0.03}$ | 0.03 | 0.022 | $19.49_{\pm 0.04}$ | 0.83 | 0.012 | $28.94_{\pm 0.10}$ | 1.52 | 0.012 | $38.92_{\pm 0.07}$ | 1.41 | 0.019 |

Table 8: Accuracy-efficiency trade-off on tabular datasets. Accuracy is measured by NLL, and efficiency is measured by NFE. Lower is better. Each method is evaluated at different NFE settings. The best results are highlighted in **bold**. In this table, OS-DRE uses the IMQ kernel. See Fig. 4a for a visual comparison of this trade-off.

| Dataset | Method | NFE = 1 | NFE = 2 | NFE = 5 | NFE = 10 | NFE = 50 |
|---|---|---|---|---|---|---|
| **POWER** | DRE-$\infty$ | – | $0.05 \pm 1.84$ | $0.35 \pm 0.50$ | $0.03 \pm 0.17$ | $0.25 \pm 0.28$ |
| | D$^3$RE | – | $3.57 \pm 1.84$ | $1.26 \pm 0.38$ | $0.49 \pm 0.39$ | $0.89 \pm 0.33$ |
| | **OS-DRE (ours)** | $\mathbf{-0.69 \pm 0.18}$ | – | – | – | – |
| **GAS** | DRE-$\infty$ | – | $-4.37 \pm 1.44$ | $-3.63 \pm 0.78$ | $-4.34 \pm 0.60$ | $-4.33 \pm 0.71$ |
| | D$^3$RE | – | $5.74 \pm 15.28$ | $-1.15 \pm 4.20$ | $-3.27 \pm 2.00$ | $-3.16 \pm 0.62$ |
| | **OS-DRE (ours)** | $\mathbf{-18.33 \pm 0.04}$ | – | – | – | – |
| **HEPMASS** | DRE-$\infty$ | – | $19.30 \pm 1.31$ | $20.24 \pm 0.47$ | $20.43 \pm 0.52$ | $20.67 \pm 0.57$ |
| | D$^3$RE | – | $23.90 \pm 0.36$ | $21.05 \pm 0.52$ | $20.30 \pm 0.55$ | $20.05 \pm 0.35$ |
| | **OS-DRE (ours)** | $\mathbf{17.45 \pm 0.05}$ | – | – | – | – |
| **MINIBOONE** | DRE-$\infty$ | – | $41.55 \pm 2.07$ | $20.90 \pm 0.84$ | $20.57 \pm 0.93$ | $20.97 \pm 0.51$ |
| | D$^3$RE | – | $55.83 \pm 9.36$ | $43.11 \pm 26.20$ | $42.65 \pm 26.87$ | $42.73 \pm 26.78$ |
| | **OS-DRE (ours)** | $\mathbf{9.97 \pm 0.37}$ | – | – | – | – |
| **BSDS300** | DRE-$\infty$ | – | $-130.68 \pm 4.17$ | $-83.70 \pm 1.35$ | $-87.65 \pm 2.24$ | $-90.24 \pm 2.14$ |
| | D$^3$RE | – | $-149.53 \pm 9.06$ | $-101.97 \pm 1.67$ | $-102.01 \pm 2.43$ | $-78.26 \pm 0.96$ |
| | **OS-DRE (ours)** | $\mathbf{-217.99 \pm 3.39}$ | – | – | – | – |

Table 9: Accuracy-efficiency trade-off in MI estimation. Accuracy is measured by MSE, and efficiency is measured by NFE (NFE $=\in \{1, 2, 5, 10, 50, 100, 200\}$). Lower is better. Results are shown across different MI settings (MI $\in \{10, 20, 30, 40\}$). The best results are highlighted in **bold**. OS-DRE uses the Gaussian RBF kernel. See Fig. 4b for a visual comparison of this trade-off.

| MI Setting | Method | 1 | 2 | 5 | 10 | 50 | 100 | 200 |
|---|---|---|---|---|---|---|---|---|
| MI = 10 | DRE-$\infty$ | – | 73.91 | 2.86 | 0.27 | 0.03 | 0.02 | 0.02 |
| | D$^3$RE | – | 2.58 | 0.01 | 0.02 | 0.01 | **0.00** | **0.00** |
| | **OS-DRE (ours)** | 0.01 | – | – | – | – | – | – |
| MI = 20 | DRE-$\infty$ | – | 283.52 | 7.09 | 0.54 | 0.04 | 0.01 | 0.01 |
| | D$^3$RE | – | 3.65 | 0.29 | 0.21 | 0.09 | 0.08 | 0.05 |
| | **OS-DRE (ours)** | **0.00** | – | – | – | – | – | – |
| MI = 30 | DRE-$\infty$ | – | 614.62 | 25.29 | 2.66 | 0.48 | 0.50 | 0.50 |
| | D$^3$RE | – | 6.21 | 7.50 | 7.72 | 8.94 | 9.00 | 9.00 |
| | **OS-DRE (ours)** | **0.07** | – | – | – | – | – | – |
| MI = 40 | DRE-$\infty$ | – | 1215.69 | 70.36 | 7.08 | 3.77 | 3.31 | 3.31 |
| | D$^3$RE | – | 500.04 | 76.80 | 59.70 | 58.19 | 58.20 | 57.28 |
| | **OS-DRE (ours)** | **2.30** | – | – | – | – | – | – |

