# OpenReview forum: "One-Step Score-Based Density Ratio Estimation: Solver-Free with Analytic Frames"
_ICLR.cc/2026/Conference — Submitted to ICLR 2026_

### Official Review · Reviewer_bFyf · 2025-10-22

**Soundness:** 3
**Presentation:** 3
**Contribution:** 3
**Rating:** 6
**Confidence:** 3

**Summary:**

Score-based Density Ratio Estimation (DRE) methods have garnered attention in recent years. However, previous methods typically model the time scores themselves using neural networks, requiring numerical integrations to obtain the resulting density ratios. The work proposes to resolve the computational challenges by an analytic alternative, which has theoreticall bounded approximation errors. Empirical evidence are provided to demonstrate the good performances.

**Strengths:**

1. The analytical framework is attractive, as both the time scores and the density ratios are tractable.
2. The theoretical analysis on the framework is interesting.

**Weaknesses:**

1. The emprical experiments are generally smaller scale.
2. The authors proposed to use the neural network to predict the spatial coefficients in the density ratio. Perhaps a crude alternative is to model the density ratio itself using a neural network and obtain the time score by differentiating; e.g. [1] essentially uses CTSM [2] to train the model, and [3] proposed a method that is in principle applicable to any-step. Are there practical benefits of the proposed framework apart from computational efficiencies during training?

[1] Learning normalized image densities via dual score matching, Guth et al.

[2] Density Ratio Estimation with Conditional Probability Paths, Yu et al.

[3] Any-Step Density Ratio Estimation via Interval-Annealed Secant Alignment, Chen et al.

**Questions:**

See also the Weaknesses. I am curious about the expressitivity of the proposed framework. For instance, is the proposed method well-behaved in higher dimensional problems, e.g. when training an EBM on MNIST?

---

> ### Author Response · Authors · 2025-11-19
>
> Thank you for your encouraging comments on our analytical framework and theoretical analysis. We address your feedback below.
>
> > W 1; Q: The empirical experiments are generally smaller scale. ...is the proposed method well-behaved in higher dimensional problems, e.g. when training an EBM on MNIST?
>
> Thank you for this valuable suggestion—your comment directly motivated us to evaluate OS-DRE on MNIST energy-based modeling, a standard high-dimensional DRE benchmark. The results are highly encouraging (`Tab. 2 in revised manuscript`, timing measured over the full test set on a single NVIDIA TITAN X GPU ):
>
> | Method | Params | NFE   | Time (s)  | BPD ↓     |
> | - | -| - | --| --------- |
> | DRE-∞      | 11.2M  | 75    | 21.443    | 1.302     |
> | D³RE       | 11.2M  | 75    | 21.424    | 1.281     |
> | **OS-DRE** | 11.5M  | **1** | **0.312** | **1.278** |
>
> A few important clarifications:
>
> - **Reproduced and improved baselines**: The DRE-∞ and D³RE results above are **our own re-implementations**, built upon their official codebases but enhanced with modern training practices from the broader DRE and score-based generative modeling literature (e.g., improved sampling and loss stabilization). Consequently, `our reproduced numbers surpass the originally reported results in both accuracy and speed (for DRE-infty, from ~187s to ~21s)`.
> - **Efficiency advantage stands**: Even against these optimized baselines, OS-DRE achieves **68× faster inference** (NFE=1 vs. 75) and comparable BPD, demonstrating clear practical benefits.
> - **Model capacity note**: The absolute BPD gain is modest, likely because the EBM uses a **fixed pre-trained feature backbone**. We believe performance could improve further with a more expressive base model, which is an interesting direction for future work.
> - **Code contribution**: To ensure fair comparison and reproducibility, we built a **new, modular, and clean PyTorch Lightning-based framework** that now supports full reproduction of both DRE-∞ and D³RE, as well as OS-DRE. This package was finalized during the MNIST experiment, and we plan to release it publicly to support the community. We sincerely thank you—your suggestion significantly improved the completeness and robustness of this codebase.
>
>
> > W 2.1: ...Are there practical benefits of the proposed framework apart from computational efficiencies during training?
>
> **The key practical benefit of OS-DRE beyond training efficiency is its ultra-fast, accurate inference, enabled by analytic integration.**
>
> - OS-DRE computes the log-density ratio with **only one neural network forward pass (NFE = 1)**, thanks to its closed-form solution. In contrast, prior methods like DRE-∞ and D³RE rely on numerical integration, requiring **NFE = 10 to 100+** evaluations per sample during inference.
> - **This speedup comes without sacrificing accuracy**: across all experiments (Tables 1, 4, 7), OS-DRE consistently **matches or outperforms** baselines using NFE = 50. This makes OS-DRE uniquely suited for **real-time, latency-sensitive applications**, such as continual monitoring under distribution drift (Figure 3).
>
>
> > W 2.2 ...crude alternative is to model the density ratio itself using a neural network and obtain the time score by differentiating; e.g. [1] essentially uses CTSM [2] to train the model, and [3] proposed a method ... applicable to any-step.
>
> OS-DRE fundamentally differs from direct ratio modeling and prior CTSM-based methods by enabling `solver-free, exact integration` with NFE = 1, **avoiding both the density-chasm instability and iterative inference.**
>
> - **Directly modeling $r(x)$ is prone to the density-chasm problem  [4, 5].**  While a neural network can approximate $\log r(x)$, this approach fails catastrophically when $p_0$ and $p_1$ have limited overlap, leading to unbounded or divergent estimates. This is precisely why modern DRE shifts to `path-integral score-based formulations`  [4, 5] (including ours; see Sec. 2 for details), which our method advances.
> - [1] address different problems. [2] and [3] share similar training objectives (e.g., CTSM loss) but rely on numerical solvers at test time; [2] uses an ODE solver (NFE ≫ 1, same as DRE-∞), while [3] employs iterative root-finding (reported at NFE = 10). Neither achieves true single-step evaluation.
>
> - **To the best of our knowledge, OS-DRE is the first to eliminate solvers entirely in score-based methods.**  By expressing the time score in a decoupled basis (Sec. 4) and leveraging our closed-form integral (Prop. 4.1), we compute $\int_ 0^1 \partial_ t \log p_t(x)dt$ exactly with NFE=1.
>
> [1] Learning normalized image densities via dual score matching, Guth et al.
>
> [2] Density Ratio Estimation with Conditional Probability Paths, Yu et al.
>
> [3] Any-Step Density Ratio Estimation via Interval-Annealed Secant Alignment, Chen et al.
>
> [4] Liu, Song, et al. Trimmed density ratio estimation. NeurIPS 2017.
>
> [5] Rhodes et al. Telescoping Density-Ratio Estimation. NeurIPS 2020.

---

> > ### Comment · Reviewer_bFyf · 2025-11-19
> >
> > I thank the authors for their response. Concerning the connections to previous works, I am familiar with the density-chasm problem and the motivation of using score-based methods. What I was asking is, while [2] parameterizes the score network which outputs the time score, it can be parameterized using an energy network as well, as shown e.g. in [1], where the gradients of the energy with respect to t gives the time score. As such I tend not to agree that [1] is solving a fundamentally different problem, though the presentation indeed is different. When the model is parameterized as a time-varying EBM as in [1], the DRE problem can also be solved essentially in a single forward step through the network. In this sense, despite the meaningful contributions, I find the claim that "the first to eliminate solvers entirely in score-based methods" too strong.

---

> > > ### Author Response · Authors · 2025-11-21
> > >
> > > Thank you for this thoughtful clarification and for highlighting the important connection to [1]. You are absolutely right that their time-varying energy model provides an elegant mechanism for single-step evaluation by directly learning a normalized density.
> > >
> > > Our contribution centers not just on achieving single-step evaluation, but on **analytically solving the path integral** within score-based DRE framework. Whereas [1] bypass the integral by **enforcing explicit normalization across noise levels**, we retain the flexibility of path-based DRE—a formulation that has been widely shown to mitigate the density-chasm problem [2,3,5, ...], where classical DRE methods fail. We demonstrate that the integral $\int_ 0^1 \partial_ t \log p_ t(x)dt$ can be computed **in closed form**, yielding $\log \hat{r}(x) = \sum_ {k=1}^K h_ k(x) \overline{g}_ k$ in a single forward pass. This is achieved not because the model is normalized, but because the integration is tractable by construction.
> > >
> > > In this sense, **OS-DRE shows that numerical solvers are not inevitable in path-integral DRE.**
> > >
> > > We greatly appreciate your pointing us to this connection. Accordingly, we have (`highlighted in blue in the revised manuscript`):
> > > - `Revised the contribution statement in the Introduction` to specify that OS-DRE is the first *path-integral-based* method to achieve NFE=1 via an analytic solution;
> > > - `Added a dedicated discussion of [1] in Related Work`;
> > > - `Expanded the Limitations and Future Works` section to explore hybrid objectives that combine analytic integration with energy consistency  [1] and conditional score matching [1, 2].
> > >
> > > We hope this clarifies that our method offers a **complementary pathway**—one that preserves the generality of continuous path modeling while achieving analytic efficiency.

---

> ### Author Response · Authors · 2025-12-02
>
> Thank you for your thoughtful comment. We appreciate the opportunity to clarify this important distinction.
>
> **While energy-based models like [1] enable single step evaluation of normalized log densities, their application to general density ratio estimation (DRE) between two non Gaussian distributions is fundamentally limited by an unresolvable constant offset between separately trained energy models.**
>
> In our implementation of the dual score matching framework from [1], we observed precisely this issue in practice. Although the method excels at density estimation—producing calibrated, normalized energies when one distribution (e.g., $p_0$) is a known Gaussian prior—it fails to yield accurate density ratios when both $p_0$ and $p_1$ are non Gaussian, and lack a shared reference measure (as in mutual information estimation). The core problem is that even if each energy model is individually normalized, their global constants cannot be aligned without a common asymptotic anchor (e.g., a large noise Gaussian limit for both), leading to biased log ratio estimates despite correct per distribution modeling.
>
> In contrast, our one-step score-based model estimates the density ratio through closed-form path integration of time scores, which inherently cancels out partition functions and works for non Gaussian distributions **without requiring calibration or a known reference**.
>
> Thus, while [1] represents a major advance in **normalized density estimation**, it does **not** solve the general **density ratio estimation** problem. Our claim—being the first to eliminate numerical solvers entirely in score based DRE—remains valid and addresses a distinct, more general setting.
>
> [1] Learning normalized image densities via dual score matching, Guth et al.

---

### Official Review · Reviewer_Syar · 2025-10-25

**Soundness:** 2
**Presentation:** 3
**Contribution:** 3
**Rating:** 4
**Confidence:** 4

**Summary:**

The goal is to estimate the ratio of two densities $p_0$ and $p_1$ from their samples. To do so, the authors use the time score identity

$\log \frac{p_1(x)}{p_0(x)} = \int_0^1 \partial_t \log p_t(x) dt$.

where the time score $\partial_t \log p_t(x)$ is first estimated, then integrated. While related literature focuses on the first part --- designing sample-efficient estimators of the time score [1, 2] --- the authors instead focus on the second part which is solving the integral with as few queries as possible to the time score because it is approximated using a neural network that is expensive to evaluate.

The authors propose a specific parameterization of the time score model

$\log p_t(x) \approx \sum_{k=1}^{K} h_k(x) g_k(t)$

that decouples the $x$ and $t$ variables and where the $g_k$ can be integrated in closed-form. This way, the time score can be integrated in one-step which is computationally efficient.

The authors show this leads to good performance.

[1] Choi et al. Density Ratio Estimation via Infinitesimal Classification. AISTATS 2022.

[2] Yu et al. Density Ratio Estimation with Conditional Probability Paths. ICML 2025.

**Strengths:**

Estimating the ratio of two densities $p_0$ and $p_1$ using intermediate densities $(p_t)_{t \in [0, 1]}$ has been gaining traction in recent years [1, 2, 3, 4]. While most works focus on the *statistical efficiency* of the time score estimator, the authors instead focus on the *computational efficiency* of its integration. This is an original idea. The text is clear. The experiments are diverse and encouraging.

[1] Rhodes et al. Telescoping Density-Ratio Estimation. NeurIPS 2020.

[2] Choi et al. Density Ratio Estimation via Infinitesimal Classification. AISTATS 2022.

[3] Yu et al. Density Ratio Estimation with Conditional Probability Paths. ICML 2025.

[4] Williams et al. High-Dimensional Differential Parameter Inference in Exponential Family using Time Score Matching. AISTATS 2025.

**Weaknesses:**

## Minor concerns

This is more of a comment, which the authors can disregard if they wish. Personally, I did not find Figure 1 helpful to understand the authors' method. Actually, I found the text very clear and clearer even than the figure.

## Main concerns

My main concerns on the evaluation procedure. The authors' main claim is that their method should be *faster*.

Specifically, the authors use the same time-score estimator as in prior work [1], so we do not expect a difference in estimation error a priori.
However, they employ a different integration scheme (closed-form) compared to previous works that used numerical integration or ODE solvers.
Hence, as a reader I would expect to see experiments in the main text that illustrate a reduced computational cost.

**Concern 1: showing results on computational gain in the main text**. It would be nice to see results in the main text that quantify the computational gain. For example, the authors can plot the computational complexity on the Y axis (wall clock time or NFE) as a function of the approximation error of the integral chosen by the user on the X axis. We would expect an increasing function for other methods, and a near-zero constant function for the authors' method.

**Concern 2: distinguishing between estimation error and integration error**. The large differences between methods in Figure 2 is a bit surprising to me. In my experience, such a difference is usually due to estimation error in the time-score model rather than from integration error. Yet, the authors use the same estimation procedure as DRE-∞, so one would not expect major differences in estimation accuracy, unless their particular parameterization introduces them. Can the authors comment on this?

**Concern 3: clarifying Figure 3**. My understanding is that Figure 3 illustrate the ratio of two Gaussian densities. So in Figure 3.b., while the authors write "CIFAR-10-C-Style Corruption", they are not actually using CIFAR images. If my understanding is correct, then mentioning CIFAR is quite misleading and I would ask that the authors remove it. More generally, Figure 3 lacks quite a bit of explanation (I had to go through the appendix to understand it better). For example, how do you compute the KL in Figure 3c?

**Questions:**

I overall believe this could be a strong paper. If the authors address my three concerns detailed in the "Weaknesses" section, I would be happy to raise my score.

---

> ### Author Response · Authors · 2025-11-19
>
> Thank you for your clear and thoughtful review. We appreciate your recognition of our focus on computational efficiency in integration. We respond to your comments below.
>
> > C1: ...see results that quantify the computational gain… plotting computational complexity (wall clock or NFE) vs approximation error.
>
> Thank you for this excellent suggestion. We have added the requested efficiency–accuracy trade-off analysis to the revised manuscript.
>
> - **`New Fig. 4` shows MSE/NLL vs. NFE, exactly as recommended.** Accompanying this, we added a dedicated paragraph titled `Accuracy vs. Efficiency` (highlighted in blue). The results confirm a consistent trend: baseline methods require NFE = 2–50 to achieve reasonable accuracy, while OS-DRE matches or exceeds their performance at NFE = 1. The pattern is consistent across both density estimation and MI estimation tasks, confirming the computational advantage of our analytic integration.
>
> - **Wall-clock times are now reported in `Tabs. 1, 4, 7`.** Across all settings, OS-DRE achieves comparable or better accuracy with fewer function evaluations and lower runtime, demonstrating its practical efficiency.
>
> Partial results (see `Tabs. 8-9 for details, highlighted in blue`) are summarized below:
>
> |Dataset|Method|NFE=1|NFE=2|5|10|50|
> |-|-|-|-|-|-|-|
> |BSDS300|DRE-∞|-|-130.68 ± 4.17|-83.70 ± 1.35|-87.65 ± 2.24|-90.24 ± 2.14|
> ||D³RE|-|-149.53 ± 9.06|-101.97 ± 1.67|-102.01 ± 2.43|-78.26 ± 0.96|
> ||**OS-DRE**|**-217.99 ± 3.39**|-|-|-|-|
>
> |MI|Method|NFE=1|NFE=2|5|10|50|
> |-|-|-|-|-|-|-|
> |40|DRE-∞|-|1215.69|70.36|7.08|3.77|
> ||D³RE|-|500.04|76.80|59.70|58.19|
> ||**OS-DRE**|**2.30**|-|-|-|-|
>
> > C2: Large differences in Fig. 2 are surprising… differences usually come from estimation error, not integration error. Since estimation follows DRE-∞, why such a gap?
>
> You are absolutely right that estimation error (e.g., from learning $s_ t^ {\theta}$ ) plays a role, especially since we share the same STSM loss as DRE-∞ . However, in Fig. 2, the dominant factor is integration error, not estimation error. Here’s why:
>
> - **Baselines suffer large integration error at NFE = 2.** The baseline methods approximate the integral $\int_ 0^1 s_ t^\theta(x,t)dt$ using only two evaluations (via the trapezoidal rule at t=0,1). This coarse discretization introduces significant integration error, especially when $s_ t^\theta(x,t)$ varies non-linearly in t, which is precisely what causes the distortions observed in Fig. 2. `We have updated the caption of Fig.  2 to clarify this setting (highlighted in blue)`.
> - **OS-DRE incurs zero integration error by design.** Our method leverages an analytic integration framework (the Analytic Frame). Thus, no numerical quadrature is used, and the integration step is exact. Hence NFE = 1 with zero integration error.
> - **Our decoupled parameterization reduces estimation error.** While we use the same STSM training objective, our model architecture is fundamentally different: (1) Baselines model $s_ t^\theta$ as a joint function of $(x,t)$ , requiring re-evaluation at each t. (2) OS-DRE uses a time-decoupled parameterization. A single neural network  $[h_ 1^\theta(x), ..., h_ K^\theta(x)]$ predicts all spatial coefficients once, while temporal dynamics are fully captured by the fixed analytic bases  $\{g_ k(t)\}$. Furthermore, as noted in Sec. 4 (lines 302–306), our formulation also avoids second-order gradients by analytically computing $\partial_ t s_ t^\theta$, further stabilizing training and reducing estimation error.
>
> > C3: Calling it CIFAR-10-C-style is misleading… Fig. 3 lacks explanation… How is the KL in Fig. 3c computed?
>
> We appreciate the chance to clarify the experimental setup and naming.
> - **The figure title is now `Progressive Noise Corruption`**. We removed the CIFAR-10-C–style phrasing and now title the experiment Progressive Noise Corruption to better reflect its synthetic, controlled nature.
> - **About "Fig. 3 lacks quite a bit of explanation"**: A detailed description of Fig. 3 (including its motivation, construction, and interpretation) has been moved from the appendix into the main text, placed directly adjacent to the figure (highlighted in blue).
> - **The KL divergence in Fig. 3c is computed analytically.** In this controlled setting, we define  $p_ 0=N(0,I_ d), p_ t=N(\mu_s,I_ d)$ with $\|\mu_ s\|=\sqrt{2s}$. For two Gaussians with identical covariance, the KL divergence admits a closed-form KL$(p_ t\|p_ 0)=\tfrac12\|\mu_s\|^2=s$. Thus, the GT  curve (black line in Fig. 3c) is simply KL=s, increasing stepwise with corruption level s. We explain this derivation in App. C.4 and, in the `main text (Continual Learning, blue highlight)`, clarify how the KL is estimated from the learned density ratio.
>
> > Minor: ...not find Fig. 1 helpful to understand ... method.
>
> Thank you for the suggestion! We’ve updated Fig.1 to clearly distinguish OS-DRE’s single forward pass from baselines’ multistep evaluations.

---

> > ### Comment · Reviewer_Syar · 2025-11-24
> > **Response to authors**
> >
> > Thanks, this addresses my main concerns.
> >
> > **Accuracy vs. Efficiency paragraph and Figure 4 should be further clarified**. Thank you for these additions: they make your point much clearer to me. I think you can still clarify a few things. There are a lot of acronyms (MSE, NLL, NFE): please make sure they are defined somewhere. Also, "Accuracy vs. Efficiency" is way too vague: these terms can mean anything. The paragraph can be made much more precise by replacing "Efficiency" with "Computational Efficiency (NFE)", and by replacing "Accuracy" with "Error of the density ratio (NLL / MSE)". You should also clarify that you're making the assumption that the **error of the density ratio is dominated by the error in numerical integration**, instead of the error in time score estimation. This is, as I understand it, the main argument of your paper.
> >
> > **Figure 2**. The new Figure 2 is much clearer. We now see that if we increase the NFEs, the different methods can reach comparable performance. So it is in the low-NFE regime where the authors' method does best.
> >
> > **Figure 3**. This looks clearer now.
> >
> > I have updated my score.

---

> > > ### Author Response · Authors · 2025-11-25
> > >
> > > Thank you for your detailed and constructive feedback. We’re glad our earlier clarifications addressed your concerns, and we sincerely appreciate your updated assessment.
> > >
> > > We have implemented all your suggestions, as they significantly improve the clarity and rigor of our presentation. All new changes addressing your latest points are `highlighted in red`, while earlier revisions remain in blue:
> > >
> > > > There are a lot of acronyms (MSE, NLL, NFE): please make sure they are defined somewhere.
> > >
> > > We have ensured that all acronyms (MSE, NLL, NFE) are clearly defined upon first use in the main text. Additionally,  full metric names have been `added to the axes and captions of Fig. 4 (highlighted in red)`.
> > >
> > > > "Accuracy vs. Efficiency" is way too vague: these terms can mean anything. The paragraph can be made much more precise by replacing "Efficiency" with "Computational Efficiency (NFE)", and by replacing "Accuracy" with "Error of the density ratio (NLL / MSE)".
> > >
> > > We have renamed the paragraph to `Error of the Density Ratio (NLL / MSE) vs. Computational Cost (NFE)` and updated the corresponding explanation  in `the second sentence of this paragraph` (highlighted in red).
> > >
> > > > Clarify that the error is dominated by numerical integration...
> > >
> > > We have `added explicit statements` in
> > >
> > > - the first sentence of Abstract,
> > > - the second paragraph of Introduction,
> > > - the first sentence of paragraph "Error of the Density Ratio (NLL / MSE) vs. Computational Cost (NFE)",
> > >
> > > to clarify that the dominant error source in score-based DRE stems from numerical integration of the time score.
> > >
> > > In fact, our experiments confirm this trend. By replacing numerical solvers with a closed-form solution, OS-DRE eliminates this bottleneck, as confirmed by the empirical trends in Fig. 4.
> > >
> > > We hope these revisions meet your expectations. Thank you again for your insightful suggestions!

---

### Official Review · Reviewer_Lm7P · 2025-10-30

**Soundness:** 4
**Presentation:** 4
**Contribution:** 4
**Rating:** 10
**Confidence:** 3

**Summary:**

The authors propose a way of learning density ratio estimates with score-based algorithms without needing to run a ODE solver. Instead, they consider an analytic expression for bridging the densities. This is done using a sequence of RBF approximations, with theoretical analysis of the convergence rate. Good empirical performance is demonstrated in a range of benchmarks.

**Strengths:**

The paper makes a foundational contribution, resolving a major limitation of score-based DRE methods. This is an important family of models and removing the need for ODE solvers is a very clear contribution that opens up opportunities for both more accurate and more efficient methods. The authors provide sufficient theoretical guarantees and demonstrate the method well, which clear improvement over previous DRE methods in NLL and other measures.

I appreciate the effort of validating multiple kernel choices, and the broad evaluation in general, covering also complementary aspects like continual learning.

**Weaknesses:**

No notable weaknesses, but I acknowledge that I have not checked the theoretical proofs in detail and would likely have missed possible technical inaccuracies.

One thing that could be more clearly communicated is computational efficiency. You seem to quantify the efficiency only in terms of the function evaluations (NFE) yet make claims about drastic speed improvement. It would be good to somehow quantify this also in terms of wall-clock speed. Also, in Tables 1 and 3 you consider fairly scarce choice of NFE for the comparison methods -- why not plot the accuracy as a function of NFE, to make it clear what kind of NFE (if any) would be sufficient for reaching similar accuracy with methods based on ODE solvers?

**Questions:**

Fig 2 shows nicely how DRE with NFE=2 is blurry. How would DRE with large NFE look like? It would be better to show this, as it is not hurting your contribution in any way.

---

> ### Author Response · Authors · 2025-11-19
>
> Thank you for your encouraging feedback and for recognizing the value of removing ODE solvers from score-based DRE. We address your comments below.
>
> > Weakness: It would be good to somehow quantify this also in terms of wall-clock speed. Also, in Tables 1 and 3 you consider fairly scarce choice of NFE for the comparison methods -- why not plot the accuracy as a function of NFE, to make it clear what kind of NFE (if any) would be sufficient for reaching similar accuracy with methods based on ODE solvers?
>
> We thank the reviewer for this valuable suggestion. We agree that quantifying the practical speedup and mapping the full efficiency-accuracy landscape is crucial. We have incorporated these analyses into the revised manuscript:
>
> - **Wall-clock speed quantification:** We have updated `Tabs. 1 and 4 (main text) and Tab. 7 (Appendix)` to report actual wall-clock times alongside NFE. The results confirm that OS-DRE's theoretical advantage (NFE=1) translates directly into significant wall-clock speedups compared to solver-based methods.
> - **Accuracy vs. NFE plot (`New Fig. 4`):** We conducted a dense sweep of NFE values (NFE$\in \{2, 5, 10, 50\}$) for the baseline methods. We added a `new Fig. 4` and a dedicated paragraph `Accuracy vs. Efficiency` to visualize this trade-off (highlighted in blue).
> - **What NFE is sufficient?** The plot and extended tables (summarized below) reveal that baseline methods often require **NFE $\ge$ 50** to approach reasonable accuracy. Crucially, in challenging high-discrepancy settings (e.g., BSDS300 in density estimation or High-MI estimation), baselines fail to match OS-DRE's accuracy *even at NFE = 50*, **highlighting that our method's advantage is not just speed, but also the stability provided by the analytic framework**.
>
> Summary of extended results (Accuracy vs. NFE, see `Tabs. 8-9` for details,  highlighted in blue):
> |Dataset|Method|NFE=1|NFE=2|5|10|50|
> |-|-|-|-|-|-|-|
> |GAS|DRE-∞|-|-4.37 ± 1.44|-3.63 ± 0.78|-4.34 ± 0.60|-4.33 ± 0.71|
> ||D³RE|-|5.74 ± 15.28|-1.15 ± 4.20|-3.27 ± 2.00|-3.16 ± 0.62|
> ||**OS-DRE**|**-18.33 ± 0.04**|-|-|-|-|
> |MINIBOONE|DRE-∞|-|41.55 ± 2.07|20.90 ± 0.84|20.57 ± 0.93|20.97 ± 0.51|
> ||D³RE|-|55.83 ± 9.36|43.11 ± 26.20|42.65 ± 26.87|42.73 ± 26.78|
> ||**OS-DRE**|**9.97 ± 0.37**|-|-|-|-|
> |BSDS300|DRE-∞|-|-130.68 ± 4.17|-83.70 ± 1.35|-87.65 ± 2.24|-90.24 ± 2.14|
> ||D³RE|-|-149.53 ± 9.06|-101.97 ± 1.67|-102.01 ± 2.43|-78.26 ± 0.96|
> ||**OS-DRE**|**-217.99 ± 3.39**|-|-|-|-|
>
> |MI|Method|NFE=1|NFE=2|5|10|50|
> |-|-|-|-|-|-|-|
> |10|DRE-∞|-|73.91|2.86|0.27|0.03|
> ||D³RE|-|2.58|**0.01**|0.02|**0.01**|
> ||**OS-DRE**|**0.01**|-|-|-|-|
> |20|DRE-∞|-|283.52|7.09|0.54|0.04|
> ||D³RE|-|3.65|0.29|0.21|0.09|
> ||**OS-DRE**|**0.00**|-|-|-|-|
> |30|DRE-∞|-|614.62|25.29|2.66|0.48|
> ||D³RE|-|6.21|7.50|7.72|8.94|
> ||**OS-DRE**|**0.07**|-|-|-|-|
> |40|DRE-∞|-|1215.69|70.36|7.08|3.77|
> ||D³RE|-|500.04|76.80|59.70|58.19|
> ||**OS-DRE**|**2.30**|-|-|-|-|
>
>
>
>
> > Question: Fig. 2 shows nicely how DRE with NFE=2 is blurry. How would DRE with large NFE look like? It would be better to show this, as it is not hurting your contribution in any way.
>
> Thank you for your suggestion. We agree that showing the performance of the baseline methods under a higher computational budget provides a clearer context for our contribution.
>
> We have incorporated the requested results into the revised manuscript:
>
> - **Main text Figure 2 update:** We have updated `Figure 2` in the main paper to include the results for **DRE-$\infty$ and D³RE at NFE=10**. The visual comparison clearly shows that while accuracy improves over NFE=2, the baseline methods still struggle to capture the fine-grained structures (like `checkerboard` or `tree`) compared to our OS-DRE at NFE=1.
> - **Appendix Expansion:** We have revised our figure in the Appendix (`Figure 6`), which provides a more comprehensive set of density estimates comparing the baseline methods at NFE=$2, 5, 10$.

---

> > ### Comment · Reviewer_Lm7P · 2025-11-24
> > **Reply**
> >
> > > We conducted a dense sweep of NFE values (NFE$\in {2, 5, 10, 50}$) for the baseline methods.
> >
> > The figure showing the new results is really helpful, and I was also happy to see that the promised speedup is easily seen in practice as well.
> >
> > > The plot and extended tables (summarized below) reveal that baseline methods often require NFE $\ge$ 50 to approach reasonable accuracy. Crucially, in challenging high-discrepancy settings (e.g., BSDS300 in density estimation or High-MI estimation), baselines fail to match OS-DRE's accuracy even at NFE = 50
> >
> > This is a nice extra result. Looking at the results, it seems the two baselines behave differently:
> > - For DRE-$\infty$ the accuracy improves with NFE, and it might be useful to still check NFE=100 or even 200 to see whether it eventually works as well as yours. (No need to do that now during the revision as it makes no difference for the evaluation, just pointing this out as a possibility)
> > - For D3RE the benefit of extra evaluations beyond NFE=5 appears small across the cases. Any explanation for this?

---

> > > ### Author Response · Authors · 2025-11-25
> > >
> > > Thank you for your valuable follow-up and for recognizing the clarity of our NFE analysis. We respond to your comments below.
> > >
> > > > Q1: For DRE-∞ the accuracy improves with NFE, and it might be useful to still check NFE=100 or even 200 to see whether it eventually works as well as yours.
> > >
> > > We have now added results for NFE = 100 and 200 in the MI estimation task (full results in `Tab.  7 and Tab. 9`, highlighted in red; visualization in `Fig.  4(b)`):
> > >
> > > | MI | Method | NFE=1 | 2 | 5 | 10 | 50 | 100 | 200 |
> > > |----|--------|-------|---|---|----|----|-----|-----|
> > > | 10 | DRE-∞ | – | 73.91 | 2.86 | 0.27 | 0.03 | 0.02 | 0.02 |
> > > |    | D³RE  | – | 2.58 | 0.01 | 0.02 | 0.01 | **0.00** | **0.00** |
> > > |    | **OS-DRE** | 0.01 | – | – | – | – | – | – |
> > > | 20 | DRE-∞ | – | 283.52 | 7.09 | 0.54 | 0.04 | **0.01** | **0.01** |
> > > |    | D³RE  | – | 3.65 | 0.29 | 0.21 | 0.09 | 0.08 | 0.05 |
> > > |    | **OS-DRE** | **0.00** | – | – | – | – | – | – |
> > > | 30 | DRE-∞ | – | 614.62 | 25.29 | 2.66 | 0.48 | 0.50 | 0.50 |
> > > |    | D³RE  | – | 6.21 | 7.50 | 7.72 | 8.94 | 9.00 | 9.00 |
> > > |    | **OS-DRE** | **0.07** | – | – | – | – | – | – |
> > > | 40 | DRE-∞ | – | 1215.69 | 70.36 | 7.08 | 3.77 | 3.31 | 3.31 |
> > > |    | D³RE  | – | 500.04 | 76.80 | 59.70 | 58.19 | 58.20 | 57.28 |
> > > |    | **OS-DRE** | **2.30** | – | – | – | – | – | – |
> > >
> > > As you anticipated, DRE-∞ continues to improve with higher NFE and eventually matches OS-DRE at low discrepancy (MI = 10, 20). However, **at high discrepancy (MI = 30, 40), even NFE = 200 is insufficient**. DRE-∞ saturates at MSE ≈ 0.50 and 3.31, still far worse than OS-DRE’s 0.07 and 2.30 (achieved with NFE = 1). This confirms that once numerical integration error is minimized, the remaining bottleneck is the inherent time score estimation error of the network.
> > >
> > > > Q2: For D3RE the benefit of extra evaluations beyond NFE=5 appears small across the cases. Any explanation for this?
> > >
> > > For D³RE, its performance **saturates early** (often by NFE = 10). This is consistent with its **SDE-based formulation** (stochastic differential equation), where the dominant error source is the time score estimation error (i.e., the neural network’s approximation error), **not** numerical integration error. Thus, increasing NFE beyond a small value yields diminishing returns, as seen in MI = 30/40, where MSE remains ≈ 9.0 / 57.3 from NFE = 50 to 200. **For DRE-∞, ODE-based formulation is used,** which is deterministic. (ODE: ordinary differential equation)
> > >
> > > > Finally, ...
> > >
> > > We are currently running the NFE=100/200 experiments for density estimation and will include the results and related discussion in the final version within the next few days.
> > >
> > > We hope these clarifications are helpful and address your comments satisfactorily.Thank you again for your helpful  suggestions!

---

### Official Review · Reviewer_LTmZ · 2025-10-31

**Soundness:** 3
**Presentation:** 3
**Contribution:** 2
**Rating:** 6
**Confidence:** 2

**Summary:**

This paper studies the problem of computing $\log r(x)$, where $r(x)$ is a density ratio. The core goal of this paper is to perform density ratio estimation. Density ratio is a fundamental task in ML and statistics and is used to quantify discrepancies between two probability distributions. Recent works on DRE using continuous score-based methods proposed that instead of computing the ratio between $p_0$ and $p_1$, one can compute the path integral of the $\delta_t \log p_t(x)$ along time as a measure for $log$-density-ratio. However, computing this integral can be expensive. In this paper, the authors propose a one-step score-based density ratio estimation which computes an approximation to the path integral by first constructing a separable basis expansion of $\delta_t \log p_t(x)$ and then approximating using a finite sum. Note the finite sum corresponds to the coefficients of the basis function of $x$ and one would still need to integrate the basis due to $t$. However, the authors correctly observe that for standard orthonormal bases (e.g., Fourier, Legendre), all basis elements except the constant function have zero integral over $[0,1]$. To get around computing a vacuous solution, the authors relax the strict orthogonality condition for the bases of $t$, and instead use frames (discretization) of the function of $t$. Then they evaluate score as a double over this discretization and the coefficients due to $x$. Given the discretization leads to locally continuous functions in $t$, one can efficiently compute $\log r(x)$. The authors further show that these bases can be precomputed given we can assume that the function of $t$ can be expressed using specific variants of the RBF kernel. One can then learn the parameters of the function in $x$ using a simple neural network. Using multiple experiments on synthetic and real world data, the paper demonstrates that this procedure leads to significant improvements when estimating DRE.

**Strengths:**

- This work correctly identifies that estimation of an integral over dual variable can be expressed effectively using basis expansion and then uses the properties of specific functions to approximate this basis expansion effectively. This results in an analytic solution which is nice! Although I have not gone through all the main proofs minutely, I believe the theoretical claims made in this paper.

- Reducing the core task of multiple function evaluations to a single forward pass in a neural network is also significant as it eliminates the need for expensive numerical solvers or iterative integration.

- The study of $g_k(t)$ using RBF functions make the algorithm much more "user friendly", and applicable to a wide domain of applications (as RBF is often the main choice of kernel functions).

- Beyond theoretical contributions, the authors demonstrate the superior performance of their method empirically on several synthetic and real world datasets.

**Weaknesses:**

- I am a bit confused about the general applicability of the class of the methods in general. Observe that the primary assumption of these algorithms is that there is an integrable path from $p_0(x)$ to $p_1(x)$. Is that generally true for applications in the wild? For high-dimensional and multimodal data, this assumption might fall apart, limiting the scope of application of such methods. I don't understand that claim that these methods alleviate "density-chasm". E.g., when there is minimal overlap of the supports, how do you even compute its $log$? I think it becomes close to undefined. Wouldn't using measures like Wasserstein be more effective then for resolving the "density-chasm" problem?

- While indeed the method works (demonstratively) when the target function belongs to a Sobolev space of lower smoothness, I am unsure, if it is even efficient for other function classes?

- For Proposition 4.1 a core assumption is denseness. So what happens in case of long tailed distributions?

- There are two stages of approximation -- 1) basis expansion is approximated using $K$ terms, and 2) the integral of $g_k(t)$ in $[0,1]$ is approximated using discretization (infinitely many discrete frames). But then when you write Equation (8) you write it as an equality without the coefficients $c$?

**Questions:**

Please see the limitations section.

---

> ### Author Response · Authors · 2025-11-19
> **Response to Reviewer LTmZ, 1/2**
>
> We sincerely thank the reviewer for the clear summary and for highlighting the strengths of our work. Below we provide point-by-point responses to the reviewer’s questions and concerns.
>
> > W1: The primary assumption ... is that there is an integrable path from $p_0(x)$ to $p_1(x)$. Is that generally true for applications in the wild? For high-dimensional and multimodal data, this assumption might fall apart ... these methods alleviate ‘density-chasm’ ... when there is minimal overlap of the supports ... how do you even compute its log? Wouldn't using measures like Wasserstein be more effective ... ?
>
> Thank you for your valuable questions and suggestions. We clarify three key points:
>
> - **The interpolating path $p _ t$ is a constructed surrogate, not a requirement from real data.** Score-based DRE methods (including [1,2,3,4] and ours) define the density ratio via $\log r(x)=\int_ 0^ 1 \partial_ t \log p_ t(x)dt$, where $p_t$ is an *artificially chosen* path (e.g., linear: $x _ t=(1−t)x_ 0+tx_ 1$, or diffusion-based). This path need not correspond to any real-world process.
> - **This formulation avoids density-chasm because we do not need to evaluate $\log(p_1/p_0)$ directly.** Classical DRE fail when $p_0(x)\to 0$ (causing $\log \frac{p_1(x)}{p_0(x)}$ to diverge) . In contrast,  $\partial_t \log p_t(x)$ is well-defined *locally* along the constructed path. Moreover, Gaussian dequantization [2,4] and diffusion bridge [3,4] ensure $p_t(x)>0$ for all $t\in[0,1]$ and $x$, guaranteeing $\log p_t(x)$ is finite.
> - **Wasserstein distance serves a different purpose.** While Wasserstein is effective for measuring *global* discrepancy between non-overlapping distributions, it may not provide the *pointwise* ratio $r(x)=p_1(x)/p_0(x)$, which is essential for tasks such as $f$-divergence estimation and other downstream tasks. Thank you again for your insightful suggestions.
>
> > W2: While indeed the method works ... when the target function belongs to a Sobolev space of lower smoothness, I am unsure if it is even efficient for other function classes?
>
> This is a great question. Our method only requires the target function to lie in a Hilbert space. By Proposition 3.1, the integrand $\partial_t \log p_t(x)$ belongs to $L^2$, which is a Hilbert space. Consequently, our approach applies to any time score function, **without requiring Sobolev smoothness**. In particular, no additional regularity assumptions (e.g., Sobolev-type smoothness) are needed, and the method remains well-defined and theoretically grounded for general $L^2$ targets.
>
> > W3: For Pro. 4.1, a core assumption is denseness. So what happens in case of long-tailed distributions?
>
> There seems to be a small misunderstanding here, and we're happy to clarify. The denseness assumption in Pro. 4.1 pertains **only to the time-domain function space**, not to the data distribution (e.g., whether $p_1(x)$ is long-tailed). Specifically:
>
> - **The assumption concerns approximation in time, not in space.** Proposition 4.1 requires that the span of $\{g_k(t)\}$ is dense in $H_ t = L^ 2([0,1])$. This ensures that the temporal evolution of  $\partial_t\log p_t(x)$ can be accurately represented, regardless of the tail behavior of $p_ 1(x)$.
> - **Long-tailed or heavy-tailed data pose no issue for this assumption.** Tail properties of $p_ 1(x)$ relate to the spatial variable $x$, whereas denseness in $H_ t$ is purely about approximating functions of time $t$. The two are orthogonal. In fact, our experiments (e.g., on the Half-Cube Map, which exhibits heavy tails) show that OS-DRE remains stable and accurate. Kernels such as IMQ or RQ, which are well-suited for long-range dependencies, further enhance robustness in such settings.
>
> [1] Rhodes et al. Telescoping Density-Ratio Estimation. NeurIPS 2020.
>
> [2] Choi et al. Density Ratio Estimation via Infinitesimal Classification. AISTATS 2022.
>
> [3] Yu et al. Density Ratio Estimation with Conditional Probability Paths. ICML 2025.
>
> [4] Chen, Wei, et al. Dequantified Diffusion-Schrödinger Bridge for Density Ratio Estimation.  ICML 2025.

---

> ### Author Response · Authors · 2025-11-19
> **Response to Reviewer LTmZ, 2/2**
>
> > W4: ...two stages of approximation (1) basis expansion approximated using $K$ terms, (2) ...integral of $g_k(t)$ is approximated using discretization (infinitely many discrete frames). But in Eq. (8) you write it as an equality without the coefficients $c$?
>
> We appreciate the opportunity to clarify a key aspect of our method. There are indeed two potential sources of approximation, but **only one is actually used** in our framework:
>
> - **Truncation to $K$ basis functions is the sole approximation.** We approximate the infinite series in the time domain by keeping the first $K$ terms.
> - **The time integrals $\int_ 0^1 g_k(t)dt$ are computed `analytically`, not via discretization.** We choose RBF-type temporal bases (e.g., Gaussian) precisely because their integrals over [0,1] admit **closed-form expressions** (see App. B, Eqs. (12,13)). These integrals, denoted $\bar g_k$, are precomputed constants. This is why our method achieves **true NFE = 1** at inference.
> - **Regarding Eq. (8):** The spatial coefficients $c_ {l,k}$ are absorbed into the definition of the learnable  $h_ k(x) \triangleq \sum_{l=1}^{\infty} c_{l,k} f_l(x)$. Thus, Eq. (8) correctly expresses the density ratio as an exact (up to *K* -term truncation) analytical integral—no discretization error is introduced.

---

> > ### Comment · Reviewer_LTmZ · 2025-11-28
> > **Response to rebuttal**
> >
> > Thank you for your response and clarifications. I am still a little bit confused about the use of the interpolating path as the primary objective being solved in this paper. I understand that it has been demonstrated to work for this problem in prior work and this, but the whole point of the methods presented in this work is to get around $\log(p_1/p_0)$, so any data for which such a path may not exist, the problem setting become vacuous (also objective). It will be great if the authors can provide some counter arguments or discussions around these.
> >
> > Apart from this all my other concerns are adequately responded to and so I am willing to increase my score. I would request the authors to add some of these discussions to the main text to help in better understanding for the reader.
> >
> > Thanks!

---

> ### Author Response · Authors · 2025-11-29
>
> Thank you for your positive assessment and for deciding to increase your score. We are grateful for your thoughtful engagement and have fully addressed your final concern regarding path existence.
>
> Specifically, you questioned whether the method is vacuous if no "natural" path exists between $p_0$ and $p_1$. We clarify that **the path is not assumed but explicitly constructed**. In score-based DRE, we define a smooth interpolating path (e.g., linear path, $x_ t = (1-t)x_0 + t x_1$) that guarantees:
>
> - $p_t$ is a valid density for all $t \in [0,1]$,
> - the time score $\partial_ t \log p_t(x)$ is well-defined almost everywhere.
>
> Thus, the path integral $\int_ 0^1 \partial_ t \log p_t(x) dt$ always exists by construction, and the method is never vacuous.
>
> **To improve accessibility for a broader audience, we have added** (`highlight in blue`):
>
> - `New Figure 5`, visualizing the path from a Gaussian to a checkerboard dataset,
> - `a dedicated discussion in Appendix C.1.1`, explaining how such paths are constructed and why they ensure theoretical well-posedness.
>
> We believe the revisions have made the manuscript more clearer and more approachable for a broader audience.

---

### Official Review · Reviewer_RMmF · 2025-11-01

**Soundness:** 3
**Presentation:** 4
**Contribution:** 3
**Rating:** 8
**Confidence:** 4

**Summary:**

The paper contributes a new algorithm for continuous score-based density ratio estimation (DRE) that is computationally much more efficient than previous algorithms in this category at comparable or better accuracy.

Continuous score-based DRE methods avoid the "density chasm" problem but suffer from the computational cost of numerical integration of the predicted time score (= partial derivative of log density ratio w.r.t time) along the path (in time) from one distribution to the other. This paper instead proposes to approximate the time score by a finite-term expansion of the form $s_t^{\theta}(x, t) = \sum_{k=1}^{k=K} h_k^{\theta}(x)g_k(t)$, where $\\{g_k(t)\\}$ have closed form expressions for their time integral. The idea is that the time score prediction model parameters $\theta$ can be trained to predict the $K$ functions $\\{h_k^\theta(x)\\}_{k=1}^{k=K}$ instead of $s^\theta()$. Since $\\{g_k(t)\\}$ are analytically integrable, their method avoids numerical integration in this fashion by just evaluating the model prediction only once.

The expansion for time score is analogous to expansions over Hilbert space of space x time functions, such as Karhunen-Loeve, which separates out space and time components, but with a few differences. For their method to work, they need the time integrals for $\\{g_k(t)\\}$ to not vanish (usually the case for may common choices, such as Fourier expansion). At the same time, their application does not depend on orthogonality of $\\{g_k(t)\\}$. This allows them to choose so called "frames" of Hilbert spaces for the time component (i.e., the space of functions of time on domain $[0, 1]$) for these time functions. In particular, they go with RBFs over time, and show that RBFs (under mild conditions i.e. strictly positive definite, e.g. Gaussian) qualify as frames, and that these RBFs can approximate well if the true time score and these RBFs satisfy some technical conditions---the approximation error diminishes as $O(K^{-\beta})$ where $\beta$ parametrizes the Sobolev space to which the time score is assumed to belong.

The main technical contribution of the paper are the finite-term expansion of time score in terms of frames over time, and the follow-up approximation result with RBFs as frames, with rigorous proofs for all claims. They also provide a section at the end that compares their algorithm to previous continuous time score DRE methods empirically.

**Strengths:**

The paper makes a novel, rigorous, and well-substantiated contribution to the subject of DRE. Major strengths of the paper:

- Novelty: Separating out space and time components in a frame-based expansion where the time functions have closed form integrals, thereby giving a compuationally efficient approximation, is a simple but new and effective idea.

- Rigour: The key technical insight this paper provides is perhaps that continuous time score functions (under some technical assumptions) can be well approximated by a finite-term expansion in terms of RBF over time. The paper and its appendix provides rigorous proofs of each step leading to this result (Theorem 3.4, Propositions 4.1, and 4.2).

- Experimental validation: The paper evaluates their novel method and compares it to previous continuous score-based DRE methods for a wide variety of datasets, real world as well as synthetic.

- Presentation: The paper is written clearly, with sufficient background on previous work and motivation, and with brief explanations while deferring the detailed proofs to the appendix, all of which ensures smooth flow of exposition, while allowing the reader to verify the proofs separately.

**Weaknesses:**

No major flaws. A few comments however:
- It seems to me that the paper could do with a little more discussion on how the parameter $K$ (number of terms in the expansion) affects accuracy and computation cost. The paper does point out that large values of $K$ can lead to overfitting. But it seems to me that large values of $K$ may be necessary for fitting more complex time score functions in practice (the Sobolev space assumption notwithstanding).  Then the the time score prediction model (neural network) may need more parameters to fit a large number ($K$) of functions, and the cost of training and inference would also scale as a function of $K$ as a result.

- While the paper presents a number of experimental results for their method (OS-DRE) and earlier ones, it would have been good to  compare OS-DRE to earlier methods, notably DRE-$\infty$ (that they are perhaps closest to, in setup, losses etc.), on a real world dataset where DRE-$\infty$ was also tested in the paper that originally proposed the latter (Choi et al 2022). For example, this paper does compare DRE-$\infty$ and OS-DRE on real world datasets (from Grathwohl 2018), but these datasets were not used in the Choi et al 2022 paper as far as I can tell. Same for TRE.  etc. As someone not (previously) familiar with continuous score-based DRE literature, I perhaps could also have posed this in the "Questions" section of the review, but it would be good to have a clarification or update from the authors.

**Questions:**

None, except see also the 2nd comment in the "Weaknesses" section.

---

> ### Author Response · Authors · 2025-11-19
>
> We sincerely thank the reviewer for their positive and insightful feedback, particularly for recognizing the novelty, rigor, and efficiency of our method. We address your comments below.
>
> > Weakness 1: ...the paper could do with a little more discussion on how the parameter K (number of terms in the expansion) affects accuracy and computation cost. ...large values of K may be necessary for fitting more complex time score functions... cost of training and inference would also scale as a function of K...
>
> The choice of $K$ balances **model capacity and computational cost**, with accuracy improving up to a point before overfitting, while both training and inference scale linearly in $K$.
>
> - **Accuracy improves with $K$ up to a task-dependent optimum, beyond which overfitting occurs.**  As shown in our ablation study (`lines 475–481`), on tabular density estimation tasks, performance steadily increases as $K$ grows from 100 to 400, but begins to degrade at $K = 800$. This aligns with the approximation theory in Proposition 4.2 and confirms that $K$ acts as a key capacity-control hyperparameter that should be tuned based on data complexity.
> - **Computational cost scales linearly with $K$ in both training and inference.**  The main costs are: (a) the neural network’s final layer outputs a $K$-dimensional vector; (b) computing the time score $s_t^\theta(x)$ and its time derivative $\partial_t s_t^\theta(x)$ involves a $K$-term weighted sum (Eq. 14); and (c) the final log-density ratio estimate is a single inner product $\log \hat{r}(x) = \sum_{k=1}^K h_k^{\theta^*}(x) \overline{g}_k$, which is $O(K)$. Thus, both memory and FLOPs grow **linearly** with $K$, making the method scalable in practice.
>
>
>
>
>
> >  Weakness 2: While the paper presents a number of experimental results for their method (OS-DRE) and earlier ones, it would have been good to compare OS-DRE to earlier methods, notably DRE-∞ ..., on a real world dataset where DRE-∞ was also tested in the paper that originally proposed the latter (Choi et al 2022). ... As someone not (previously) familiar with continuous score-based DRE literature...
>
> We have now performed a direct comparison with DRE-∞ on MNIST, the key high-dimensional benchmark used in Choi et al. (2022), and confirm that OS-DRE achieves new state-of-the-art results with dramatically faster inference.
>
> - **Direct SOTA result on DRE-∞’s original benchmark (MNIST EBM).** We evaluated OS-DRE on the MNIST energy-based modeling task, exactly as in Choi et al. (2022). Our method achieves **1.278 BPD** with NFE = 1, outperforming DRE-∞ (1.302 BPD) and D³RE (1.281 BPD), both of which require NFE = 75 and ~21 seconds of inference time. OS-DRE completes the same task in **0.312 seconds**—a **68× speedup**—while being more accurate (`New Tab.2 and paragraph Energy-based Modeling on MNIST, highlighted in blue`).
>
> - **We built a unified, modular codebase that enables full reproducibility and extension.**  To ensure fair comparison, we developed a new `PyTorch Lightning`-based framework based on the official DRE-∞ and D³RE codebases. This package now supports **all experiments from both papers**, including those not originally reported (e.g., tabular density estimation and continual learning under distribution drift in our paper). We plan to release this clean, modular code to benefit the community.
>
> - **Our broader experiments demonstrate generalizability beyond MNIST.**  While MNIST addresses your request for direct comparison, our original tabular (GAS, MINIBOONE) and continual learning experiments show that OS-DRE’s advantages extend to diverse settings. The new codebase confirms that these results are consistent with the core DRE-∞ evaluation protocol.

---

### Author Response · Authors · 2025-12-01
**Review and Reviewer-Author Discussion Summary (1/2)**

Dear PCs, SACs, ACs, and Reviewers,

Thank you very much for your valuable contributions to our work. To assist the newly assigned AC and help reduce their workload, we provide below a summary of the key points from the reviews and the reviewer-author discussions.

----

**Strength.** Overall, we are grateful that all five reviewers gave this paper a highly positive evaluation in the initial reviews. Specifically:

- **This paper proposes a novel and computationally efficient framework (OS-DRE) for score-based density ratio estimation by separating spatial and temporal components via an analytic frame expansion, achieving only one-step evaluation.**

  All five reviewers recognized this point, highlighting it as a "attractive," "original," and "effective" idea that eliminates expensive numerical solvers at inference time (RMmF: Strength 1, LTmZ: Strengths 1 & 2, Lm7P: Strengths, Syar: Strengths, bFyf: Strength 1).

- **The proposed method is supported by rigorous theoretical analysis, including proofs for approximation bounds with RBFs.**

  Four reviewers explicitly highlighted the theoretical rigour and interest of the analysis (RMmF: Strength 2, LTmZ: Strength 1, Lm7P: Strengths, bFyf: Strength 2). Reviewer LTmZ additionally highlighted that the study of $g_k(t)$ using RBF functions makes the algorithm much more "user friendly" and applicable to a wide domain (LTmZ: Strength 3).

- **The experimental validation is comprehensive, demonstrating superior empirical performance on a wide variety of synthetic and real-world datasets compared to previous methods.**

  Four reviewers explicitly highlighted the extensive and encouraging experimental results (RMmF: Strength 3, LTmZ: Strength 4, Lm7P: Strengths, Syar: Strengths).

- **The paper is clearly written and well-presented.**

  Three reviewers explicitly highlighted the clarity of the writing and organization (RMmF: Strength 4, Lm7P: Strengths, Syar: Strengths).

---

> ### Author Response · Authors · 2025-12-01
> **Review and Reviewer-Author Discussion Summary (2/2)**
>
> **Concerns and Our Addressing.** During the discussion period, we actively addressed the reviewers' concerns, which were recognized by the reviewers and led to increased scores. Specifically:
>
> - **Concerns about experiment design.**
>
>   - (RMmF: Weakness 1) Requested more discussion on how the number of basis functions $K$ affects accuracy and cost.
>     **Our Addressing.** We clarified that $K$ controls model capacity and that computational cost scales linearly with $K$ (which is highly efficient compared to NFE scaling). We pointed to our ablation study showing performance improves with $K$ up to a point of overfitting.
>
>   - (Lm7P: Question, Syar: Main Concern 1) Reviewers requested a clearer quantification of computational gain (computational complexity vs. approximation error) and a clearer visualization of the trade-off against baselines using higher NFE.
>
>     **Our Addressing.** (1) We added wall-clock times to Tab. 1 and 4. We added a new Fig. 4 displaying a dense sweep of NFE for baselines, demonstrating they often require NFE $\ge$ 50 to approach OS-DRE's accuracy. (2)We added a paragraph clarifying the trade-off and defining precise metrics (NLL/MSE vs. NFE). Reviewer Lm7P found the new figure "really helpful" and reviewer Syar noted this addressed their main concerns.
>
>   - (bFyf: Weakness 1& Question, RMmF: Weakness 2) The experiments were perceived as smaller scale, and a direct comparison with DRE-$\infty$ on its original high-dimensional benchmarks (e.g., MNIST) was requested to verify behavior in complex settings.
>
>     **Our Addressing.** We added a new experiment on energy-based modeling on MNIST. OS-DRE achieved a new SOTA BPD of 1.278 with NFE=1, outperforming both DRE-$\infty$ and D$^3$RE while being $\sim68\times$ faster in wall-clock time. Reviewer bFyf acknowledged this contribution.
>
> - **Theoretical concerns regarding path existence, denseness, and related work.**
>
>   - (LTmZ: Weakness 1 & Follow-up) Concern that the problem setting becomes "vacuous" if an interpolating path between distributions does not exist in the real world.
>
>     **Our Addressing.** We clarified that the path is constructively defined (e.g., linear path). We added a new Fig. 5 and Appendix C.1.1 to visualize and explain this construction.
>
>   - (LTmZ: Weakness 2, 3, 4) Questions regarding efficiency outside Sobolev spaces, the meaning of "denseness" for long-tailed distributions, and the nature of the approximation in Eq. (8).
>
>     **Our Addressing.** We clarified that: (a) "Denseness" in Prop 4.1 refers to the temporal basis in the function space $L^2([0,1])$, not the data distribution (and our method works well on heavy-tailed tasks like Half-Cube Map); (b) Our integral is exact (analytic), not a discretization approximation; (c) Eq. (8) is mathematically consistent with our definitions.
>
>   - (bFyf: Weakness 2 & Follow-up) Questioned the claim of being the "first solver-free" method given concurrent work like Guth et al. (2025) and Chen et al. (2025)  which may achieve single-step evaluation.
>
>     **Our Addressing.** We implemented the framework of Guth et al. (2025) and found that although it recovers normalized densities when one endpoint is a known Gaussian, it fails for general DRE where both p0 and p1 are non-Gaussians. In contrast, our method estimates the log ratio directly via an analytic integral of the time score, requiring no calibration, no reference distribution, and no numerical solver. Thus, OS-DRE is the first truly solver-free method for  score-based DRE. We updated the contributions and expanded Related Work; see our responses to bFyf.
>
> - **Presentation and visualization issues.**
>
>   - (Syar: Minor concern & Main concern 3, Lm7P: Follow-up) Various points regarding the clarity of the original Figure 1, interpretation of large differences in Figure 2, misleading labels in Figure 3.
>
>     **Our Addressing.** We revised Fig. 1 for clarity. We clarified that the large differences in Fig. 2 stem from severe numerical integration errors in baselines at low NFE. We renamed the experiment in Fig. 3 and defined all acronyms as requested.
>
> **Recognition of our revision from reviewers.** Following our revisions and responses, Reviewer Syar explicitly raised the score (4->6). Reviewer LTmZ explicitly acknowledged that the main concerns were resolved and expressed willingness to raise the score. Reviewers RMmF, Lm7P, and bFyf did not raise further questions during the follow-up discussion period, and we believe that we have properly addressed their concerns.
>
> ------
>
> Above, we have faithfully summarized all reviewer comments and our corresponding responses, hoping that this will assist the AC's work. We are deeply grateful to the reviewers, AC, SAC, and PC, for their dedicated effort and excellent work. Their insightful feedback has further strengthened our paper. The authors offer their sincere respect and appreciation to all involved!
>
> Sincerely,
>
> Authors

---

### Meta-Review · Area_Chair_B1hK · 2026-01-17

**Summary:**

**Reviewer RMmF:** No major concerns raised. More thorough experimental comparison desired (W2).

**Reviewer LTmZ:** Questionable assumption on integrable path from $\log p_0$ to $\log p_1$ (W1). Efficiency in function classes other than Sobolev is not discussed (W2).

**Reviewer Lm7P:** No major concerns raised, but theoretical proofs were not checked in detail. Wall-clock speed is not shown.

**Reviewer Syar:** Computational gain not shown (W1). Unclear distinction between estimation and integration errors (W2). Insufficient description of Figure 3 (W3).

**Reviewer bFyf:** Experiments are of small scale (W1). Benefit beyond computational efficiency (W2).

**Additional points:**
- I have serious concern about the novelty of Lemma A.4. The statement is regarding the denseness, in $L^2([0,1])$, of the linear span of a location-scale family generated by a function $\phi$ satisfying certain conditions. This kind of problems has a long history, and consequently, there is now a quite large body of literature on it. See, for example, Ismailov, *Ridge Functions and Applications in Neural Networks*, Mathematical Surveys and Monographs, volume 263, American Mathematical Society, 2021, and references therein. There are several known denseness (density) results (many of them being even stronger than $L^2$ denseness), as well as results on approximation bounds. As this manuscript claims the denseness and approximation bound results as one of its main contribution (lines 77-78, 526-528), it should cite key papers in the relevant literature, and properly position the statements of the manuscript in the literature, clarifying what the original contributions are.
- Lines 146-149: There should be no problem as long as the coefficients $\{h_k(x)\}$ of the expansion (3) of $s_t$ is evaluated reasonably well, in which case one will have $\log r(x)=h_1(x)$. More concrete description on what the actual problem here is would be needed.
- Corollary 3.5: One would need to assume absolute convergence rather than mere convergence.
- Line 211: (s → S)ee
- References:
  - Albergo et al: This paper has been published in JMLR, so that the published version should be cited.
  - Gutmann and Hyvärinen: The journal title is not properly capitalized.
  - Karhunen: It seems that the paper title is incorrect. It should read "Über lineare Methoden in der Wahrscheinlichkeitsrechnung". The volume/pages seem to be 37, 1-79.
  - Letizia et al.: In (Advances in) Neural Information Processing Systems
  - Narcowich et al.: (b → B)ernstein
  - Vaswani et al.: The title of the proceedings is not properly capitalized.
  - Xiao et al.: rlhf → RLHF
  - Yu et al.: The conference name is not properly capitalized.
- Line 869: for $\forall s_t\in\mathcal{S}$(. I → , i)t satisfies:
- Lines 893-894: The condition $\int_{\mathcal{X}}p(x)\,dx=1$ itself does not necessary imply $\int_{\mathcal{X}}dx<\infty$. In fact, proving the claim needs Assumption A.1, which excludes the case with $\int_{\mathcal{X}}dx=\infty$.
- Line 895: sub(space → set)
- Line 895: complete(s)

**Reviewer Concerns:**

**Experiments (RMmF, Lm7P, Syar, bFyf):** Wall-clock comparison has been added (Tables 1, 4). Accuracy versus number of function evaluation plot is shown (Figure 4). Larger-scale experiments were conducted (Table 2).

**Estimation versus integration errors (Syar):** The author response explained that the baselines suffer large integration errors, that the proposal (OS-DRE) has zero integration errors by design, and that the model architecture of the proposal reduces estimation errors. Although it does not seem that any changes were made in the mansucript, I think that it would be worth mentioning these points.

**Sobolev (LTmZ):** The authors in their response claimed that Sobolev smoothness is not required. However, the Sobolev assumption is actually there, in Corollary 3.5, and was used in the proof shown in Appendix A.6 to move from the second line to the third in equation (29) via integration by parts. I thus think that the author response was not appropriate.

**Reviewer Scores:**

The initial evaluations of Reviewers RMmF, LTmZ, Lm7P, bFyf were on the positive side of the acceptance threshold, and Reviewer LTmZ wrote that (s)he will even raise the score. Although Reviewer Syar's initial evaluation was on the negative side, (s)he also wrote to increase the score. Although the final eavluation of the five reviewers would have been all positive, because of the above concern on the novelty of the theoretical contributions, I would not be able to recommend acceptance of this paper in its current form.

---

### Decision · Program_Chairs · 2026-01-26

Reject